# Reinforcement Learning for Causal Discovery without Acyclicity Constraints

**Bao Duong**                                                    *b.duong@deakin.edu.au*
*Applied Artificial Intelligence Institute (A²I²), Deakin University*

**Hung Le**                                                       *thai.le@deakin.edu.au*
*Applied Artificial Intelligence Institute (A²I²), Deakin University*

**Biwei Huang**                                                     *bih007@ucsd.edu*
*University of California, San Diego*

**Thin Nguyen**                                               *thin.nguyen@deakin.edu.au*
*Applied Artificial Intelligence Institute (A²I²), Deakin University*

## Abstract

Recently, reinforcement learning (RL) has proved a promising alternative for conventional local heuristics in score-based approaches to learning directed acyclic causal graphs (DAGs) from observational data. However, the intricate acyclicity constraint still challenges the efficient exploration of the vast space of DAGs in existing methods. In this study, we introduce **ALIAS** (reinforced dAg Learning wIthout Acyclicity conStraints), a novel approach to causal discovery powered by the RL machinery. Our method features an efficient policy for generating DAGs in just a single step with an optimal quadratic complexity, fueled by a novel parametrization of DAGs that directly translates a continuous space to the space of all DAGs, bypassing the need for explicitly enforcing acyclicity constraints. This approach enables us to navigate the search space more effectively by utilizing policy gradient methods and established scoring functions. In addition, we provide compelling empirical evidence for the strong performance of **ALIAS** in comparison with state-of-the-arts in causal discovery over increasingly difficult experiment conditions on both synthetic and real datasets. Our implementation is provided at `https://github.com/baosws/ALIAS`.

## 1  Introduction

The knowledge of causal relationships is crucial to understanding the nature in many scientific sectors (Sachs et al., 2005; Hünermund & Bareinboim, 2023; Cao et al., 2019). This is especially relevant in intricate situations where randomized experiments are impractical, and therefore, over the last decades, it has motivated the development of causal discovery methods that aim to infer cause-effect relationships from purely passive data. Causal discovery is typically formulated as finding the directed acyclic graph (DAG) representing the causal model that most likely generated the observed data. Among the broad literature, score-based methods are one of the most well-recognized approaches, which assigns each possible DAG $\mathcal{G}$ a "score" $\mathcal{S}(\mathcal{D}, \mathcal{G})$ quantifying how much it can explain the observed data $\mathcal{D}$, and then optimize the score over the space of DAGs:

$$\mathcal{G}^{\star} = \underset{\mathcal{G} \in \text{DAGs}}{\arg\max}\ \mathcal{S}(\mathcal{D}, \mathcal{G}). \tag{1}$$

Solving this optimization problem is generally NP-hard (Chickering, 1996), due to the *huge combinatorial search space* that grows super-exponentially with the number of variables (Robinson, 1977) and the *intricate acyclicity constraint* that is difficult to characterize and maintain efficiently because of its combinatorial nature. Most methods therefore resort to local heuristics, such as GES (Chickering, 2002) which gradually adds edges into a graph one-by-one while laboriously maintaining acyclicity. With the introduction of soft DAG

Table 1: **Positioning ALIAS among the score-based causal discovery literature.**

| Search type[†] | Method (year) | Search space | Generation steps | Generation complexity | Constraint[‡] | Acyclicity assurance[♭] | Nonlinear data | Differentiable score not required |
|---|---|---|---|---|---|---|---|---|
| Local | GES (2002) (Chickering, 2002) | DAGs | | | Hard | ✓ | ✓ | ✓ |
| | NOTEARS (2020) (Zheng et al., 2018; 2020) | Graphs | | | Soft | ✗ | ✓ | ✗ |
| | NOCURL (2021) (Yu et al., 2021) | DAGs | - | - | None | ✓ | ✗ | ✗ |
| | DAGMA (2022) (Bello et al., 2022) | Graphs | | | Soft | ✗ | ✓ | ✗ |
| | BaDAG (2023) (Annadani et al., 2023) | DAGs | | | None | ✓ | ✓ | ✗ |
| | COSMO (2024) (Massidda et al., 2024) | Graphs | | | None | ✗ | ✓ | ✗ |
| Global | RL-BIC (2020) (Zhu et al., 2020) | Graphs | Single | Quadratic | Soft | ✗ | ✓ | ✓ |
| | BCD-Nets (2021) (Cundy et al., 2021) | DAGs | Single | Cubic | None | ✓ | ✗ | ✗ |
| | CORL (2021) (Wang et al., 2021) | Orderings | Multiple (Autoregressive) | Cubic | Hard | ✓ | ✓ | ✓ |
| | DAG-GFN (2022) (Deleu et al., 2022) | DAGs | Multiple (Autoregressive) | Cubic | Hard | ✓ | ✗ | ✓ |
| | GARL (2023) (Yang et al., 2023b) | Orderings | Multiple (Autoregressive) | Cubic | Hard | ✓ | ✓ | ✓ |
| | RCL-OG (2023) (Yang et al., 2023a) | Orderings | Multiple (Autoregressive) | Cubic | Hard | ✓ | ✓ | ✓ |
| | **ALIAS** (Ours) | DAGs | Single | Quadratic | None | ✓ | ✓ | ✓ |

[†]Local methods start with an initial graph and update it every iteration, while global methods typically concern with DAG generation parameters.

[‡]Methods with Hard constraints explicitly identify and discard the actions that lead to cycles, while Soft constraints refer to the use of DAG regularizers.

[♭]Methods that guarantee acyclicity only in an annealing limit are considered as do not ensure acyclicity.

characterizations (Zheng et al., 2018; Yu et al., 2019; Zhang et al., 2022; Bello et al., 2022), the combinatorial optimization problem above is relaxed to a continuous optimization problem, allowing for exploring graphs more effectively, as multiple edges can be added or removed simultaneously in an update. Alternatively, interventional causal discovery methods exploit available interventional data to identify the causal graph (Hauser & Bühlmann, 2012; Brouillard et al., 2020; Lippe et al., 2022). However, our focus in this study is the challenging observational causal discovery setting where no interventional data is accessible.

Recently, reinforcement learning (RL) has emerged into score-based causal discovery (Zhu et al., 2020; Wang et al., 2021; Yang et al., 2023a;b) as the improved search strategy, thanks to its exploration and exploitation abilities. However, existing RL-based methods handle acyclicity either by fusing the soft DAG regularization from Zheng et al. (2018) into the reward (Zhu et al., 2020), which wastes time for exploring non-DAGs but still does not prohibit all cycles (Wang et al., 2021), or designing *autoregressive policies* (Wang et al., 2021; Deleu et al., 2022; Yang et al., 2023a;b; Deleu et al., 2024) that hinder parallel DAG generation and necessitates learning the transition policies over a multitude of discrete state-action combinations.

In this study, we address the aforementioned limitations of score-based causal discovery methods with a novel RL approach, named **ALIAS** (reinforced d**A**g Learning w**I**thout **A**cyclicity con**S**traints). Our approach employs a generative policy that is capable of generating DAGs in a single-step fashion without any acyclicity regularization or explicit acyclicity maintenance. This enables us to effectively explore and exploit the full DAG space with arbitrary score functions, rather than the restricted ordering space. Specifically, we make the following contributions in this study:

1. At the core of **ALIAS**, taking inspirations from NoCurl (Yu et al., 2021) and subsequent works (Massidda et al., 2024; Annadani et al., 2023), we design **Vec2DAG**, a *surjective map* from a continuous domain into the space of all DAGs. We prove that given a fixed number of nodes, this function can translate an unconstrained real-valued vector into a binary matrix that represents a valid DAG, and vice versa–there always exists a vector mapped to every possible DAGs.

2. Thanks to **Vec2DAG**, we are able to devise a policy outputting actions in the continuous domain that are directly associated with high-reward DAGs. The policy is one-step, unconstrained, and costs only a quadratic number of parallel operations w.r.t. the number of nodes, allowing our agent to explore the DAG space very effectively with arbitrary RL method and scoring function. To our knowledge, **ALIAS** is the first score-based causal discovery method based on RL that can explore the exact space of DAGs with an efficient one-step generation, rendering it an efficient realization of Eq. (1).

3. We demonstrate the effectiveness of the proposed **ALIAS** method in comparison with various state-of-the-arts on a systematic set of numerical evaluations on both synthetic and real-world datasets. Empirical evidence shows that our method consistently surpasses all state-of-the-art baselines under multiple evaluation metrics on varying degrees of nonlinearity, dimensionality, graph density, and model misspecification. For example, our method can achieve an $SHD = 0.2 \pm 0.2$ on very dense graphs with 30 nodes and 8 parents per node on average, and on large graphs with 200 nodes and 400 edges on average, **ALIAS** can still obtain a very low SHD of $2.0 \pm 0.9$.

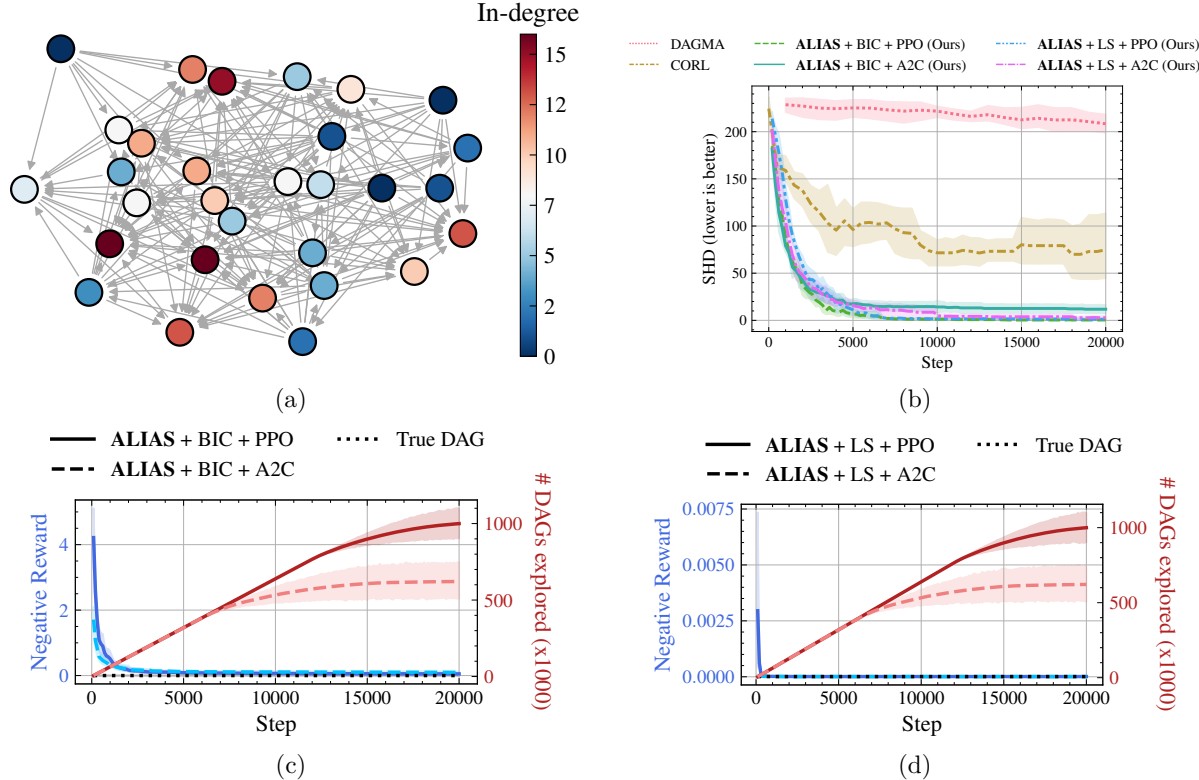

Figure 1: Using merely observational data, the proposed **ALIAS** method **correctly identifies all edges** of (a) a very complex causal dataset with extremely dense connections (linear-Gaussian data with Erdős-Rényi graph of 30 nodes and expected in-degree of 8). (b) We evaluate DAG learning performance in Structural Hamming Distance (SHD, lower is better) on 5 of such datasets with respect to the first 20 000 training steps of four **ALIAS** variants, by combining scoring functions Bayesian Information Criterion (BIC) & Least Squares (LS) with RL methods PPO (Schulman et al., 2017) & A2C (Mnih et al., 2016), in comparison with the best baselines in this setting, namely CORL (Wang et al., 2021) and DAGMA (Bello et al., 2022) (as evaluated in Section 6.2). The best method in this scenario is our **ALIAS** + BIC + PPO variant with zero SHD at the end of the learning process. (c) & (d) For both scores, our method's rewards always approach those of the ground truth DAGs very sharply, which is made possible largely thanks to our efficient DAG parameterization, as well as the continuous exploitation and exploration of the RL algorithms, especially PPO.

We summarize the advantages of **ALIAS** compared with the state-of-the-arts in causal discovery in Table 1. In addition, Figure 1 shows a snapshot of **ALIAS**'s strong performance in a case of highly complex structures, in which our method can achieve absolute accuracy, while the best baselines in this setting still struggle.

## 2 Related Work

**Constraint-based[1] methods** like PC, FCI (Spirtes & Glymour, 1991; Spirtes et al., 2000) and RFCI (Colombo et al., 2012) form a prominent class of causal discovery approaches. They first exploit conditional independence relationships statistically exhibited in data via a series of hypothesis tests to recover the skeleton, which is the undirected version of the DAG, and then orient the remaining edges using probabilistic graphical rules. However, their performance heavily relies on the quality of the conditional independence tests, which can deteriorate rapidly with the number of conditioning variables (Ramdas et al., 2015), rendering them unsuitable for large or dense graphs.

---

[1]Note that the term "constraint" here largely refers to statistical constraints, such as conditional independence, while "constraint" in our method refers to the acyclicity enforcement.

**Score-based methods** is another major class of DAG learners, where each DAG is assigned a properly defined score based on its compatibility with observed data, then the DAG learning problem becomes the optimization problem for the DAG yielding the best score. Score-based methods can be further categorized based on the search approach as follows.

**Combinatorial greedy search methods** such as GES (Chickering, 2002) and FGES (Ramsey et al., 2017) resort to greedy heuristics to reduce the search space and enforce acyclicity by adding one edge at a time after explicitly checking that it would not introduce any cycle, yet this comes at the cost of the sub-optimality of the result.

**Continuous optimization methods** improve upon combinatorial optimization methods in scalability by the ingenious smooth acyclicity constraint, introduced and made popular with NOTEARS (Zheng et al., 2018), which turns said combinatorial optimization into a continuous optimization problem. This enables bypassing the adversary between combinatorial enumeration and scalability to allow for exploring the DAG space much more effectively, where multiple edges can be added or removed in an update. Following developments, e.g., Yu et al. (2019); Lee et al. (2020); Zheng et al. (2020); Ng et al. (2020); Yu et al. (2021); Zhang et al. (2022); Wei et al. (2020) contribute to extending and improving the soft DAG characterization in scalability and convergence. Notably, unconstrained DAG parameterizations are also proposed by Yu et al. (2021) and Massidda et al. (2024), which simplify the optimization problem from a constrained to an unconstrained problem. However, continuous optimization methods restrict the choices of the score to be differentiable functions, which exclude many well-studied scores such as BIC, BDe, MDL, or independence-based scores (Bühlmann et al., 2014)..

**Reinforcement learning methods** have emerged in recent years as the promising replacement for the greedy search heuristics discussed so far, thanks to its search ability via exploration and exploitation. As the pioneer in this line of work, Zhu et al. (2020) introduced the first RL agent that is trained to generate high-reward graphs. To handle acyclicity, they incorporate the soft DAG constraint from Zheng et al. (2018) into the reward function to penalize cyclic graphs. Unfortunately, this may not discard all cycles in the solution, but also increase computational cost drastically due to the unnecessary reward calculations for non-DAGs. To mitigate this issue, subsequent studies (Wang et al., 2021; Yang et al., 2023b;a) turn to finding the best-scoring causal ordering instead and subsequently apply variable selection onto the result to obtain a DAG, which naturally relieves our concerns with cycles. More particularly, CORL (Wang et al., 2021) is the first RL method operating on the ordering space, which defines states as incomplete permutations and actions as the element to be added next. GARL (Yang et al., 2023b) is proposed to enhance ordering generation by exploiting prior structural knowledge with the help of graph attention networks. Meanwhile, RCL-OG (Yang et al., 2023a) introduces a notion of order graph that drastically reduces the state space size from $\mathcal{O}(d!)$ to only $\mathcal{O}(2^d)$. It is also worth noting that the emerging generative flow networks (GFlowNet, Bengio et al., 2021; 2023) offer another technique for learning (distributions of) DAGs (Deleu et al., 2022; 2024), in which the generation of DAGs is also viewed as a sequential generation problem, where edges are added one-by-one with explicit exclusions of edges introducing cycles, and the transition probabilities are learned via flow matching objectives. However, the generation of these orderings and DAGs are usually formulated as a Markov decision process, in which elements are iteratively added to the structure in a multiple-step fashion, which prevents efficient concurrent DAG generations and requires learning the transition functions, which is computationally involved given the multitude of discrete state-action combinations.

## 3 Background

### 3.1 Functional Causal Model

Let $\mathbf{X} = (X_1, \ldots, X_d)^\top$ be the $d$-dimensional random (column) vector representing the variables of interest, $\mathbf{x}^{(k)} = \left(x_1^{(k)}, \ldots, x_d^{(k)}\right)^\top \in \mathbb{R}^d$ denotes the $k$-th observation of $\mathbf{X}$, and $\mathcal{D} = \left\{\mathbf{x}^{(k)}\right\}_{k=1}^n$ indicates the observational dataset containing $n$ i.i.d. samples of $\mathbf{X}$. Assuming *causal sufficiency*, that is, there are no unobserved

endogenous variables, the causal structure among said variables can be described by a DAG $\mathcal{G} = (\mathcal{V}, \mathcal{E})$ where each vertex $i \in \mathcal{V} = \{1, \ldots, d\}$ corresponds to a random variable $X_i$, and each edge $(j \to i) \in \mathcal{E} \subset \mathcal{V} \times \mathcal{V}$ implies that $X_j$ is a direct cause of $X_i$. We also denote the set of all direct causes of a variable as its *parents*, i.e., $\mathrm{pa}_i = \{j \in \mathcal{V} \mid (j \to i) \in \mathcal{E}\}$. The DAG $\mathcal{G}$ is also represented algebraically with a binary *adjacency matrix* $\mathbf{A} \in \{0, 1\}^{d \times d}$ where the $(i, j)$-th entry is 1 iff $(i \to j) \in \mathcal{E}$. Then, the space of all (adjacency matrices of) DAGs of $d$ nodes is denoted by $\mathbb{D}_d \subset \{0, 1\}^{d \times d}$. We follow the Functional Causal Model framework (FCM, Pearl, 2009) to assume the data generation process as $X_i := f_i\left(\mathbf{X}_{\mathrm{pa}_i}, E_i\right), \; \forall i = 1, \ldots, d$, where the noises $E_i$ are mutually independent. In addition, we also consider *causal minimality* (Peters et al., 2014) to ensure each function $f_i$ is non-constant to any of its arguments. For any joint distribution over $\mathbf{E} = (E_1, \ldots, E_d)$, the functions $\{f_i\}_i^d$ induce a joint distribution over $\mathbf{X}$. The goal of causal discovery is then to recover the acyclic graph $\mathcal{G}$ from empirical samples of $P(\mathbf{X})$.

## 3.2 DAG Scoring

Among multiple DAG scoring functions well-developed in the literature (Schwarz, 1978; Heckerman et al., 1995; Rissanen, 1978), here we focus on the popular Bayesian Information Criterion (BIC) (Schwarz, 1978), which is adopted in many works (Chickering, 2002; Zhu et al., 2020; Wang et al., 2021; Yang et al., 2023a) for its flexibility, computational straightforwardness, and consistency.

More particularly, BIC is a parametric score that assumes a model family for the causal model parameters, e.g., linear-Gaussian, which comes with a set of parameters $\psi$ containing the parameters of the causal mechanisms and noise distribution. This score is used to approximate the likelihood of data given the model after marginalizing out the model parameters using Laplace's approximation (Schwarz, 1978):

$$\ln p(\mathcal{D} \mid \mathcal{G}) = \ln \int p(\mathcal{D} \mid \psi, \mathcal{G}) \, p(\psi \mid \mathcal{G}) \, \mathrm{d}\psi \approx \frac{\mathcal{S}_{\mathrm{BIC}}}{2}. \tag{2}$$

Given a DAG $\mathcal{G}$, the BIC score is defined generally as follows:

$$\mathcal{S}_{\mathrm{BIC}}(\mathcal{D}, \mathcal{G}) = 2 \ln p\left(\mathcal{D} \mid \hat{\psi}, \mathcal{G}\right) - |\mathcal{G}| \ln n, \tag{3}$$

where $\hat{\psi}$ is the maximum-likelihood estimator of $p(\mathcal{D} \mid \psi, \mathcal{G})$, $|\mathcal{G}|$ is the number of edges in $\mathcal{G}$, and $n$ is the number of samples in $\mathcal{D}$.

The BIC is consistent in the sense that if a causal model is identifiable, asymptotically, the true DAG has the highest score among all other DAGs (Haughton, 1988). Meanwhile, for limited samples, it prevents overfitting by penalizing edges that do not improve the log-likelihood significantly. More formally:

**Lemma 1.** *Let $\mathcal{G}^*$ be the ground truth DAG of an identifiable SCM satisfying causal minimality (Peters et al., 2014) (i.e., there are no redundant edges) inducing the dataset $\mathcal{D}$, and let $n$ be the sample size of $\mathcal{D}$. Then, in the limit of large $n$, $\mathcal{S}_{BIC}(\mathcal{D}, \mathcal{G}^*) > \mathcal{S}_{BIC}(\mathcal{D}, \mathcal{G})$ for any $\mathcal{G} \neq \mathcal{G}^*$.*

The proof can be found in Appendix B.1.

As an example, for additive noise models (ANM) $X_i := f_i\left(\mathbf{X}_{\mathrm{pa}_i}\right) + E_i, \; \forall i = 1, \ldots, d$ with Gaussian noises $E_i \sim \mathcal{N}\left(0; \sigma_i^2\right)$, the BIC-based score can be specified as

$$\mathcal{S}_{\mathrm{BIC-NV}}(\mathcal{D}, \mathcal{G}) = -(n \sum_{i=1}^{d} \ln \frac{\mathrm{SSR}_i}{n} + |\mathcal{G}| \ln n),$$

where $\mathrm{SSR}_i = \sum_{k=1}^{n} (\hat{x}_i^{(k)} - x_i^{(k)})^2$ is the sum of squared residuals after regressing $X_i$ on its parents in $\mathcal{G}$, and we adopt the convention that $|\mathcal{G}|$ is the number of edges in $\mathcal{G}$. Additionally assuming equal noise variances gives us with

$$\mathcal{S}_{\mathrm{BIC-EV}}(\mathcal{D}, \mathcal{G}) = -(nd \ln \frac{\sum_{i=1}^{d} \mathrm{SSR}_i}{nd} + |\mathcal{G}| \ln n).$$

The derivations of BIC scores are presented in Appendix A. A simpler yet widely adopted alternative is the least squares (LS) (Zheng et al., 2018; Lachapelle et al., 2020; Yu et al., 2021; Bello et al., 2022; Massidda et al., 2024). With an additional $l_0$ regularization, we define the LS score as $\mathcal{S}_{\mathrm{LS}}(\mathcal{D}, \mathcal{G}) = -(\sum_{i=1}^{d} \mathrm{SSR}_i + \lambda_0 |\mathcal{G}|)$, where $\lambda_0 \geq 0$ is a hyper-parameter for penalizing dense graphs. In our empirical studies, following common practices (Zhu et al., 2020; Wang et al., 2021; Yang et al., 2023b;a), linear regression is used for linear data and Gaussian process regression is adopted for nonlinear data. That being said, any valid regression technique can be seamlessly integrated into our method.

### 3.3 DAG Representations

Typically, to search over the space of DAGs, modern causal discovery methods either optimize over directed graphs with differentiable DAG regularizers (Zheng et al., 2018; Yu et al., 2019; Lee et al., 2020; Zheng et al., 2020; Ng et al., 2020; Wei et al., 2020; Zhu et al., 2020; Zhang et al., 2022), or search over causal orderings and then apply variable selection to suppress redundant edges (Cundy et al., 2021; Charpentier et al., 2022; Chen et al., 2019; Wang et al., 2021; Rolland et al., 2022; Sanchez et al., 2023; Yang et al., 2023b). The former approach does not guarantee the acyclicity of the returned graph with absolute certainty, while the latter approach faces challenges in efficiently generating permutations. For instance, in Cundy et al. (2021) the permutation matrix representing the causal ordering is parametrized by the Sinkhorn operator (Sinkhorn, 1964) followed by the Hungarian algorithm (Kuhn, 1955) with a considerable cost of $\mathcal{O}(d^3)$. Other examples include ordering-based RL methods (Wang et al., 2021; Yang et al., 2023a) that cost at least $\mathcal{O}(d^2)$ just to generate a single ordering element, thus totaling an $\mathcal{O}(d^3)$ complexity for generating a DAG.

Our work takes inspiration from Yu et al. (2021), where a novel unconstrained characterization of *weighted* adjacency matrices of DAGs is proposed. Particularly, a "node potential" vector $\mathbf{p} \in \mathbb{R}^d$ is introduced to model an *implicit* causal ordering, where $i$ precedes $j$ if $p_j > p_i$. Hence, the weight matrix

$$\mathbf{A} = \mathbf{W} \odot \mathrm{ReLU}(\mathrm{grad}(\mathbf{p})), \tag{4}$$

where $\mathbf{W} \in \mathbb{R}^{d \times d}$ and $\mathrm{grad}(\mathbf{p})_{ij} := p_j - p_i$ is the gradient flow operator (Lim, 2020), can be shown to correspond to a valid DAG (Theorem 2.1, Yu et al., 2021). However, this weight matrix is only applicable for linear models, and the representation is only used as a refinement for the result returned by a constrained optimization problem. An alternative to this characterization is recently introduced by Massidda et al. (2024), where $\mathbf{A} = \mathbf{W} \odot \mathrm{sigmoid}(\mathrm{grad}(\mathbf{p})/\tau)$, yet this approach only ensures acyclicity at the limit of the annealing temperature $\tau \to 0^+$, which is usually not exactly achieved in practice. Additionally, the equivalent DAG formulation of Annadani et al. (2023) uses the node potential $\mathbf{p}$ to represent a smooth *explicit* permutation matrix: $\sigma(\mathbf{p}) := \lim_{\tau \to 0^+} \mathrm{Sinkhorn}(\mathbf{p} \cdot [1 \dots d]/\tau)$, again necessitating a temperature scheduler and the expensive Sinkhorn operator, which reportedly requires 300 iterations to converge and an $\mathcal{O}(d^3)$ complexity of the Hungarian algorithm, for generating a single DAG.

## 4 Vec2DAG: Unconstrained Parametrization of DAGs

### 4.1 The Vec2DAG Operator

Extending from the formulation in Eq. (4), we design a deterministic translation from an unconstrained continuous space to the space of general *binary* adjacency matrices of all DAGs, not restricted to linear models. To be more specific, in addition to the node potential vector $\mathbf{p} \in \mathbb{R}^d$, we introduce a strictly upper-triangular "edge potential" matrix $\mathbf{E} \in \mathbb{R}^{d \times d}$, which can be described using $\frac{d \cdot (d-1)}{2}$ parameters. We then combine them with $\mathbf{p}$ to create a unified representation vector $\mathbf{z} \in \mathbb{S}_d = \mathbb{R}^{d \cdot (d+1)/2}$, which is the parameter space of all $d$-node DAGs in our method. Furthermore, we denote by $\mathbf{p}(\mathbf{z})$ and $\mathbf{E}(\mathbf{z})$ the node and edge potential components constituting $\mathbf{z}$, respectively. Specifically, $\mathbf{p}(\mathbf{z})$ represents the node potential vector formed by the first $d$ elements of $\mathbf{z}$, while $\mathbf{E}(\mathbf{z})$ is the edge potential matrix, with the elements above the main diagonal derived from the last $\frac{d \cdot (d-1)}{2}$ elements of $\mathbf{z}$ (see our code in Figure 5). Then, our unconstrained DAG parametrization $\mathbf{Vec2DAG}_d$ for $d$ nodes can be defined as follows.

**Definition 1.** For all $d \in \mathbb{N}^+$ and $\mathbf{z} \in \mathbb{S}_d$:

$$\textbf{Vec2DAG}_d\left(\mathbf{z}\right) := H\left(\mathbf{E}\left(\mathbf{z}\right) + \mathbf{E}\left(\mathbf{z}\right)^{\top}\right) \odot H\left(\text{grad}\left(\mathbf{p}\left(\mathbf{z}\right)\right)\right), \tag{5}$$

where $H\left(x\right) := \begin{cases} 1 & \text{if } x > 0, \\ 0 & \text{otherwise.} \end{cases}$ is known as the Heaviside step function and $\odot$ is the Hadamard (element-wise) product operator.

The intuition behind Vec2DAG is that the first term in Eq. (5) defines a symmetric binary adjacency matrix, determining whether two nodes are connected. The directions of these connections are then dictated by the second term in Eq. (5), resulting in a binary matrix that represents a directed graph. Additionally, this directed graph is guaranteed to be acyclic due to the use of the gradient flow operator.

The procedure to sample a DAG is then denoted as $\mathbf{z} \sim P\left(\mathbf{z}\right)$, $\mathbf{A} = \textbf{Vec2DAG}_d\left(\mathbf{z}\right)$, which can be implemented in a few lines of code, as illustrated in Figure 5 of the Appendix. The validity of our parametrization is justified by the following theorem.

**Theorem 1.** *For all $d \in \mathbb{N}^{+}$, let $\textbf{Vec2DAG}_d : \mathbb{S}_d \to \{0,1\}^{d \times d}$ be defined as in Eq. (5). Then, $\text{Im}\left(\textbf{Vec2DAG}_d\right) = \mathbb{D}_d$, where $\text{Im}\left(\cdot\right)$ is the Image operator, and $\mathbb{D}_d$ is the space of all d-node DAGs.*

The proof can be found in Appendix B.2. Our formulation directly represents a DAG by a real-valued vector, which is in stark contrary to existing unconstrained methods that only aim for a DAG sampler (Cundy et al., 2021; Wang et al., 2021; Charpentier et al., 2022; Deleu et al., 2022; Annadani et al., 2023; Yang et al., 2023b;a). More notably, this approach requires **no temperature annealing** like in Massidda et al. (2024); Annadani et al. (2023), and can generate a valid DAG in a **single step** since sampling $\mathbf{z}$ can be done instantly in an unconstrained manner. In addition, this merely costs $\mathcal{O}\left(d^2\right)$ **parallelizable operations** compared with the $\mathcal{O}\left(d^3\right)$ cost of sequentially generating permutations using the Sinkhorn operator (Cundy et al., 2021; Charpentier et al., 2022; Annadani et al., 2023) and multiple-step RL methods (Wang et al., 2021; Yang et al., 2023b;a). Moreover, our generation technique is one-step, and thus **does not require learning any transition function**, which vastly reduces the computational burden compared with RL methods based on sequential decisions.

### 4.2 Properties of Vec2DAG

In this section, we show that our parameterization **Vec2DAG** has some important additional properties that set it apart from past formulations.

**Lemma 2.** *(Scaling and Translation Invariance). For all $d \in \mathbb{N}^{+}$, let $\textbf{Vec2DAG}_d : \mathbb{S}_d \to \{0,1\}^{d \times d}$ be defined as in Eq. (5). Then, for all $\mathbf{z} \in \mathbb{S}_d$, $\alpha > 0$, and $\boldsymbol{\beta} \in \mathbb{S}_d$ such that $\left|\mathbf{p}\left(\boldsymbol{\beta}\right)_i\right| < 1/2 \min_j \left|\mathbf{p}\left(\mathbf{z}\right)_i - \mathbf{p}\left(\mathbf{z}\right)_j\right|$ and $\left|\mathbf{E}\left(\boldsymbol{\beta}\right)_{ij}\right| < \left|\mathbf{E}\left(\mathbf{z}\right)_{ij}\right| \forall i, j$, we have $\textbf{Vec2DAG}_d\left(\mathbf{z}\right) = \textbf{Vec2DAG}_d\left(\alpha \cdot \left(\mathbf{z} + \boldsymbol{\beta}\right)\right)$.*

This insight is proven in Appendix B.3. Intuitively, this indicates that scaling the potential by any positive constant $\alpha$ results in the same DAG (Vec2DAG($\mathbf{z}$) = Vec2DAG($\alpha \cdot \mathbf{z}$)), and translating the potential by an amount $\beta$ (which can be large, provided it does not change the ordering of $\mathbf{p}$ or the element-wise positivity of $\mathbf{E}$) also results in the same DAG (Vec2DAG($\mathbf{z}$) = Vec2DAG($\mathbf{z}+\boldsymbol{\beta}$)). In other words, any DAG can be diversely constructed by infinitely many representations, suggesting a dense parameter space where representations of different DAGs are close to each other. This leads us to the next point, which shows an upper bound of the distance between an arbitrary representation with a representation of any DAG.

**Lemma 3.** *(Proximity between DAGs). Let $\mathbf{z} \in \mathbb{S}_d$. Then, for any DAG $\mathbf{A} \in \mathbb{D}_d$ and $\epsilon > 0$, there exists $\mathbf{z_A}$ in the unit ball $B\left(\infty; \|\mathbf{z}\|_{\infty} + \epsilon\right)$ around $\mathbf{z}$ such that $\textbf{Vec2DAG}_d\left(\mathbf{z_A}\right) = \mathbf{A}$.*

We provide the proof in Appendix B.4. This property is not straightforward in existing constrained optimization approaches (Zheng et al., 2018; Lee et al., 2020; Zheng et al., 2020; Lachapelle et al., 2020; Bello et al., 2022), and suggests that the true DAG may be found closer to the initial position if we start from a smaller scale in our framework. We leverage this result in our implementation by restricting $\mathbb{S}_d$ to a hypercube $[-\gamma, \gamma]^{d \cdot (d+1)/2}$ with a relatively small $\gamma = 10$. This has the effect of regularizing the search space

but still does not invalidate our Theorem 1, i.e., we can still reach every possible DAGs when searching in this restricted space.

# 5 ALIAS: Reinforced DAG Learning without Acyclicity Constraints

## 5.1 Motivation for Reinforcement Learning

Using the **Vec2DAG** representation, the score-based causal discovery problem may seem to simplify into a maximization problem: $\mathbf{z}^* = \arg\max\limits_{\mathbf{z} \in \mathbb{R}^{d \cdot (d+1)/2}} \mathcal{S}_{\text{BIC}}(\mathcal{D}, \textbf{Vec2DAG}(\mathbf{z}))$, which could, in principle, be addressed to certain extents using off-the-shelf black-box optimization techniques such as Bayesian optimization, which is one of the most popular blackbox optimization methods. However, solving this optimization problem is far from straightforward due to the high dimensionality of the search space, which grows quadratically with the number of nodes (e.g., for 30 nodes, the space is 465-dimension).

Since RL is the only black-box optimization approach that has been studied in the score-based causal discovery literature (at the time writing this manuscript, to the best of our knowledge), we align with the established line of works (Zhu et al., 2020; Wang et al., 2021; Yang et al., 2023a;b) to specifically focus on leveraging RL to solve this optimization problem. The idea is that an RL agent with a stochastic policy can autonomously decide where to explore based on the uncertainty of the learned policy, which is continuously updated through incoming reward signals (Zhu et al., 2020). That said, we note that, our method, which only involves one-step trajectories as shown below, is better perceived as a policy gradient approach rather than a general RL method that requires a multi-step Markov decision process (MDP).

As shown below, policy gradient provides our method with built-in exploration-exploitation capabilities and linear scalability with respect to both dimensionality and sample size, making it a practical choice for this problem. In addition, our policy gradient point of view also enables the adaptability of various established methods, such as vanilla policy gradient (Sutton et al., 1999), A2C (Mnih et al., 2016), and PPO (Schulman et al., 2017), to efficiently optimize our objective, effectively establishing a clear association between our proposed approach and policy gradient in RL.

## 5.2 Policy Gradient for DAG Search

**Policy and Action.** Utilizing RL, we seek for a policy $\pi$ that outputs a continuous action $\mathbf{z} \in \mathbb{S}_d = \mathbb{R}^{d \cdot (d+1)/2}$, which is the parameter space of DAGs of $d$ nodes. In this work, we consider stochastic policies for better exploration, i.e., we parametrize our policy by an isotropic Gaussian distribution with learnable means and variances: $\pi_{\boldsymbol{\theta}}(\mathbf{z}) = \mathcal{N}\left(\mathbf{z}; \boldsymbol{\mu_\theta}, \text{diag}\left(\boldsymbol{\sigma_\theta^2}\right)\right)$. Since our policy generates a DAG representation in just one step, every trajectory starts with the same initial state and terminates after only one transition, so the agent does not need to be aware of the state in our method. We note that, similar to RL-BIC (Zhu et al., 2020), the one-step nature of the environment does not preclude the application of RL in our approach. This is because a one-step environment is simply a special case of a MDP, which remains compatible with most RL algorithms.

**Reward.** The reward of an action in our method is set as the graph score of the DAG induced by that action with respect to the observed dataset $\mathcal{D}$ (Section 3.2), and divided by $n \times d$ to maintain numerical stability without modifying the monotonicity of the score:

$$\mathcal{R}(\mathbf{z}) := \frac{1}{n \times d} \mathcal{S}(\mathcal{D}, \textbf{Vec2DAG}(\mathbf{z})). \tag{6}$$

**Policy Gradient Algorithm.** Since our action space is continuous, we employ policy gradient methods, which are well established for handling continuous actions, rather than the value-based approach as in recent RL-based techniques (Wang et al., 2021; Yang et al., 2023b;a). The training objective is to maximize the expected return defined as $\mathcal{J}(\boldsymbol{\theta}) = \mathbb{E}_{\mathbf{z} \sim \pi_{\boldsymbol{\theta}}}[\mathcal{R}(\mathbf{z})]$.

Under identifiable causal models, causal minimality, and a consistent scoring function, the optimal policy obtained by maximizing this objective will return the true DAG:

**Algorithm 1 ALIAS** with vanilla policy gradient for causal discovery.

---

**Input:** Dataset $\mathcal{D} = \left\{ \mathbf{x}^{(k)} \right\}_{k=1}^{n}$, score function $\mathcal{S}\left(\mathcal{D}, \cdot\right)$, batch size $B$, and learning rate $\eta$.

**Output:** Estimated causal DAG $\hat{\mathcal{G}}$.

1: **while** not terminated **do**

2:    Draw a minibatch of $B$ actions from the policy: $\left\{ \mathbf{z}^{(k)} \sim \pi_{\boldsymbol{\theta}} \right\}_{k=1}^{B}$.

3:    Collect rewards $\left\{ r^{(k)} := \frac{1}{n \times d} \mathcal{S}\left(\mathcal{D}, \mathbf{Vec2DAG}_d\left(\mathbf{z}^{(k)}\right)\right) \right\}_{k=1}^{B}$.      ▷ Sec. 4.1

4:    Update policy as: $\boldsymbol{\theta} := \boldsymbol{\theta} + \eta \left( \frac{1}{B} \sum_{k=1}^{B} \nabla_{\boldsymbol{\theta}} \ln \pi_{\boldsymbol{\theta}}\left(\mathbf{z}^{(k)}\right) \cdot r^{(k)} \right)$.      ▷ Sec. 5.2

5: **end while**

6: $\mathbf{z} \sim \pi_{\theta}$, $\hat{\mathcal{G}} := \mathbf{Vec2DAG}\left(\mathbf{z}\right)$.

7: Post-process $\hat{\mathcal{G}}$ by pruning if needed and return.      ▷ Sec. 5.3

---

**Lemma 4.** *Assuming causal identifiability and causal minimality, that is, there is a unique causal model with no redundant edges that can produce the observed dataset, and BIC score is used to define the reward $\mathcal{R}\left(\mathbf{z}\right)$ as in Eq. (6). Let $n$ be the sample size of the observed dataset $\mathcal{D}$, $\theta^* \in \arg\max_{\theta \in \Theta} \mathbb{E}_{\mathbf{z} \sim \pi_{\boldsymbol{\theta}}} \left[ \mathcal{R}\left(\mathbf{z}\right) \right]$, where $\pi_{\boldsymbol{\theta}}\left(\mathbf{z}\right) = \mathcal{N}\left(\mathbf{z}; \boldsymbol{\mu_{\theta}}, diag\left(\boldsymbol{\sigma_{\theta}^2}\right)\right)$. Then, as $n \to \infty$, $\mathcal{G} = \mathbf{Vec2DAG}\left(\mathbf{z}\right)$ is the true DAG, where $\mathbf{z} \sim \pi_{\theta^*}$.*

The proof is presented in Appendix B.5. The differential entropy of the policy can also be added to the expected return as a regularization term to encourage exploration (Mnih et al., 2016), however we find in our experiments that the stochasticity offered by the policy suffices for exploration. That said, we also investigate the effect of entropy regularization in our empirical studies. During training, the parameter $\boldsymbol{\theta}$ is updated in the direction suggested by the policy gradient algorithm. For example, using vanilla policy gradient, the gradient is given by the policy gradient theorem (Sutton et al., 1999) as: $\nabla_{\boldsymbol{\theta}} \mathcal{J}\left(\boldsymbol{\theta}\right) = \mathbb{E}_{\mathbf{z} \sim \pi_{\boldsymbol{\theta}}} \left[ \nabla_{\boldsymbol{\theta}} \ln \pi_{\boldsymbol{\theta}}\left(\mathbf{z}\right) \cdot \mathcal{R}\left(\mathbf{z}\right) \right]$.

Note that since our trajectories are one-step and our environment is deterministic, the state-action value function is always equal to the immediate reward, and therefore there is no need for a critic to estimate the value function. Hence, vanilla policy gradient works well out-of-the-box for our framework, yet in practice our method can be implemented with more advanced algorithms for improved training efficiency. Our practical implementation considers the basic policy algorithm Advantage Actor-Critic (A2C, Mnih et al., 2016) and a more advanced method Proximal Policy Optimization (PPO, Schulman et al., 2017). In addition, while policy gradient only ensures local convergence under suitable conditions (Sutton et al., 1999), our empirical evidence remarks that our method can reach the exact ground truth DAG in notably many cases.

### 5.3  Post Processing

With limited sample sizes, due to overfitting, redundant edges may still be present in the returned DAG that achieves the highest score. One approach towards suppressing the false discovery rate is to greedily remove edges with non-substantial contributions in the score. For linear models, a standard approach is to threshold the absolute values of the estimated weight matrix $\mathbf{W}$ at a certain level $\delta$ (Zheng et al., 2018; Ng et al., 2020; Bello et al., 2022), i.e., removing all edges $(i \to j)$ with $|\mathbf{W}_{ij}| < \delta$. For nonlinear models, the popular CAM pruning method (Bühlmann et al., 2014) can be employed for generalized additive models (GAMs), which performs a GAM regression on the parents set and exclude the parents that do not pass a predefined significance level. An alternative pruning method that does not depend on the causal model is based on conditional independence (CI), i.e., by imposing Faithfulness (Spirtes et al., 2000), for each $j \in \text{pa}_i$ in the graph found so far, we remove the edge $(j \to i)$ if $X_i \perp\!\!\!\perp X_j \mid X_{\text{pa}_i \setminus \{j\}}$, which is a direct consequence of the Faithfulness assumption and can be realized with available CI tests like KCIT (Zhang et al., 2011). In addition, for the least squares score, we can increase the regularization strength on the number of edges to encourage sparsity during the learning process. Our numerical experiments investigate the effects of all these approaches.

To summarize, Algorithm 1 highlights the key steps of our **ALIAS** method for the case with vanilla gradient policy.

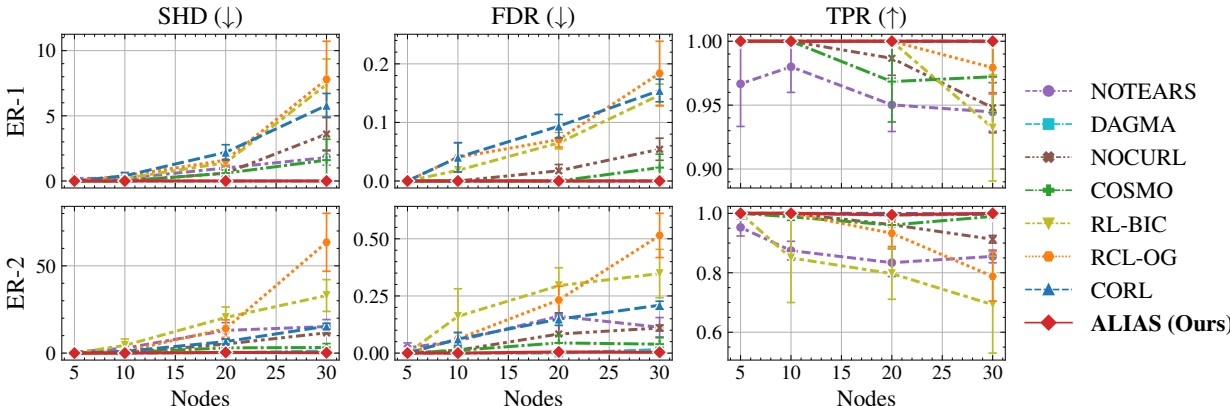

Figure 2: **Causal Discovery Performance on Linear-Gaussian Data.** ER-1 and ER-2 denote Erdős-Rényi graph models with expected in-degrees of 1 and 2, respectively. The weight range is $\mathcal{U}\left(\left[-5, -2\right] \cup \left[2, 5\right]\right)$, which is wider than prior studies, making our setting more challenging due to higher data variance. We compare the proposed **ALIAS** method with NOTEARS (Zheng et al., 2018), DAGMA (Bello et al., 2022), NOCURL (Yu et al., 2021), COSMO (Massidda et al., 2024), RL-BIC (Zhu et al., 2020), CORL (Wang et al., 2021), and RCL-OG (Yang et al., 2023a). The performance metrics are Structural Hamming Distance (SHD, lower is better), False Detection Rate (FDR, lower is better), and True Positive Rate (TPR, higher is better). Shaded areas depict standard errors over 5 independent runs.

## 6 Numerical Evaluations

In this section, we validate our method in the causal discovery task across a comprehensive set of settings, including *different nonlinearities*, *varying graph types*, *sizes and densities*, *varying sample sizes*, as well as *different degrees of model misspecification* on *both synthetic and real data*. In addition, we also analyze the computational efficiency of our method, as well as demonstrate the significance of different components of our method, especially the choice of reinforcement learning as the optimizer, in our extensive ablation studies.

### 6.1 Experiment Setup

We conduct extensive empirical evaluations on both simulated and real datasets, where the ground truth DAGs are available, to compare the efficiency of the proposed **ALIAS** method with up-to-date state-of-the-arts in causal discovery, including the constrained continuous optimization approaches with soft DAG constraints NOTEARS (Zheng et al., 2018; 2020) and DAGMA (Bello et al., 2022), unconstrained continuous optimization approaches NOCURL (Yu et al., 2021) and COSMO (Massidda et al., 2024), as well as three RL-based methods RL-BIC (Zhu et al., 2020), CORL (Wang et al., 2021), and RCL-OG (Yang et al., 2023a). A brief description of these methods along with their implementation details and hyper-parameter specifications are provided in Appendix C and the evaluation metrics are described in Appendix C.2.1. For the main experiments, we use the variant with BIC score and PPO algorithm for our method, and examine other variants in the ablation studies. We report supplementary results, including additional ablation studies, in Appendix D.

### 6.2 Linear Data with Gaussian and non-Gaussian Noises

For a given number of nodes $d$, we first generate a DAG following the Erdős-Rényi graph model (Erdős & Rényi, 1960) with an expected in-degree of $k \in \mathbb{N}^+$, denoted by ER-$k$. Next, edge weights are randomly sampled from the uniform distribution $P\left(\mathbf{W}\right)$, and the noises are drawn from the standard Gaussian $E_i \sim \mathcal{N}\left(0, 1\right)$. To make this setting more challenging, we use a wider range $P\left(\mathbf{W}\right) = \mathcal{U}\left(\left[-5, -2\right] \cup \left[2, 5\right]\right)$ compared with the common range of $\mathcal{U}\left(\left[-2, -0.5\right] \cup \left[0.5, 2\right]\right)$ in previous studies (Zheng et al., 2018; Zhu et al., 2020; Wang et al., 2021; Bello et al., 2022). We then sample $n = 1\,000$ observations for each dataset. This causal model is

Table 2: **Causal discovery performance on dense graphs (30-node ER-8) and high-dimensional graphs (200-node ER-2) with linear-Gaussian data.** The performance metrics are Structural Hamming Distance (SHD, lower is better), False Detection Rate (FDR, lower is better), and True Positive Rate (TPR, higher is better). The numbers are *mean ± standard error* over 5 independent runs. **Bold**: best performance, underline: second-best performance. RL-BIC & CORL fail to run high-dimensional tasks.

| Method | Dense graphs (30 nodes, ≈ 240 edges) | | | High-dimensional graphs (200 nodes, ≈ 400 edges) | | |
|---|---|---|---|---|---|---|
| | SHD ($\downarrow$) | FDR ($\downarrow$) | TPR ($\uparrow$) | SHD ($\downarrow$) | FDR ($\downarrow$) | TPR ($\uparrow$) |
| NOTEARS (Zheng et al., 2018) | $141.2 \pm 11.9$ | $0.25 \pm 0.03$ | $0.55 \pm 0.03$ | $53.8 \pm 6.5$ | $0.06 \pm 0.01$ | $0.93 \pm 0.01$ |
| DAGMA (Bello et al., 2022) | $\underline{67.6 \pm 8.0}$ | $\underline{0.14 \pm 0.02}$ | $0.82 \pm 0.02$ | $\underline{9.6 \pm 2.7}$ | $\underline{0.02 \pm 0.00}$ | $\underline{0.99 \pm 0.00}$ |
| NOCURL (Yu et al., 2021) | $147.6 \pm 5.7$ | $0.32 \pm 0.01$ | $0.63 \pm 0.00$ | $227.6 \pm 17.5$ | $0.20 \pm 0.03$ | $0.59 \pm 0.02$ |
| COSMO (Massidda et al., 2024) | $97.4 \pm 6.8$ | $0.24 \pm 0.01$ | $0.80 \pm 0.02$ | $158.0 \pm 19.5$ | $0.25 \pm 0.03$ | $0.87 \pm 0.02$ |
| RL-BIC (Zhu et al., 2020) | $180.6 \pm 21.7$ | $0.43 \pm 0.06$ | $0.42 \pm 0.14$ | - | - | - |
| CORL (Wang et al., 2021) | $82.4 \pm 22.3$ | $0.23 \pm 0.05$ | $\underline{0.87 \pm 0.04}$ | - | - | - |
| RCL-OG (Yang et al., 2023a) | $199.7 \pm 7.1$ | $0.47 \pm 0.01$ | $0.51 \pm 0.04$ | $1076.6 \pm 28.8$ | $0.89 \pm 0.00$ | $0.32 \pm 0.01$ |
| **ALIAS** (Ours) | $\mathbf{0.2 \pm 0.2}$ | $\mathbf{0.00 \pm 0.00}$ | $\mathbf{1.00 \pm 0.00}$ | $\mathbf{2.0 \pm 0.9}$ | $\mathbf{0.00 \pm 0.00}$ | $\mathbf{1.00 \pm 0.00}$ |

identifiable due to the equal noise variances (Peters et al., 2014). For fairness, we also apply the same pruning procedure with linear regression coefficients thresholded at 0.3 for all methods and use the equal-variance BIC (Section 3.2) for RL-BIC, CORL, RCL-OG, and **ALIAS**.

**Small to moderate graphs.** In Figure 2 we report the causal discovery performance for linear-Gaussian data with small to moderate graph sizes and densities, showing that our method consistently achieves near-perfect performance in all metrics, which can be expected thanks to its ability to explore the DAG space competently. Overall, the closest method with comparable performance to our method in this case is DAGMA, followed by COSMO, which are among the most advanced continuous optimization approaches, while other methods, including RL-based ones, still struggle even in this simplest scenario.In addition, the results on Scale-Free (SF) graphs and the common weight range $\mathcal{U}([-2, -0.5] \cup [0.5, 2])$ can also be found in Appendix D.

**Dense & High-dimensional graphs.** We next test the proposed method's ability to adapt to highly complex scenarios, including the cases with very dense graphs (ER-8 graphs) and larger number of nodes (200-node graphs). This is to demonstrate the advantages of our proposed method over existing approaches that typically struggle on slightly dense graphs of ER-4 at most (Zheng et al., 2018; Yu et al., 2021; Bello et al., 2022; Massidda et al., 2024) or small graphs of only tens of nodes (Zhu et al., 2020; Yang et al., 2023a). In this case, we use the common weight range of $\mathcal{U}([-2, -0.5] \cup [0.5, 2])$ to avoid numerical instabilities due to more complex graphs. Table 2 depicts that for dense graphs, **ALIAS** makes almost no mistake while the best baseline in this case, which is DAGMA, still has a significantly large SHD of $67.6 \pm 8.0$. The performance gap is narrower in the high-dimensional setting with 200 nodes, yet our method remains the leading approach with an SHD of only 2, compared with an SHD of nearly 10 for the second-best method DAGMA.

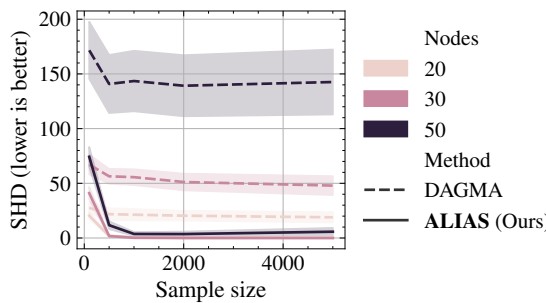

Figure 3: **Causal Discovery performance (linear-Gaussian data on ER-8 graphs) as function of sample size** ($100$ **to** $5\,000$). Shaded areas depict standard errors over 5 independent runs.

**Effect of sample size.** We further investigate the behavior of **ALIAS** under data scarcity and redundancy. We again consider the difficult configuration of ER-8 graphs, and vary the sample size from very limited ($100$) to redundant ($5\,000$) in Figure 3, where it is shown that our method with just 100 samples can surpass DAGMA even with $5\,000$ samples.

Table 3: **Causal discovery performance under noise misspecification on linear data with 30-node ER-2 graphs.** The numbers are *mean ± standard error* over 5 runs. **Bold**: best performance, underline: second-best performance.

| | SHD (lower is better) | | | |
|---|---|---|---|---|
| **Method\Noise** | Exp$(1)$ | Gumbel$(0,1)$ | Laplace$(0,1)$ | Uniform$(-1,1)$ |
| NOTEARS (Zheng et al., 2018) | $6.0 \pm\ 1.6$ | $4.0 \pm\ 1.9$ | $3.0 \pm\ 1.4$ | $6.4 \pm\ 4.3$ |
| DAGMA (Bello et al., 2022) | $\underline{1.0 \pm\ 1.0}$ | $\mathbf{0.2 \pm\ 0.2}$ | $\underline{1.0 \pm\ 1.0}$ | $\underline{4.8 \pm\ 1.5}$ |
| NOCURL (Yu et al., 2021) | $10.8 \pm\ 0.8$ | $6.6 \pm\ 1.7$ | $4.2 \pm\ 1.0$ | $29.0 \pm\ 3.0$ |
| COSMO (Massidda et al., 2024) | $5.4 \pm\ 2.0$ | $5.0 \pm\ 2.3$ | $7.6 \pm\ 2.5$ | $5.6 \pm\ 1.8$ |
| RL-BIC (Zhu et al., 2020) | $66.8 \pm 13.2$ | $34.4 \pm 9.5$ | $31.4 \pm 10.1$ | $31.4 \pm 11.5$ |
| CORL (Wang et al., 2021) | $12.0 \pm\ 1.4$ | $15.4 \pm\ 2.4$ | $15.6 \pm 2.2$ | $15.0 \pm\ 0.9$ |
| RCL-OG (Yang et al., 2023a) | $77.3 \pm 13.5$ | $79.8 \pm 22.8$ | $41.3 \pm 16.4$ | $57.0 \pm 23.1$ |
| **ALIAS (Ours)** | $\mathbf{0.4 \pm\ 0.3}$ | $\underline{0.4 \pm\ 0.4}$ | $\mathbf{0.8 \pm\ 0.4}$ | $\mathbf{0.4 \pm 0.3}$ |

Table 4: **Causal discovery performance on nonlinear data with Gaussian processes on 10-node ER-4 graphs.** The performance metrics are Structural Hamming Distance (SHD, lower is better), False Detection Rate (FDR, lower is better), and True Positive Rate (TPR, higher is better). The numbers are *mean ± standard error* over 5 runs. **Bold**: best performance, underline: second-best performance. Since the graphs are dense and the noise is additive, we also study the effect of pruning the output graphs with CAM pruning (Bühlmann et al., 2014).

| | No Pruning | | | CAM Pruning | | |
|---|---|---|---|---|---|---|
| **Method** | **SHD ($\downarrow$)** | **FDR ($\downarrow$)** | **TPR ($\uparrow$)** | **SHD ($\downarrow$)** | **FDR ($\downarrow$)** | **TPR ($\uparrow$)** |
| NOTEARS (Zheng et al., 2020) | $28.4 \pm 1.4$ | $0.33 \pm 0.06$ | $0.33 \pm 0.04$ | $28.6 \pm 1.0$ | $0.32 \pm 0.06$ | $0.32 \pm 0.04$ |
| DAGMA (Bello et al., 2022) | $25.8 \pm 1.7$ | $0.32 \pm 0.04$ | $0.40 \pm 0.05$ | $26.0 \pm 1.8$ | $0.31 \pm 0.04$ | $0.39 \pm 0.06$ |
| NOCURL (Yu et al., 2021) | $35.2 \pm 0.9$ | $0.47 \pm 0.09$ | $0.15 \pm 0.04$ | $35.0 \pm 0.8$ | $0.46 \pm 0.08$ | $0.15 \pm 0.04$ |
| COSMO (Massidda et al., 2024) | $26.4 \pm 2.0$ | $0.30 \pm 0.04$ | $0.39 \pm 0.04$ | $27.0 \pm 2.5$ | $0.28 \pm 0.04$ | $0.35 \pm 0.05$ |
| RL-BIC (Zhu et al., 2020) | $39.0 \pm 2.0$ | $\underline{0.06 \pm 0.06}$ | $0.05 \pm 0.04$ | $39.2 \pm 1.8$ | $\underline{0.05 \pm 0.05}$ | $0.04 \pm 0.04$ |
| CORL (Wang et al., 2021) | $8.4 \pm 1.8$ | $0.19 \pm 0.04$ | $0.90 \pm 0.03$ | $9.6 \pm 1.7$ | $0.10 \pm 0.04$ | $0.82 \pm 0.04$ |
| RCL-OG (Yang et al., 2023a) | $\underline{7.0 \pm 1.4}$ | $0.16 \pm 0.03$ | $\underline{0.94 \pm 0.02}$ | $\underline{9.2 \pm 1.3}$ | $0.12 \pm 0.04$ | $\underline{0.84 \pm 0.02}$ |
| **ALIAS** (Ours) | $\mathbf{0.8 \pm 0.4}$ | $\mathbf{0.01 \pm 0.01}$ | $\mathbf{0.99 \pm 0.00}$ | $\mathbf{4.6 \pm 0.9}$ | $\mathbf{0.01 \pm 0.01}$ | $\mathbf{0.89 \pm 0.02}$ |

**Model misspecification.** Next, we consider the model misspecification scenarios when the Gaussian noise assumption is violated. In Table 3, we benchmark all methods on linear data with four types of non-Gaussian noises. The results indicate that our method is still the most robust to noise mis-specification, with an SHD of less than one in all four cases, and is the lead performer in three out of four configurations. In addition, we further study the performance of our method under different model misspecification scenarios, including mismatched causal model, noisy data, and the presence of hidden confounders, in Appendix D.4.

## 6.3 Nonlinear Data with Gaussian Processes

In this section, to answer the question of whether our method can operate beyond the standard linear-Gaussian setting, we follow the evaluations in Zhu et al. (2020); Wang et al. (2021); Yang et al. (2023a) to sample each causal mechanism $f_i$ from a Gaussian process with an RBF kernel of unit bandwidth, and the noises follow normal distributions with different variances sampled uniformly. We also follow Wang et al. (2021); Yang et al. (2023a) to apply Gaussian process regression using the RBF kernel with learnable length scale and regularization $\alpha = 1$ to calculate the BIC with non-equal variances (Section 3.2) for RL-BIC, CORL, RCL-OG, and our **ALIAS** method. For NOTEARS, DAGMA, and COSMO, we use their nonlinear versions where Multiple-layer Perceptrons (MLP) are used to model nonlinear relationships.

The empirical results reported in Table 4 verify the effectiveness of our method even on nonlinear data. Our method outperforms all other baselines in all metrics, either with or without pruning. Most remarkably,

Table 5: **Causal discovery performance on real-world flow cytometry data (Sachs et al., 2005) with 11 nodes, 17 edges, and 853 samples.** Running time is compared among RL-based methods. The figures for RL-BIC are as originally reported. **Bold**: best performance, underline: second-best performance. Since the causal model is potentially non-additive, we also consider CIT-based pruning with KCIT (Zhang et al., 2011).

| Method | CAM Pruning | | | CIT Pruning | | |
|---|---|---|---|---|---|---|
| | Total edges | Correct (↑) edges | SHD (↓) | Total edges | Correct (↑) edges | SHD (↓) |
| NOTEARS (Zheng et al., 2020) | 8 | 5 | 13 | 7 | 5 | 13 |
| DAGMA (Bello et al., 2022) | 6 | 2 | 15 | 6 | 2 | 15 |
| NOCURL (Yu et al., 2021) | 4 | 2 | 15 | 4 | 2 | 15 |
| COSMO (Massidda et al., 2024) | 5 | 2 | 16 | 5 | 2 | 16 |
| RL-BIC (Zhu et al., 2020) | 10 | 7 | 11 | - | - | - |
| CORL (Wang et al., 2021) | 9 | 3 | 14 | 10 | 3 | 15 |
| RCL-OG (Yang et al., 2023a) | 9 | 5 | 13 | 9 | 5 | 13 |
| **ALIAS** (Ours) | 10 | **8** | **10** | 9 | **8** | **9** |

even without pruning, our method correctly identifies nearly every edge with an expected SHD lower than 1. However, by using CAM pruning, there is a slight degrade in performance of most methods, which could be due to CAM's inability to capture complex causal mechanisms drawn from Gaussian processes.

Furthermore, following Lachapelle et al. (2020), we also study the case of causal model misspecification with Post-nonlinear models (Zhang & Hyvärinen, 2009) in Appendix D.4. In addition, nonlinear models generated using MLPs are also studied in Appendix D.5.

### 6.4 Real Data

Next, to confirm the validity of our method past synthetic data, we evaluate it on the popular benchmark flow cytometry dataset (Sachs et al., 2005), which involves a protein signaling network based on expression levels of proteins and phospholipids. We employ the observational partition of the dataset with 853 samples, 11 nodes, and 17 edges.

The empirical results provided in Table 5 show that our method **ALIAS** both achieves the best SHD and number of correct edges among all approaches. Specifically, using CAM pruning under the assumption of generalized additive noise models, we achieve the lowest SHD of 10 compared with the second-best of 11 by RL-BIC. Meanwhile, when using CIT-based pruning, we can even further reduce the SHD to 9, with 8 out of 9 identified edges are correct. This is a state-of-the-art level of SHD among existing studies on this dataset.

### 6.5 Runtime Analysis

Here, we analyze the efficiency of **ALIAS** in details. First, to show the significance of our optimal quadratic complexity for sampling DAGs, our Figure 4a compares the runtime of our policy with the autoregressive sampling approaches in CORL and RCL-OG with cubic complexity. It can be seen that, our DAG sampling policy is much faster than other approaches and does not significantly slow down with increasing graph sizes. Meanwhile, CORL and RCL-OG are nearly 140 times slower than **ALIAS** at 50 nodes, and the speedup ratio drastically increases with the growth of the graph. Second, in Figure 4b, we detail the runtime and performance of all methods with varying numbers of nodes. For small- to moderate-sized graphs of up to 50 nodes, our method is even faster than gradient-based methods NOCURL and COSMO. For larger graphs, RL-BIC and CORL become computationally expensive very rapidly, and while other methods become faster than **ALIAS**, their performance quickly degrade with significantly larger SHDs than our method.

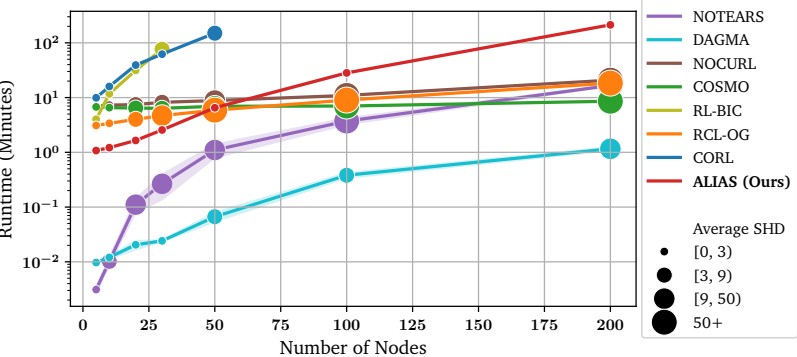

| Nodes | ALIAS | CORL | RCL-OG |
|---|---|---|---|
| 50 | 0.3 | 45.7 ( 135.7×) | 46.4 ( 137.7×) |
| 100 | 0.2 | 146.8 ( 900.0×) | 188.6 (1,155.9×) |
| 150 | 0.3 | 359.6 (1,261.5×) | 471.3 (1,653.2×) |
| 200 | 0.4 | 640.6 (1,596.8×) | 915.5 (2,282.0×) |

**(a) Average sampling time for each DAG in milliseconds.**

**(b) Causal Discovery Runtime.**

Figure 4: **Runtime analysis of ALIAS.** (a) We demonstrate the efficiency of our $\mathcal{O}\left(d^2\right)$ sampling technique compared with the $\mathcal{O}\left(d^3\right)$ approaches in CORL and RCL-OG. (b) We study the runtime of all methods with respect to graph size. The shaded areas depicts standard errors over 5 random linear-Gaussian datasets with the regular weight range $\mathcal{U}\left([-2, -0.5] \cup [0.5, 2]\right)$ on ER-2 graphs.

## 6.6 Ablation Studies

In Figure 1, we study the effect of the choice of graph scorer, RL method, and number of training steps onto the performance of **ALIAS** compared with DAGMA and CORL as the representatives for continuous optimization and RL-based approaches. It can be seen that all variants of our method surpass the baselines, using as few as 1 000 training steps. While all variants perform equivalently well, PPO proves to be a better choice than A2C, with both variants PPO + BIC and PPO + LS can reach very close to zero SHD, while those of A2C are not as performant (Figure 1b). The influence of other hyper-parameters can be found in Appendix D.3.

Furthermore, to show that the effectiveness of **ALIAS** is not only thanks to the **Vec2DAG** parametrization alone, but also the application of RL, as opposed to the gradient-based optimization approach commonly employed in the literature, we replace RL in our method with a gradient-based optimizer, which is popular among modern causal discovery methods, and compare the performances. However, since Vec2DAG is discrete, which renders the objective non-differentiable as is, we make slight modifications to make it amenable for continuous optimization, and the adapted version for linear data is given as:

$$\text{Vec2DAG}^{\text{cont.}} := \left(\mathbf{E}(\mathbf{z}) + \mathbf{E}(\mathbf{z})^\top\right) \odot H(\text{grad}(\mathbf{p}(\mathbf{z}))),$$

which still represents the weighted adjacency matrix of a DAG. Still, $H(\cdot)$ is not differentiable, so we further use the Straight-Through estimator (Bengio et al., 2013) to estimate its gradients. Specifically, we use $H(\text{grad}(\mathbf{p}(\mathbf{z})))$ for the forward pass whereas the gradients of the inner part $\frac{\partial \text{grad}(\mathbf{p}(\mathbf{z}))}{\partial \mathbf{z}}$ is used for the backward pass. This is not done similarly for the first term in **Vec2DAG** because that would require an additional weight matrix to represent linear coefficients, which is redundant compared with the above. Then, since the BIC score used in our RL approach is also non-differentiable, we adopt a likelihood-based loss similar to BIC as follows (which is also used in, e.g., GOLEM, Ng et al., 2020):

$$\mathcal{L}(\mathbf{z}) = \ln\left(\frac{1}{n \times d}\|\mathbf{X} - \mathbf{X} \cdot \text{Vec2DAG}^{\text{cont.}}(\mathbf{z})\|_2^2\right) + \lambda_1|\mathbf{z}|,$$

where $\lambda_1$ is the sparsity regularization coefficient. We minimize this loss until convergence using the Adam optimizer (Kingma, 2014) (same as NOCURL, COSMO, and the RL algorithm PPO in our method).

Table 6: **Role of RL in ALIAS.** We replace RL with continuous optimization using Adam (Kingma, 2014) and compare with the RL version. The numbers are *mean ± standard error* over 5 random datasets on 10-node ER-2 graphs.

| Method | SHD (↓) | FDR (↓) | TPR (↑) |
|---|---|---|---|
| Vec2DAG + continuous optimization ($\mathrm{lr} = 10^{-2}, \lambda_1 = 10^{-3}$) | $14.0 \pm 0.6$ | $0.52 \pm 0.05$ | $0.42 \pm 0.06$ |
| Vec2DAG + continuous optimization ($\mathrm{lr} = 10^{-2}, \lambda_1 = 10^{-5}$) | $9.6 \pm 2.6$ | $0.36 \pm 0.06$ | $0.65 \pm 0.08$ |
| Vec2DAG + continuous optimization ($\mathrm{lr} = 10^{-2}, \lambda_1 = 10^{-7}$) | $11.8 \pm 1.6$ | $0.45 \pm 0.08$ | $0.48 \pm 0.06$ |
| Vec2DAG + continuous optimization ($\mathrm{lr} = 10^{-3}, \lambda_1 = 10^{-3}$) | $10.0 \pm 1.7$ | $0.37 \pm 0.05$ | $0.56 \pm 0.06$ |
| Vec2DAG + continuous optimization ($\mathrm{lr} = 10^{-3}, \lambda_1 = 10^{-5}$) | $11.8 \pm 2.4$ | $0.44 \pm 0.06$ | $0.55 \pm 0.08$ |
| Vec2DAG + continuous optimization ($\mathrm{lr} = 10^{-3}, \lambda_1 = 10^{-7}$) | $8.6 \pm 0.4$ | $0.32 \pm 0.03$ | $0.62 \pm 0.01$ |
| Vec2DAG + continuous optimization ($\mathrm{lr} = 10^{-4}, \lambda_1 = 10^{-3}$) | $14.2 \pm 1.9$ | $0.53 \pm 0.06$ | $0.49 \pm 0.05$ |
| Vec2DAG + continuous optimization ($\mathrm{lr} = 10^{-4}, \lambda_1 = 10^{-5}$) | $12.0 \pm 2.1$ | $0.45 \pm 0.05$ | $0.59 \pm 0.04$ |
| Vec2DAG + continuous optimization ($\mathrm{lr} = 10^{-4}, \lambda_1 = 10^{-7}$) | $12.8 \pm 2.8$ | $0.45 \pm 0.08$ | $0.55 \pm 0.04$ |
| Vec2DAG + continuous optimization ($\mathrm{lr} = 10^{-5}, \lambda_1 = 10^{-3}$) | $12.6 \pm 3.7$ | $0.49 \pm 0.10$ | $0.55 \pm 0.12$ |
| Vec2DAG + continuous optimization ($\mathrm{lr} = 10^{-5}, \lambda_1 = 10^{-5}$) | $13.0 \pm 3.2$ | $0.51 \pm 0.09$ | $0.51 \pm 0.10$ |
| Vec2DAG + continuous optimization ($\mathrm{lr} = 10^{-5}, \lambda_1 = 10^{-7}$) | $12.2 \pm 2.5$ | $0.49 \pm 0.08$ | $0.51 \pm 0.08$ |
| **Vec2DAG + RL** | $\mathbf{0.0 \pm 0.0}$ | $\mathbf{0.00 \pm 0.00}$ | $\mathbf{1.00 \pm 0.00}$ |

We provide the results in Table 6 with a wide range of hyperparameter choices for the above approach. It can be seen that even in this simple case, the continuous optimization approach performs poorly and cannot compete with our RL approach, confirming that **ALIAS**'s effectiveness is attributed greatly by RL, not just **Vec2DAG** alone.

## 7  Conclusions

In this study, a novel causal discovery method based on RL is proposed. With the introduction of a new DAG characterization that bridges an unconstrained continuous space to the constrained DAG space, we devise an RL policy that can generate DAGs efficiently without any enforcement of the acyclicity constraint, which helps improve the search for the optimal score drastically. Experiments on a wide array of both synthetic and real datasets confirm the effectiveness of our method compared with state-of-the-art baselines.

Regarding limitations, the RL approaches in our study, which are online RL methods, may be limited in sample efficiency, as exploration data is not effectively ultilized to prioritize visiting promising DAGs, thus potentially requiring more explorations than needed to reach the optimal DAG. Towards this end, more sample-efficient RL approaches, such as Optimistic PPO (Cai et al., 2020), or reward redesign can be considered to enhance exploration in our method, and thus further improve its efficiency.

Future work may involve deepening the understanding on the convergence properties of our method and extending it to more intriguing settings like causal discovery with interventional data and hidden variables.

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

```
1  def Vec2DAG(z) -> np.ndarray:
2      p = z[:d]                    # R^d
3      E = np.zeros((d, d))
4      E[np.triu_indices(d, -1)] = z[d:]    # R^(d(d-1)/2)
5
6      A = (E + E.T > 0) * (p[:, None] < p[None, :])
7      return A
```

Figure 5: Unconstrained DAG parameterization. This function takes as input a real-valued vector $\mathbf{z} \in \mathbb{R}^{d \cdot (d+1)/2}$ and deterministically transforms it into an adjacency matrix of a $d$-node DAG.

## Appendix

## A  Details about BIC scores

### A.1  Non-equal variances BIC

Recall that the additive noise model under Gaussian noise is given by $X_i := f_i\left(\mathbf{X}_{\mathrm{pa}_i}\right) + E_i$, where $E_i \sim \mathcal{N}\left(0, \sigma_i^2\right)$. This implies $X_i \sim \mathcal{N}\left(f_i\left(\mathbf{X}_{\mathrm{pa}_i}\right), \sigma_i^2\right)$ and the log-likelihood of an empirical dataset $\mathcal{D} = \left\{\mathbf{x}^{(k)}\right\}_{k=1}^{n}$ is given by

$$\mathcal{L} = \ln p\left(\mathcal{D} \mid f, \boldsymbol{\sigma}, \mathcal{G}\right) \tag{7}$$

$$= -\frac{1}{2} \sum_{k=1}^{n} \sum_{i=1}^{d} \frac{\left(x_i^{(k)} - f_i\left(x_{\mathrm{pa}_i}^{(k)}\right)\right)^2}{\sigma_i^2} \tag{8}$$

$$- \frac{n}{2} \sum_{i=1}^{d} \ln \sigma_i^2 + \mathrm{const}, \tag{9}$$

where the constant does not depend on any variable.

The maximum likelihood estimator for $f_i$ can be found via least square methods, and that of $\sigma_i^2$ can be found by solving $\frac{\partial \mathcal{L}}{\partial \sigma_i^2} = 0$, which yields

$$\hat{\sigma}_i^2 = \frac{1}{n} \underbrace{\sum_{k=1}^{n} \left(x_i^{(k)} - \hat{f}_i\left(x_{\mathrm{pa}_i}^{(k)}\right)\right)^2}_{\mathrm{SSR}_i}. \tag{10}$$

Plugging this back to Eqn. (7) gives

$$\hat{\mathcal{L}} = -\frac{n}{2} \sum_{i=1}^{d} \ln \frac{\mathrm{SSR}_i}{n} + \mathrm{const}. \tag{11}$$

Finally, we obtain the BIC score for the non-equal variances case by incorporating this into Eqn. (3):

$$\mathrm{BIC}\left(\mathcal{D}, \mathcal{G}\right) = n \sum_{i=1}^{d} \ln \frac{\mathrm{SSR}_i}{n} + |\mathcal{G}| \ln n + \mathrm{const}, \tag{12}$$

## A.2 Equal variances BIC

Similarly to the unequal variances case, by assuming $\sigma_1 = \ldots = \sigma_d = \sigma$, we solve for $\frac{\partial \mathcal{L}}{\partial \sigma^2} = 0$ and obtain

$$\hat{\sigma}^2 = \frac{1}{nd} \sum_{i=1}^{d} \underbrace{\sum_{k=1}^{n} \left( x_i^{(k)} - \hat{f}_i \left( x_{\mathrm{pa}_i}^{(k)} \right) \right)^2}_{\mathrm{SSR}_i}. \tag{13}$$

Substituting this estimate into Eqn. (7) yields us with

$$\hat{\mathcal{L}} = -\frac{nd}{2} \ln \frac{\sum_{i=1}^{d} \mathrm{SSR}_i}{nd}. \tag{14}$$

For the last step, the BIC score for the equal variance case is given by substitution as

$$\mathrm{BIC} \left( \mathcal{D}, \mathcal{G} \right) = nd \ln \frac{\sum_{i=1}^{d} \mathrm{SSR}_i}{nd} + |\mathcal{G}| \ln n + \mathrm{const.} \tag{15}$$

# B   Proofs

## B.1   Proof of Lemma 1

**Lemma.** *Let $\mathcal{G}^*$ be the ground truth DAG of an identifiable SCM satisfying causal minimality (Peters et al., 2014) (i.e., there are no redundant edges) inducing the dataset $\mathcal{D}$, and let $n$ be the sample size of $\mathcal{D}$. Then, in the limit of large $n$, $\mathcal{S}_{BIC}(\mathcal{D}, \mathcal{G}^*) > \mathcal{S}_{BIC}(\mathcal{D}, \mathcal{G})$ for any $\mathcal{G} \neq \mathcal{G}^*$.*

*Proof.* Let $\psi^*$ be the parameter of the causal model generating the dataset $\mathcal{D}$. Because the causal model is identifiable and $(\psi^*, \mathcal{G}^*)$ are the true parameters generating the data $\mathcal{D}$, the likelihood $p(\mathcal{D} \mid \psi^*, \mathcal{G}^*)$ is the highest possible. For any incorrect DAG $\mathcal{G} \neq \mathcal{G}^*$, there only exists parameter $\psi$ such that $p(\mathcal{D} \mid \psi^*, \mathcal{G}^*) = p(\mathcal{D} \mid \psi, \mathcal{G})$ if $\mathcal{G}^* \subset \mathcal{G}$, because of causal minimality. Otherwise, the difference between the likelihoods are given by

$$\ln p\left(\mathcal{D} \mid \psi^*, \mathcal{G}^*\right) - \ln p\left(\mathcal{D} \mid \psi, \mathcal{G}\right) = \sum_{k=1}^{n} \ln p\left(\mathbf{x}^{(k)} \mid \psi^*, \mathcal{G}^*\right) - \ln p\left(\mathbf{x}^{(k)} \mid \psi, \mathcal{G}\right) \tag{16}$$

$$= n \cdot \underbrace{\mathrm{KL}\left(p\left(\mathcal{D} \mid \psi^*, \mathcal{G}^*\right) \parallel p\left(\mathcal{D} \mid \psi, \mathcal{G}\right)\right)}_{k(\mathcal{G}^*, \mathcal{G})} + o(n) \tag{17}$$

where the second equality follows from the asymptotic behavior of the log-likelihoods. Therefore, for any incorrect DAG $\mathcal{G} \not\supseteq \mathcal{G}^*$:

$$\mathcal{S}_{\mathrm{BIC}}\left(\mathcal{D}, \mathcal{G}^*\right) - \mathcal{S}_{\mathrm{BIC}}\left(\mathcal{D}, \mathcal{G}\right) = n \cdot k\left(\mathcal{G}^*, \mathcal{G}\right) + o(n) + \left(|\mathcal{G}| - |\mathcal{G}^*|\right) \ln n. \tag{18}$$

As $n \to \infty$, if $\mathcal{G}^* \nsubseteq \mathcal{G}$ then the KL divergence term grows linearly with $n$, dominating the logarithmic growth of the penalty term. On the other hand, if $\mathcal{G}^* \subset \mathcal{G}$ then the difference between the likelihoods vanishes and is therefore dominated by the penalty term. Thus, for any $\mathcal{G} \neq \mathcal{G}^*$, the BIC score satisfies $\mathcal{S}_{\mathrm{BIC}}(\mathcal{D}, \mathcal{G}^*) > \mathcal{S}_{\mathrm{BIC}}(\mathcal{D}, \mathcal{G})$ as $n \to \infty$. $\qquad\qquad\square$

## B.2   Proof of Theorem 1

**Theorem.** *For all $d \in \mathbb{N}^+$, let $\mathbf{Vec2DAG}_d : \mathbb{S}_d \to \{0, 1\}^{d \times d}$ be defined as in Eq. 5. Then, $\mathrm{Im}\left(\mathbf{Vec2DAG}_d\right) = \mathbb{D}_d$.*

To show that **Vec2DAG** is surjective, we first show that every point in the parameter space maps to a directed graph without any cycle, and vice-versa, for any DAG, there always exists a vector that maps to it.

**Lemma.** *For all $d \in \mathbb{N}^+$, let $\mathbf{Vec2DAG}_d : \mathbb{S}_d \to \{0,1\}^{d \times d}$ be defined as in Eq. (5). Then, $\mathbf{Vec2DAG}_d(\mathbf{z}) \in \mathbb{D}_d \ \forall \mathbf{z} \in \mathbb{S}_d$.*

*Proof.* The proof follows the same argument with Theorem 2.1 of Yu et al. (2021). Specifically, let $\mathbf{A} = \mathbf{Vec2DAG}_d(\mathbf{z}) = H\left(\mathbf{E}(\mathbf{z}) + \mathbf{E}(\mathbf{z})^\top\right) \odot H\left(\mathrm{grad}\left(\mathbf{p}(\mathbf{z})\right)\right)$. Then, the necessary condition for an edge from $i$ to $j$ to exist is $p_i < p_j$. By contradiction, assuming there exists a cycle $(i_1 \to \ldots \to i_k \to i_1)$ in $\mathbf{A}$, then it follows that $p_{i_1} < \cdots < p_{i_k} < p_{i_1}$. This contradicts with the total-ordering property of real values, thus concluding our argument. $\qquad\square$

**Lemma.** *For all $d \in \mathbb{N}^+$, let $\mathbf{Vec2DAG}_d : \mathbb{S}_d \to \{0,1\}^{d \times d}$ be defined as in Eq. (5). Then, for any DAG $\mathbf{A} \in \mathbb{D}_d$ and $\epsilon > 0$, there exists $\mathbf{z} \in (-\epsilon, \epsilon)^{d \cdot (d+1)/2} \subset \mathbb{S}_d$ such that $\mathbf{Vec2DAG}_d(\mathbf{z}) = \mathbf{A}$.*

*Proof.* We prove by construction. Let $\mathrm{an}_i = \{j \neq i \mid \text{there is a path from } j \text{ to } i\}$ be the set of ancestors of $i$. It follows that if $i \to j$ is in $\mathbf{A}$ then every ancestor of $i$ is an ancestor of $j$, so $\mathrm{an}_i \subset \mathrm{an}_j$ and $\mathrm{an}_j = \mathrm{an}_i \cup \{i\} \cup (\mathrm{an}_j \setminus \mathrm{an}_i \setminus \{i\})$, which means $|\mathrm{an}_i| < |\mathrm{an}_j|$. Therefore, we can construct the unnormalized node potentials as $\tilde{p}_i = |\mathrm{an}_i|$ and normalize it to fit into the $(-\epsilon, \epsilon)$ range as

$$p_i := \frac{\tilde{p}_i - \min \tilde{p}_i}{\max \tilde{p}_i - \min \tilde{p}_i} \times \epsilon - \frac{1}{2}\epsilon. \tag{19}$$

We then construct the upper-triangular edge potentials matrix as

$$E_{ij} = \begin{cases} 1/2\epsilon & \text{if } i < j \text{ and } A_{ij} + A_{ji} = 1, \\ -1/2\epsilon & \text{otherwise.} \end{cases} \tag{20}$$

We now verify that this pair of potentials leads to the exact binary matrix $\mathbf{A}$ via Eq. (5). First, $E_{ij} + E_{ji} = 1/2\epsilon > 0$ if $i$ and $j$ are directly connected in $\mathbf{A}$, and $E_{ij} + E_{ji} = -1/2\epsilon < 0$ otherwise, thus $H\left(\mathbf{E} + \mathbf{E}^\top\right)$ is the undirected version of $\mathbf{A}$. Second, for any directed edge $j \to i$ in $\mathbf{A}$, we have $|\mathrm{an}_i| < |\mathrm{an}_j|$, which leads to $\mathrm{grad}(\mathbf{p})_{ij} = p_j - p_i > 0$, and thus $H(\mathrm{grad}(\mathbf{p}))$ encodes the direction for every directed edge in $\mathbf{A}$. Therefore, using the Hadamard product, we mask out the edges in $H(\mathrm{grad}(\mathbf{p}))$ that do not exist in $\mathbf{A}$, and at the same time give direction to any available edge in $H\left(\mathbf{E} + \mathbf{E}^\top\right)$, resulting in exactly $\mathbf{A}$. $\qquad\square$

Lemmas B.2 and B.2 then completes our proof of Theorem 1.

### B.3 Proof of Lemma 2

**Lemma.** *(Scaling and Translation Invariance). For all $d \in \mathbb{N}^+$, let $\mathbf{Vec2DAG}_d : \mathbb{S}_d \to \{0,1\}^{d \times d}$ be defined as in Eq. 5. Then, for all $\mathbf{z} \in \mathbb{S}_d$, $\alpha > 0$, and $\boldsymbol{\beta} \in \mathbb{S}_d$ such that $|\mathbf{p}(\boldsymbol{\beta})_i| < 1/2 \min_j \left|\mathbf{p}(\mathbf{z})_i - \mathbf{p}(\mathbf{z})_j\right|$ and $\left|\mathbf{E}(\boldsymbol{\beta})_{ij}\right| < \left|\mathbf{E}(\mathbf{z})_{ij}\right| \ \forall i,j$, we have $\mathbf{Vec2DAG}_d(\mathbf{z}) = \mathbf{Vec2DAG}_d(\alpha \cdot (\mathbf{z} + \boldsymbol{\beta}))$.*

*Proof.* We first show that $\mathbf{Vec2DAG}_d(\mathbf{z}) = \mathbf{Vec2DAG}_d(\alpha \cdot \mathbf{z}) \ \forall \alpha > 0$. Let $\mathbf{A} = \mathbf{Vec2DAG}_d(\mathbf{z})$. This follows from the fact that

$$\mathbf{Vec2DAG}_d(\mathbf{z})_{ij} = 1 \Leftrightarrow \begin{cases} \mathbf{p}(\mathbf{z})_i < \mathbf{p}(\mathbf{z})_j \\ \mathbf{E}(\mathbf{z})_{ij} + \mathbf{E}(\mathbf{z})_{ji} > 0 \end{cases} \tag{21}$$

$$\Leftrightarrow \begin{cases} \alpha \mathbf{p}(\mathbf{z})_i < \alpha \mathbf{p}(\mathbf{z})_j \\ \alpha \mathbf{E}(\mathbf{z})_{ij} + \alpha \mathbf{E}(\mathbf{z})_{ji} > 0 \end{cases} \tag{22}$$

$$\Leftrightarrow \mathbf{Vec2DAG}_d(\alpha \cdot \mathbf{z})_{ij} = 1, \tag{23}$$

or equivalently, $\mathbf{Vec2DAG}_d(\mathbf{z}) = \mathbf{Vec2DAG}_d(\alpha \cdot \mathbf{z})$.

Next, we prove $\mathbf{Vec2DAG}_d(\mathbf{z}) = \mathbf{Vec2DAG}_d(\mathbf{z} + \boldsymbol{\beta})$ if $|\mathbf{p}(\boldsymbol{\beta})_i| < {}^1\!/_2 \min_j \left|\mathbf{p}(\mathbf{z})_i - \mathbf{p}(\mathbf{z})_j\right|$ and $\left|\mathbf{E}(\boldsymbol{\beta})_{ij}\right| < \left|\mathbf{E}(\mathbf{z})_{ij}\right| \forall i, j$. These conditions state that translating the potentials by an amount $\boldsymbol{\beta}$ that does not change the ordering of $\mathbf{p}$ or the element-wise positivity of $\mathbf{E}$ also leads to the same DAG. First, it can be seen that if $p_i < p_j$ and $p'_i, p'_j \in (-{}^1\!/_2(p_j - p_i), {}^1\!/_2(p_j - p_i))$, then $p_i + p'_i < {}^1\!/_2(p_i + p_j) < p_j + p'_j$. Therefore, $\mathbf{p}(\mathbf{z})_i < \mathbf{p}(\mathbf{z})_j \Leftrightarrow \mathbf{p}(\mathbf{z})_i + \mathbf{p}(\boldsymbol{\beta})_i < \mathbf{p}(\mathbf{z})_j + \mathbf{p}(\boldsymbol{\beta})_j$ if $|\mathbf{p}(\boldsymbol{\beta})_i| < {}^1\!/_2 \left|\mathbf{p}(\mathbf{z})_i - \mathbf{p}(\mathbf{z})_j\right|$, which also applies when $|\mathbf{p}(\boldsymbol{\beta})_i| < {}^1\!/_2 \min_j \left|\mathbf{p}(\mathbf{z})_i - \mathbf{p}(\mathbf{z})_j\right|$.

Similarly, $\mathbf{E}(\mathbf{z})_{ij} > 0 \Leftrightarrow \mathbf{E}(\mathbf{z})_{ij} + \mathbf{E}(\boldsymbol{\beta})_{ij} > 0$ if $\left|\mathbf{E}(\boldsymbol{\beta})_{ij}\right| < \left|\mathbf{E}(\mathbf{z})_{ij}\right|$. Combining these two points entails that if $\left|\mathbf{E}(\boldsymbol{\beta})_{ij}\right| < \left|\mathbf{E}(\mathbf{z})_{ij}\right| \forall i, j$ then we obtain

$$\mathbf{Vec2DAG}_d(\mathbf{z})_{ij} = 1 \Leftrightarrow \begin{cases} \mathbf{p}(\mathbf{z})_i < \mathbf{p}(\mathbf{z})_j \\ \mathbf{E}(\mathbf{z})_{ij} + \mathbf{E}(\mathbf{z})_{ji} > 0 \end{cases} \tag{24}$$

$$\Leftrightarrow \begin{cases} \mathbf{p}(\mathbf{z})_i + \mathbf{p}(\boldsymbol{\beta})_i < \mathbf{p}(\mathbf{z})_j + \mathbf{p}(\boldsymbol{\beta})_j \\ \left(\mathbf{E}(\mathbf{z})_{ij} + \mathbf{E}(\boldsymbol{\beta})_{ij}\right) + \left(\mathbf{E}(\mathbf{z})_{ji} + \mathbf{E}(\boldsymbol{\beta})_{ji}\right) > 0 \end{cases} \tag{25}$$

$$\Leftrightarrow \mathbf{Vec2DAG}_d(\mathbf{z} + \boldsymbol{\beta})_{ij} = 1. \tag{26}$$

Finally, combining Eqs. (23) and (26) concludes our proof for Lemma 2:

$$\mathbf{Vec2DAG}_d(\alpha \cdot (\mathbf{z} + \boldsymbol{\beta})) = \mathbf{Vec2DAG}_d(\mathbf{z} + \boldsymbol{\beta}) = \mathbf{Vec2DAG}_d(\mathbf{z}).$$

$\square$

### B.4 Proof of Lemma 3

**Lemma.** *(Proximity between DAGs). Let $\mathbf{z} \in \mathbb{S}_d$. Then, for any DAG $\mathbf{A} \in \mathbb{D}_d$ and $\epsilon > 0$, there exists $\mathbf{z_A}$ in the unit ball $B(\infty; \|\mathbf{z}\|_\infty + \epsilon)$ around $\mathbf{z}$ such that $\mathbf{Vec2DAG}_d(\mathbf{z_A}) = \mathbf{A}$.*

*Proof.* Lemma B.2 established that there exists a representation $\mathbf{z_A} \in (-\epsilon, \epsilon)^{d \cdot (d+1)/2}$ for any DAG $\mathbf{A}$ and $\epsilon > 0$. Therefore, the distance between $\mathbf{z}$ and $\mathbf{z_A}$ is bounded in $l_\infty$ norm by the triangle inequality as $|\mathbf{z} - \mathbf{z_A}|_\infty \leq |\mathbf{z}|_\infty + |\mathbf{z_A}| < |\mathbf{z}|_\infty + \epsilon$. $\square$

### B.5 Proof of Lemma 4

**Lemma.** *Assuming causal identifiability and causal minimality, that is, there is a unique causal model with no redundant edges that can produce the observed dataset, and BIC score is used to define the reward $\mathcal{R}(\mathbf{z})$ as in Eq. (6). Let $n$ be the sample size of the observed dataset $\mathcal{D}$, $\theta^* \in \arg\max_{\theta \in \Theta} \mathbb{E}_{\mathbf{z} \sim \pi_\theta}[\mathcal{R}(\mathbf{z})]$, where $\pi_\theta(\mathbf{z}) = \mathcal{N}\left(\mathbf{z}; \boldsymbol{\mu_\theta}, diag(\boldsymbol{\sigma_\theta^2})\right)$. Then, as $n \to \infty$, $\mathcal{G} = \mathbf{Vec2DAG}(\mathbf{z})$ is the true DAG, where $\mathbf{z} \sim \pi_{\theta^*}$.*

*Proof.* Let $\mathcal{G}^*$ be the true DAG. By the identifiability and minimality of the causal model and Lemma 1, as $n \to \infty$, $\mathbf{z}^* \in \arg\max \mathcal{R}(\mathbf{z})$ if and only if $\mathbf{Vec2DAG}(\mathbf{z}^*) = \mathcal{G}^*$. This implies that the reward $\mathcal{R}(\mathbf{z})$ is only maximized when $\mathbf{z}$ corresponds to $\mathcal{G}^*$.

Now, consider the objective $\theta^* \in \arg\max_{\theta \in \Theta} \mathbb{E}_{\mathbf{z} \sim \pi_\theta}[\mathcal{R}(\mathbf{z})]$. One solution to this maximization problem is when $\pi_\theta$ places all of its probability mass on an arbitrary $\mathbf{z}^* \in \arg\max \mathcal{R}(\mathbf{z})$, i.e., $\pi_{\theta^*}(\mathbf{z}) = \delta(\mathbf{z} - \mathbf{z}^*)$, where $\delta(\cdot)$ is the Dirac delta function. This is achieved when $\boldsymbol{\mu_\theta} = \mathbf{z}^*$ and $\boldsymbol{\sigma}^2 = \mathbf{0}$. In this case, any sample $\mathbf{z} \sim \pi_{\theta^*}$ satisfies $\mathbf{z} = \mathbf{z}^*$.

On the other hand, if $\pi_{\boldsymbol{\theta}}$ assigns probability mass to multiple values of $\arg\max \mathcal{R}(\mathbf{z})$, then it is a regular Gaussian distribution and thus will also put non-zero masses on the values with smaller rewards, thus strictly reducing the value of the expected reward. Thus, $\pi_{\boldsymbol{\theta}}$ cannot achieve the maximum expected reward unless it is a Dirac delta distribution concentrated at $\mathbf{z}^*$.

Finally, since $\mathbf{Vec2DAG}(\mathbf{z}^*) = \mathcal{G}^*$, we have $\mathcal{G} = \mathbf{Vec2DAG}(\mathbf{z}) = \mathcal{G}^*$ with probability 1 when $\mathbf{z} \sim \pi_{\boldsymbol{\theta}^*}$. Therefore, as $n \to \infty$, $\mathcal{G} = \mathbf{Vec2DAG}(\mathbf{z})$ is the true DAG. $\qquad\square$

## C  Experiment Details

### C.1  Datasets

#### C.1.1  Synthetic Linear-Gaussian Data

To simulate data for a given number of nodes $d$, we first generate a DAG following the Erdős-Rényi graph model (Erdős & Rényi, 1960) with a random ordering and an expected in-degree of $k \in \mathbb{N}^+$, denoted by ER-$k$. Next, edge weights are randomly sampled from the uniform distribution $\mathcal{U}([-5, -2] \cup [2, 5])$, giving a weighted matrix $\mathbf{W} \in \mathbb{R}^{d \times d}$ where zero entries indicate no connections, then the noises are sampled from the standard Gaussian distribution $E_i \sim \mathcal{N}(0, 1)$.

Finally, we sample $n = 1,000$ observations for each dataset following the linear assignment $X_i := \sum_{j \in \mathrm{pa}_i} W_{ji} X_j + E_i$. While linear-Gaussian models are non-identifiable in general (Spirtes et al., 2000), the instances with equal noise variances adopted in this experiment are known to be fully identifiable (Peters et al., 2014). This data generation process is similar to multiple other works such as Zheng et al. (2018); Zhu et al. (2020); Wang et al. (2021) and is conducted using the well-established gCastle[2] utility for causality research (Zhang et al., 2021).

#### C.1.2  Nonlinear Data with Gaussian Processes

We evaluate the performance of the proposed method **ALIAS** with competitors on the exact 5 datasets used by Zhu et al. (2020) in their experiment, which are produced by Lachapelle et al. (2020) (`https://github.com/kurowasan/GraN-DAG`, MIT license). In addition, we also consider the first 5 datasets of the PNL-GP portion of their datasets for the model misspecification experiment.

### C.2  Experiment Details

#### C.2.1  Evaluation Metrics

The estimated graphs are assessed against ground truth DAGs on multiple evaluation metrics, including the commonly employed Structural Hamming Distance (SHD, lower is better), False Detection Rate (FDR, lower is better), and True Positive Rate (TPR, higher is better). SHD counts the minimal number of edge additions, removals, and reversals in order to turn one graph into another. FDR is the ratio of incorrectly estimated edges over all estimated edges, while TPR is the proportion of correctly identified edges over all true edges. We also utilize gCastle for the calculations of the aforementioned metrics.

#### C.2.2  Implementations of Methods

Our set of baseline methods contains a wide range of both well-established and recent approaches, including the constrained continuous optimization approaches with soft DAG constraints NOTEARS (Zheng et al., 2018; 2020) and DAGMA (Bello et al., 2022), unconstrained continuous optimization approaches NOCURL (Yu et al., 2021) and COSMO (Massidda et al., 2024), as well as three RL-based methods RL-BIC (Zhu et al., 2020), CORL (Wang et al., 2021), and RCL-OG (Yang et al., 2023a), where the implementations are publicly available. More specifically,

---

[2]`https://github.com/huawei-noah/trustworthyAI/tree/master/gcastle`, version 1.0.3, Apache-2.0 license.

- NOTEARS (Zheng et al., 2018; 2020) is a continuous optimization method that optimizes over the continuous space of weighted adjacency $\mathbf{W} \in \mathbb{R}^{d \times d}$ for linear models, with a soft constraint $\mathbf{W}$ is DAG $\Leftrightarrow h(\mathbf{W}) = e^{\mathbf{W} \circ \mathbf{W}} - d = 0$, which is solved using augmented Lagrangian methods. For nonlinear data, $\mathbf{W}$ is used to mask the input of the MLPs that model the nonlinear causal mechanisms.

- DAGMA (Bello et al., 2022) is an alternative proposal to NOTEARS, with the acyclicity constraint $\mathbf{W}$ is DAG $\Leftrightarrow h(\mathbf{W}) = \log \det(s\mathbf{I} - \mathbf{W} \circ \mathbf{W}) - d \log s = 0$ for all $s > 0$.

- NOCURL (Yu et al., 2021) is proposed as an unconstrained continuous optimization method that represents the weight matrix of a linear model by $\mathbf{A} = \mathbf{W} \odot \text{ReLU}(\text{grad}(\mathbf{p}))$, where $\mathbf{W} \in \mathbb{R}^{d \times d}$ and $\text{grad}(\mathbf{p})_{ij} := p_j - p_i$. Their optimization strategy is to first find a solution from an unconstrained method like NOTEARS, then refine it with this new unconstrained representation.

- COSMO (Massidda et al., 2024) improves upon NOCURL by parametrizing the adjacency matrix with $\mathbf{A} = \mathbf{W} \odot \text{sigmoid}(\text{grad}(\mathbf{p})/\tau)$, which can be shown to converge to a DAG when $\tau \to 0^+$. They optimize directly with this formulation instead of employing the two-stage approach like NOCURL.

- RL-BIC (Zhu et al., 2020) is the first RL-based method, which involves an actor-critic agent that learns to output high-reward graphs. The acyclicity is incorporated into the reward to penalize cyclic graphs.

- CORL (Wang et al., 2021) is the first RL method for ordering-based causal discovery, which revolves around an agent that learns to produce causal orderings with high rewards. The policy sequentially generates each element of the ordering, then the causal order is pruned to obtain a DAG.

- RCL-OG (Yang et al., 2023a) is an alternative RL approach for ordering-based causal discovery, which improves the state space design of CORL. It reduces the state space size from permutations with a factorial size $\mathcal{O}(d!)$ to only $\mathcal{O}(2^d)$.

For NOTEARS, RL-BIC, and CORL, we also adopt the implementations from the gCastle package. For RCL-OG, we employ the implementation provided by the authors at `https://www.sdu-idea.cn/codes.php?name=RCL-OG` (no license provided). Since RCL-OG is essentially a Bayesian method, we use the best-scoring DAG from its 1 000 posterior samples as the output for a fair comparison with other methods.

Default hyper-parameters for each implementation are used unless specifically indicated. The detailed configurations of all methods are provided in Appendix C.2.3.

Our proposed **ALIAS** method is implemented using the Stable-Baselines3[3] toolset (Raffin et al., 2021) with the Advantage Actor-Critic (A2C, Mnih et al., 2016) and Proximal Policy Optimization (PPO, Schulman et al., 2017) methods, and a custom DAG environment built on top of Gymnasium[4] (Towers et al., 2023). See Appendix C.2.3 the hyper-parameters for our methods.

Experiments are executed on a mix of several machines running Ubuntu 20.04/22.04 with the matching Python environments, including the following configurations:

- AMD EPYC 7742 CPU, 1TB of RAM, and 8 Nvidia A100 40GB GPUs.

- Intel Xeon Platinum 8452Y CPU, 1TB of RAM, 4 Nvidia H100 80GB GPUs.

- Intel Core i9 13900KF CPU, 128GB of RAM, 1 Nvidia 4070Ti Super 16GB GPU.

The first configuration is used for batched executions of all methods, while the second is for resource-intensive experiments like high-dimensional ones with RL-based baselines, and the last configuration is for prototyping experiments. We find that RL-based baselines heavily rely on GPU with much slower (hours of) runtime on CPU, even on 30-node graphs, while our method is less dependent on GPU and can even handle datasets of up to 100 nodes in 45 minutes purely on CPU (single process).

---

[3]`https://github.com/DLR-RM/stable-baselines3`, MIT license.

[4]`https://github.com/Farama-Foundation/Gymnasium`, MIT license.

Table 7: Default hyper-parameters for our method **ALIAS** throughout the experiments. Unmentioned hyper-parameters are left unchanged.

| Experiment | Linear data (Figures 1, 2 and 3, Tables 2, 3, 14, 15, 16, 17, and 18) | Nonlinear GP data (Table 4 and Table 19) | Real data (Table 5) |
|---|---|---|---|
| Batch size (No. parallel environments) | | 64 | |
| Training steps | | 20,000 | |
| No. steps to run for each update | | 1 | |
| Data standardization (dimension-wise) | No (variances should remain equal) | No (data std. is near unit) | Yes (data std. is large) |
| RL method | | PPO | |
| Advantage normalization | | Yes | |
| Learning rate | | Stable-Baseline3' default (0.0003 for PPO and 0.0007 for A2C) | |
| Regression method | Linear Regression | Gaussian Process Regression with RBF kernel | |
| Scoring method | BIC equal variances, assuming Gaussian noises | BIC non-equal variances, assuming Gaussian noises | |
| $l_0$ regularization for LS score (not used in comparative experiments) | 0.000001 | - | - |

Table 8: Default hyper-parameters for CORL (Wang et al., 2021) throughout the experiments. Unmentioned hyper-parameters are left unchanged.

| Experiment | Linear data (Figures 1, 2 and 3, Tables 2, 3, 14, 15, 16, 17, and 18) | Nonlinear GP data (Table 4 and Table 19) | Real data (Table 5) |
|---|---|---|---|
| Batch size | | 64 | |
| Training steps | | 10,000 | |
| Data standardization (dimension-wise) | No (variances should remain equal) | No (data std. is near unit) | Yes (data std. is large) |
| Regression method | Linear Regression | Gaussian Process Regression with RBF kernel | |
| Scoring method | BIC equal variances, assuming Gaussian noises | BIC non-equal variances, assuming Gaussian noises | |

### C.2.3 Hyper-parameters

We provide the hyper-parameters for each method and experiment scenario as follows:

- **ALIAS** (Ours): Table 7.

- NOTEARS and DAGMA: Table 13.

- NOCURL: Table 11.

- COSMO: Table 12.

- RL-BIC: Table 9.

- CORL: Table 8.

- RCL-OG: Table 10.

## D  Additional Results

### D.1  Results on ER graphs

In Table 14, we provide detailed numerical results for Figure 3, i.e., effect of dimensionality and sample size on linear-Gaussian data and ER-8 graphs.

Table 9: Default hyper-parameters for RL-BIC (Zhu et al., 2020) throughout the experiments. Unmentioned hyper-parameters are left unchanged.

| Experiment | Linear data (Figures 1, 2 and 3, Tables 2, 3, 14, 15, 16, 17, and 18) | Nonlinear GP data (Table 4 and Table 19) | Real data (Table 5) |
|---|---|---|---|
| Batch size | | 64 | |
| Training steps | | 20,000 | |
| Data standardization (dimension-wise) | No (variances should remain equal) | No (data std. is near unit) | Yes (data std. is large) |
| Regression method | Linear Regression | Gaussian Process Regression with RBF kernel | |
| Scoring method | BIC equal variances, assuming Gaussian noises | BIC non-equal variances, assuming Gaussian noises | |

Table 10: Default hyper-parameters for RCL-OG (Yang et al., 2023a) throughout the experiments. Unmentioned hyper-parameters are left unchanged.

| Experiment | Linear data (Figures 1, 2 and 3, Tables 2, 3, 14, 15, 16, 17, and 18) | Nonlinear GP data (Table 4 and Table 19) | Real data (Table 5) |
|---|---|---|---|
| Batch size | | 32 | |
| Training steps | | 40,000 | |
| Data standardization (dimension-wise) | No (variances should remain equal) | No (data std. is near unit) | Yes (data std. is large) |
| Regression method | Linear Regression | Gaussian Process Regression with RBF kernel | |
| Scoring method | BIC equal variances, assuming Gaussian noises | BIC non-equal variances, assuming Gaussian noises | |

Table 11: Default hyper-parameters for NOCURL (Yang et al., 2023a) throughout the experiments. The values are the best parameters yielding the lowest SHD over linear-Gaussian datasets with 30-node ER-4 graphs, found via the tuning process provided by COSMO's implementation. Unmentioned hyper-parameters are left unchanged.

| Experiment | Linear data (Figures 1, 2 and 3, Tables 2, 3, 14, 15, 16, 17, and 18) | Nonlinear GP data (Table 4 and Table 19) | Real data (Table 5) |
|---|---|---|---|
| Batch size | | 64 | |
| Inner iterations | | 5000 | |
| Data standardization (dimension-wise) | No (variances should remain equal) | No (data std. is near unit) | Yes (data std. is large) |
| Learning rate | | 0.0009747753554628831 | |
| Regularization strength | | 0.007478648909986116 | |

Table 12: Default hyper-parameters for COSMO (Massidda et al., 2024) throughout the experiments. The values are the best parameters yielding the lowest SHD over linear-Gaussian datasets with 30-node ER-4 graphs for linear data, and MLP datasets with 40-node ER-4 graphs for nonlinear data. They are found via the tuning process provided by COSMO's implementation. Unmentioned hyper-parameters are left unchanged.

| Experiment | Linear data (Figures 1, 2 and 3, Tables 2, 3, 14, 15, 16, 17, and 18) | Nonlinear GP data (Table 4 and Table 19) | Real data (Table 5) |
|---|---|---|---|
| Batch size | | 64 | |
| Max epochs | 5000 | 2000 | |
| Data standardization (dimension-wise) | No (variances should remain equal) | No (data std. is near unit) | Yes (data std. is large) |
| Learning rate | 0.004424703475697184 | 0.0011606444486776536 | |
| $l_1$ regularization strength | 0.0007589315865487066 | 0.000999704401738756 | |
| $l_2$ regularization strength | 0.029171977709975934 | 0.0011406751425196925 | |
| Priority regularization strength | 0.0007604972601205271 | 0.0014549388704106592 | |
| Temperature | 0.0008407426566089702 | 0.0007651953117707655 | |
| Shift | 0.005860546462049756 | 0.009324030532123762 | |
| MLP hidden units | 0 | 10 | |

Table 13: Default hyper-parameters for NOTEARS Zheng et al. (2018; 2020) and DAGMA Bello et al. (2022) throughout the experiments. Unmentioned hyper-parameters are left unchanged.

| Experiment | Linear data (Figures 1, 2 and 3, Tables 2, 3, 14, 15, 16, 17, and 18) | Nonlinear GP data (Table 4 and Table 19) | Real data (Table 5) |
|---|---|---|---|
| Data standardization (dimension-wise) | No (variances should remain equal) | No (data std. is near unit) | Yes (data std. is large) |
| SEM | Linear | MLP | |
| Loss type | $l_2$ | | |

Table 14: Causal discovery performance as function of dimensionalities and sample size on ER-8 graphs. The numbers are *mean ± standard error* over 5 random datasets.

| Nodes | Samples | Method | SHD ($\downarrow$) | FDR ($\downarrow$) | TPR ($\uparrow$) |
|---|---|---|---|---|---|
| 20 | 100 | DAGMA | $40.6 \pm 3.5$ | $0.09 \pm 0.01$ | $0.78 \pm 0.02$ |
| | | **ALIAS** | $24.8 \pm 4.6$ | $0.04 \pm 0.01$ | $0.86 \pm 0.02$ |
| | 500 | DAGMA | $36.8 \pm 3.8$ | $0.08 \pm 0.01$ | $0.8 \pm 0.02$ |
| | | **ALIAS** | $3.0 \pm 1.1$ | $0.01 \pm 0.00$ | $0.98 \pm 0.00$ |
| | 1000 | DAGMA | $35.6 \pm 3.6$ | $0.07 \pm 0.01$ | $0.8 \pm 0.02$ |
| | | **ALIAS** | $1.0 \pm 0.6$ | $0.0 \pm 0.00$ | $0.99 \pm 0.00$ |
| | 2000 | DAGMA | $31.8 \pm 3.6$ | $0.07 \pm 0.01$ | $0.82 \pm 0.03$ |
| | | **ALIAS** | $0.2 \pm 0.2$ | $0.0 \pm 0.0$ | $1.0 \pm 0.0$ |
| | 5000 | DAGMA | $31.2 \pm 4.2$ | $0.07 \pm 0.01$ | $0.83 \pm 0.03$ |
| | | **ALIAS** | $0.2 \pm 0.2$ | $0.0 \pm 0.0$ | $1.0 \pm 0.0$ |
| 30 | 100 | DAGMA | $87.0 \pm 10.6$ | $0.19 \pm 0.03$ | $0.78 \pm 0.02$ |
| | | **ALIAS** | $51.8 \pm 7.9$ | $0.1 \pm 0.02$ | $0.86 \pm 0.02$ |
| | 500 | DAGMA | $67.0 \pm 9.5$ | $0.13 \pm 0.02$ | $0.81 \pm 0.02$ |
| | | **ALIAS** | $3.8 \pm 0.6$ | $0.01 \pm 0.0$ | $0.99 \pm 0.0$ |
| | 1000 | DAGMA | $67.6 \pm 8.0$ | $0.14 \pm 0.02$ | $0.82 \pm 0.02$ |
| | | **ALIAS** | $0.2 \pm 0.2$ | $0.0 \pm 0.0$ | $1.0 \pm 0.0$ |
| | 2000 | DAGMA | $65.8 \pm 7.9$ | $0.13 \pm 0.02$ | $0.82 \pm 0.02$ |
| | | **ALIAS** | $0.0 \pm 0.0$ | $0.0 \pm 0.0$ | $1.0 \pm 0.0$ |
| | 5000 | DAGMA | $68.4 \pm 10.2$ | $0.14 \pm 0.02$ | $0.81 \pm 0.03$ |
| | | **ALIAS** | $0.0 \pm 0.0$ | $0.0 \pm 0.0$ | $1.0 \pm 0.0$ |
| 50 | 100 | DAGMA | $242.0 \pm 15.6$ | $0.34 \pm 0.02$ | $0.76 \pm 0.01$ |
| | | **ALIAS** | $94.2 \pm 10.1$ | $0.14 \pm 0.02$ | $0.9 \pm 0.01$ |
| | 500 | DAGMA | $213.2 \pm 20.8$ | $0.31 \pm 0.03$ | $0.78 \pm 0.01$ |
| | | **ALIAS** | $13.6 \pm 5.5$ | $0.02 \pm 0.01$ | $0.99 \pm 0.00$ |
| | 1000 | DAGMA | $215.2 \pm 26.6$ | $0.3 \pm 0.04$ | $0.77 \pm 0.02$ |
| | | **ALIAS** | $2.2 \pm 1.0$ | $0.0 \pm 0.0$ | $1.0 \pm 0.0$ |
| | 2000 | DAGMA | $213.2 \pm 25.8$ | $0.31 \pm 0.03$ | $0.78 \pm 0.02$ |
| | | **ALIAS** | $1.8 \pm 0.8$ | $0.0 \pm 0.0$ | $1.0 \pm 0.0$ |
| | 5000 | DAGMA | $210.2 \pm 32.4$ | $0.3 \pm 0.04$ | $0.78 \pm 0.02$ |
| | | **ALIAS** | $0.7 \pm 0.5$ | $0.0 \pm 0.0$ | $1.0 \pm 0.0$ |

## D.2 Results on SF graphs

Table 15 investigates the effect of dimensionality and sample size on linear-Gaussian data and Scale-Free (SF) graphs with 8 parents per node.

Table 15: Causal discovery performance as function of dimensionalities and sample size on SF-8 graphs. The numbers are *mean ± standard error* over 5 random datasets.

| Nodes | Samples | Method | SHD (↓) | FDR (↓) | TPR (↑) |
|---|---|---|---|---|---|
| 20 | 100 | DAGMA | $14.6 \pm 3.5$ | $0.07 \pm 0.03$ | $0.87 \pm 0.02$ |
| | | **ALIAS** | $16.0 \pm 2.8$ | $0.09 \pm 0.02$ | $0.89 \pm 0.03$ |
| | 500 | DAGMA | $6.8 \pm 1.4$ | $0.02 \pm 0.01$ | $0.93 \pm 0.01$ |
| | | **ALIAS** | $0.6 \pm 0.3$ | $0.01 \pm 0.00$ | $1.0 \pm 0.00$ |
| | 1000 | DAGMA | $7.0 \pm 1.7$ | $0.02 \pm 0.01$ | $0.93 \pm 0.02$ |
| | | **ALIAS** | $0.2 \pm 0.2$ | $0.0 \pm 0.00$ | $1.0 \pm 0.00$ |
| | 2000 | DAGMA | $9.0 \pm 3.0$ | $0.04 \pm 0.02$ | $0.91 \pm 0.03$ |
| | | **ALIAS** | $0.2 \pm 0.2$ | $0.0 \pm 0.00$ | $1.0 \pm 0.00$ |
| | 5000 | DAGMA | $6.6 \pm 2.5$ | $0.03 \pm 0.02$ | $0.94 \pm 0.02$ |
| | | **ALIAS** | $0.2 \pm 0.2$ | $0.0 \pm 0.00$ | $1.0 \pm 0.00$ |
| 30 | 100 | DAGMA | $47.2 \pm 7.1$ | $0.15 \pm 0.04$ | $0.84 \pm 0.01$ |
| | | **ALIAS** | $30.2 \pm 4.4$ | $0.1 \pm 0.01$ | $0.92 \pm 0.01$ |
| | 500 | DAGMA | $45.8 \pm 7.9$ | $0.16 \pm 0.03$ | $0.85 \pm 0.02$ |
| | | **ALIAS** | $0.2 \pm 0.2$ | $0.0 \pm 0.0$ | $1.0 \pm 0.0$ |
| | 1000 | DAGMA | $43.6 \pm 10.0$ | $0.14 \pm 0.04$ | $0.86 \pm 0.03$ |
| | | **ALIAS** | $0.2 \pm 0.2$ | $0.0 \pm 0.0$ | $1.0 \pm 0.0$ |
| | 2000 | DAGMA | $36.6 \pm 9.7$ | $0.12 \pm 0.04$ | $0.89 \pm 0.02$ |
| | | **ALIAS** | $0.0 \pm 0.0$ | $0.0 \pm 0.0$ | $1.0 \pm 0.0$ |
| | 5000 | DAGMA | $27.4 \pm 4.7$ | $0.09 \pm 0.03$ | $0.9 \pm 0.01$ |
| | | **ALIAS** | $0.0 \pm 0.0$ | $0.0 \pm 0.0$ | $1.0 \pm 0.0$ |
| 50 | 100 | DAGMA | $101.2 \pm 18.3$ | $0.18 \pm 0.04$ | $0.84 \pm 0.02$ |
| | | **ALIAS** | $54.2 \pm 7.9$ | $0.11 \pm 0.01$ | $0.92 \pm 0.01$ |
| | 500 | DAGMA | $68.4 \pm 10.6$ | $0.13 \pm 0.02$ | $0.89 \pm 0.01$ |
| | | **ALIAS** | $9.8 \pm 5.1$ | $0.03 \pm 0.01$ | $0.99 \pm 0.00$ |
| | 1000 | DAGMA | $71.8 \pm 12.6$ | $0.13 \pm 0.03$ | $0.89 \pm 0.01$ |
| | | **ALIAS** | $5.2 \pm 3.3$ | $0.01 \pm 0.01$ | $1.0 \pm 0.0$ |
| | 2000 | DAGMA | $65.2 \pm 9.8$ | $0.12 \pm 0.02$ | $0.9 \pm 0.02$ |
| | | **ALIAS** | $5.4 \pm 4.4$ | $0.02 \pm 0.01$ | $1.0 \pm 0.0$ |
| | 5000 | DAGMA | $75.0 \pm 25.1$ | $0.14 \pm 0.05$ | $0.89 \pm 0.04$ |
| | | **ALIAS** | $8.8 \pm 5.2$ | $0.02 \pm 0.01$ | $0.99 \pm 0.00$ |

## D.3 Ablation Studies

### D.3.1 Effect of RL method and learning rate

In Table 16, we conduct ablation studies on the effect of the choices of RL method and learning rate on the performance of our **ALIAS** method with the BIC score.

Table 16: Performance sensitivity of **ALIAS** with BIC score subjected to the variations of learning rate. We employ linear-Gaussian datasets with 30 nodes on ER-8 graphs and use CORL (Wang et al., 2021) and DAGMA (Bello et al., 2022) as reference. For each row, we compute the means and standard errors over 5 independent datasets. **Bold**: better performance than the baselines. Unless otherwise indicated, the remaining hyper-parameters are used according to Table 7 in the Appendix.

| Method | RL method | Learning rate | SHD ($\downarrow$) | FDR ($\downarrow$) | TPR ($\uparrow$) |
|---|---|---|---|---|---|
| CORL | - | - | $82.4 \pm 22.3$ | $0.23 \pm 0.05$ | $0.87 \pm 0.04$ |
| DAGMA | - | - | $67.6 \pm 8.0$ | $0.14 \pm 0.02$ | $0.82 \pm 0.02$ |
| **ALIAS** (Ours) | A2C | 0.00001 | $79.4 \pm 6.7$ | $0.22 \pm 0.02$ | $0.86 \pm 0.01$ |
| | | 0.00005 | $\mathbf{12.6 \pm 3.1}$ | $\mathbf{0.04 \pm 0.01}$ | $\mathbf{0.98 \pm 0.00}$ |
| | | 0.0001 | $\mathbf{1.2 \pm 0.4}$ | $\mathbf{0.00 \pm 0.00}$ | $\mathbf{1.00 \pm 0.00}$ |
| | | 0.0005 | $\mathbf{2.2 \pm 1.3}$ | $\mathbf{0.01 \pm 0.00}$ | $\mathbf{1.00 \pm 0.00}$ |
| | | 0.001 | $\mathbf{22.8 \pm 3.9}$ | $\mathbf{0.07 \pm 0.01}$ | $\mathbf{0.96 \pm 0.01}$ |
| | | 0.005 | $123.8 \pm 49.7$ | $0.34 \pm 0.16$ | $0.54 \pm 0.21$ |
| | | 0.01 | $223.4 \pm 5.6$ | $0.48 \pm 0.03$ | $0.13 \pm 0.04$ |
| | | 0.05 | $232.6 \pm 3.0$ | $0.59 \pm 0.03$ | $0.11 \pm 0.01$ |
| | PPO | 0.00001 | $77.8 \pm 7.5$ | $0.21 \pm 0.02$ | $0.86 \pm 0.01$ |
| | | 0.00005 | $\mathbf{9.0 \pm 2.9}$ | $\mathbf{0.03 \pm 0.01}$ | $\mathbf{0.99 \pm 0.00}$ |
| | | 0.0001 | $\mathbf{1.2 \pm 0.4}$ | $\mathbf{0.00 \pm 0.00}$ | $\mathbf{1.00 \pm 0.00}$ |
| | | 0.0005 | $\mathbf{0.4 \pm 0.3}$ | $\mathbf{0.00 \pm 0.00}$ | $\mathbf{1.00 \pm 0.00}$ |
| | | 0.001 | $\mathbf{4.0 \pm 2.0}$ | $\mathbf{0.01 \pm 0.01}$ | $\mathbf{0.99 \pm 0.00}$ |
| | | 0.005 | $204.8 \pm 38.8$ | $0.65 \pm 0.13$ | $0.21 \pm 0.18$ |
| | | 0.01 | $225.4 \pm 6.8$ | $0.53 \pm 0.04$ | $0.11 \pm 0.04$ |
| | | 0.05 | $231.4 \pm 5.1$ | $0.54 \pm 0.02$ | $0.13 \pm 0.01$ |

### D.3.2 Effect of Entropy regularization

In Table 17, we conduct ablation studies on the effect of the choices of entropy regularization on the performance of our **ALIAS** method with the BIC score.

Table 17: Performance sensitivity of **ALIAS** with BIC score subjected to the variations of Entropy regularization weight. We employ linear-Gaussian datasets with 30 nodes on ER-8 graphs and use CORL (Wang et al., 2021) and DAGMA (Bello et al., 2022) as reference. Our scoring function is BIC. For each row, we compute the means and standard errors over 5 independent datasets. **Bold**: better performance than the baselines. Unless otherwise indicated, the remaining hyper-parameters are used according to Table 7.

| Method | RL method | Learning rate | Entropy Coef. | SHD ($\downarrow$) | FDR ($\downarrow$) | TPR ($\uparrow$) |
|---|---|---|---|---|---|---|
| CORL | - | - | - | $82.4 \pm 22.3$ | $0.23 \pm 0.05$ | $0.87 \pm 0.04$ |
| DAGMA | - | - | - | $67.6 \pm 8.0$ | $0.14 \pm 0.02$ | $0.82 \pm 0.02$ |
| **ALIAS** (Ours) | A2C | 0.0001 | 0 | $\mathbf{1.20 \pm 0.4}$ | $\mathbf{0.00 \pm 0.00}$ | $\mathbf{1.00 \pm 0.00}$ |
| | | | 0.001 | $\mathbf{1.2 \pm 0.4}$ | $\mathbf{0.00 \pm 0.00}$ | $\mathbf{1.00 \pm 0.00}$ |
| | | | 0.01 | $\mathbf{1.6 \pm 0.5}$ | $\mathbf{0.01 \pm 0.00}$ | $\mathbf{1.00 \pm 0.00}$ |
| | | | 0.1 | $\mathbf{28.8 \pm 7.7}$ | $\mathbf{0.09 \pm 0.03}$ | $\mathbf{0.97 \pm 0.01}$ |
| | | | 1 | $127.0 \pm 13.1$ | $0.31 \pm 0.02$ | $0.78 \pm 0.03$ |
| | | 0.0005 | 0 | $\mathbf{2.2 \pm 1.3}$ | $\mathbf{0.01 \pm 0.00}$ | $\mathbf{1.00 \pm 0.00}$ |
| | | | 0.001 | $\mathbf{3.4 \pm 1.9}$ | $\mathbf{0.01 \pm 0.00}$ | $\mathbf{0.99 \pm 0.00}$ |
| | | | 0.01 | $\mathbf{0.2 \pm 0.2}$ | $\mathbf{0.00 \pm 0.00}$ | $\mathbf{1.00 \pm 0.00}$ |
| | | | 0.1 | $\mathbf{48.2 \pm 11.7}$ | $0.14 \pm 0.04$ | $\mathbf{0.93 \pm 0.01}$ |
| | | | 1 | $189.2 \pm 8.5$ | $0.40 \pm 0.01$ | $0.51 \pm 0.02$ |
| | PPO | 0.0001 | 0 | $\mathbf{1.2 \pm 0.4}$ | $\mathbf{0.00 \pm 0.00}$ | $\mathbf{1.00 \pm 0.00}$ |
| | | | 0.001 | $\mathbf{1.0 \pm 0.3}$ | $\mathbf{0.00 \pm 0.00}$ | $\mathbf{1.00 \pm 0.00}$ |
| | | | 0.01 | $\mathbf{1.6 \pm 0.5}$ | $\mathbf{0.01 \pm 0.00}$ | $\mathbf{1.00 \pm 0.00}$ |
| | | | 0.1 | $\mathbf{28.6 \pm 6.8}$ | $\mathbf{0.09 \pm 0.02}$ | $\mathbf{0.97 \pm 0.00}$ |
| | | | 1 | $130.2 \pm 12.1$ | $0.32 \pm 0.02$ | $0.77 \pm 0.03$ |
| | | 0.0005 | 0 | $\mathbf{0.4 \pm 0.3}$ | $\mathbf{0.00 \pm 0.00}$ | $\mathbf{1.00 \pm 0.00}$ |
| | | | 0.001 | $\mathbf{0.2 \pm 0.2}$ | $\mathbf{0.00 \pm 0.00}$ | $\mathbf{1.00 \pm 0.00}$ |
| | | | 0.01 | $\mathbf{1.6 \pm 1.3}$ | $\mathbf{0.00 \pm 0.00}$ | $\mathbf{1.00 \pm 0.00}$ |
| | | | 0.1 | $\mathbf{51.8 \pm 11.8}$ | $0.15 \pm 0.04$ | $\mathbf{0.92 \pm 0.01}$ |
| | | | 1 | $178.4 \pm 9.5$ | $0.38 \pm 0.02$ | $0.58 \pm 0.02$ |

### D.3.3  Effect of sparsity regularization

In Table 18, we conduct ablation studies on the effect of the choices of sparsity regularization strength of our **ALIAS** method with the LS score.

Table 18: Performance sensitivity of **ALIAS** with LS score subjected to the variations of Sparsity regularization weight for the LS score. We employ linear-Gaussian datasets with 30 nodes on ER-8 graphs and use CORL Wang et al., 2021 and DAGMA Bello et al., 2022 as reference. For each row, we compute the means and standard errors over 5 independent datasets. **Bold**: better performance than the baselines. Unless otherwise indicated, the remaining hyper-parameters are used according to Table 7.

| Method | RL method | Sparsity regularizer $\lambda_0$ | SHD ($\downarrow$) | FDR ($\downarrow$) | TPR ($\uparrow$) |
|---|---|---|---|---|---|
| CORL | - | - | $82.4 \pm 22.3$ | $0.23 \pm 0.05$ | $0.87 \pm 0.04$ |
| DAGMA | - | - | $67.6 \pm 8.0$ | $0.14 \pm 0.02$ | $0.82 \pm 0.02$ |
| **ALIAS** (Ours) | A2C | 0 | $\mathbf{8.4 \pm 4.2}$ | $\mathbf{0.02 \pm 0.01}$ | $\mathbf{0.98 \pm 0.01}$ |
| | | 0.000001 | $\mathbf{2.8 \pm 1.0}$ | $\mathbf{0.01 \pm 0.00}$ | $\mathbf{0.99 \pm 0.00}$ |
| | | 0.0001 | $\mathbf{50.0 \pm 27.7}$ | $\mathbf{0.07 \pm 0.04}$ | $0.83 \pm 0.10$ |
| | | 0.01 | $189.0 \pm 14.2$ | $0.34 \pm 0.04$ | $0.28 \pm 0.07$ |
| | | 1 | $226.4 \pm 4.7$ | $0.50 \pm 0.03$ | $0.08 \pm 0.02$ |
| | PPO | 0 | $\mathbf{2.0 \pm 0.5}$ | $\mathbf{0.01 \pm 0.00}$ | $\mathbf{1.00 \pm 0.00}$ |
| | | 0.000001 | $\mathbf{1.2 \pm 0.6}$ | $\mathbf{0.00 \pm 0.00}$ | $\mathbf{1.00 \pm 0.00}$ |
| | | 0.0001 | $\mathbf{42.0 \pm 26.4}$ | $\mathbf{0.04 \pm 0.02}$ | $0.84 \pm 0.10$ |
| | | 0.01 | $178.8 \pm 19.4$ | $0.29 \pm 0.05$ | $0.31 \pm 0.08$ |
| | | 1 | $221.6 \pm 6.3$ | $0.42 \pm 0.04$ | $0.09 \pm 0.02$ |

### D.4 Model Misspecification Results

In Table 19, we study causal model misspecification on nonlinear data. We assume data is generated via additive noise models with Gaussian process similarly to Zhu et al. (2020); Wang et al. (2021); Yang et al. (2023a), but test with datasets generated by the Post-nonlinear Gaussian Process model $X_i := \sigma\left(f_i\left(\mathbf{X}_{\mathrm{pa}_i}\right) + \mathrm{Laplace}(0, 1)\right)$, which is identifiable (Zhang & Hyvärinen, 2009) and produced by Lachapelle et al. (2020).

Table 19: Causal discovery performance on nonlinear data with PNL-GP model. The data is generated with 10-node ER-4 graphs and post nonlinear Gaussian processes as causal mechanisms. The performance metrics are Structural Hamming Distance (SHD), False Detection Rate (FDR), and True Positive Rate (TPR). Lower SHD and FDR values are preferable, while higher values are better for TPR. The numbers are *mean ± standard errors* over 5 independent runs. Since the graphs are dense, we also study the effect of pruning the output graphs.

| | No Pruning | | | CAM Pruning | | |
|---|---|---|---|---|---|---|
| Method | SHD ($\downarrow$) | FDR ($\downarrow$) | TPR ($\uparrow$) | SHD ($\downarrow$) | FDR ($\downarrow$) | TPR ($\uparrow$) |
| NOTEARS (Zheng et al., 2020) | $29.4 \pm 1.1$ | $0.47 \pm 0.02$ | $0.34 \pm 0.03$ | $29.4 \pm 1.1$ | $0.46 \pm 0.02$ | $0.34 \pm 0.03$ |
| DAGMA (Bello et al., 2022) | $27.6 \pm 3.0$ | $0.45 \pm 0.06$ | $0.38 \pm 0.06$ | $27.4 \pm 3.1$ | $0.51 \pm 0.08$ | $0.38 \pm 0.06$ |
| NOCURL (Yu et al., 2021) | $34.8 \pm 1.1$ | $0.51 \pm 0.08$ | $0.15 \pm 0.03$ | $34.8 \pm 1.1$ | $0.45 \pm 0.07$ | $0.15 \pm 0.03$ |
| COSMO (Massidda et al., 2024) | $29.4 \pm 0.8$ | $0.41 \pm 0.04$ | $0.33 \pm 0.02$ | $29.4 \pm 0.8$ | $0.41 \pm 0.04$ | $0.33 \pm 0.02$ |
| CORL (Wang et al., 2021) | $\mathbf{6.6 \pm 0.9}$ | $0.15 \pm 0.02$ | $\mathbf{0.95 \pm 0.01}$ | $\mathbf{4.2 \pm 0.7}$ | $0.08 \pm 0.02$ | $\mathbf{0.94 \pm 0.02}$ |
| RCL-OG (Yang et al., 2023a) | $8.6 \pm 1.4$ | $0.19 \pm 0.03$ | $\underline{0.90 \pm 0.02}$ | $7.2 \pm 1.3$ | $0.13 \pm 0.03$ | $\underline{0.89 \pm 0.02}$ |
| **ALIAS** (Ours) | $\underline{6.8 \pm 1.3}$ | $\mathbf{0.04 \pm 0.03}$ | $0.85 \pm 0.03$ | $\underline{6.8 \pm 1.3}$ | $\mathbf{0.03 \pm 0.03}$ | $0.84 \pm 0.03$ |

In addition, we also study the robustness of our method under noisy data. Specifically, we corrupt the data by adding Gaussian noises ($\sigma^2 = 0.1$) to $p\%$ random entries of the design matrix $\mathcal{D} \in \mathbb{R}^{n \times d}$, and report the results for the challenging 30-node ER-8 graphs in Table 20, showing that our method still consistently surpasses the baselines, even with increasing noise levels.

Table 20: Causal Discovery Performance under Noisy Data. The numbers are *mean ± standard error* over 5 random datasets on 30-node ER-8 graphs

| Method \ $p$ | 1% | 3% | 5% | 10% |
|---|---|---|---|---|
| NOTEARS | $181.4 \pm 4.0$ | $189.0 \pm 2.5$ | $190.8 \pm 2.8$ | $194.4 \pm 3.9$ |
| DAGMA | $67.0 \pm 8.5$ | $75.0 \pm 9.7$ | $79.0 \pm 6.5$ | $94.0 \pm 6.1$ |
| NOCURL | $142.2 \pm 4.2$ | $147.0 \pm 4.8$ | $146.0 \pm 5.1$ | $153.0 \pm 5.3$ |
| COSMO | $96.6 \pm 6.1$ | $112.8 \pm 7.6$ | $111.0 \pm 8.5$ | $136.2 \pm 14.3$ |
| **ALIAS** (Ours) | $\mathbf{8.4 \pm 0.8}$ | $\mathbf{27.2 \pm 3.0}$ | $\mathbf{41.4 \pm 5.0}$ | $\mathbf{66.0 \pm 4.0}$ |

Moreover, we investigate our method's performance under the dependence of noises, which is equivalent to the existence of hidden confounders. Specifically, we create datasets with hidden variables by generating datasets with $k$ additional variables then remove them. We present the results (for 30-node ER-8 graphs) on such datasets in Table 21, where our method also outperforms all baselines, even with more hidden confounders.

Table 21: Causal Discovery Performance under Hidden Confounders. The numbers are *mean ± standard error* over 5 random datasets on 30-node ER-8 graphs.

| Method \ $k$ | 1 | 2 | 3 | 4 |
|---|---|---|---|---|
| COSMO | $93.0 \pm 16.2$ | $133.6 \pm 11.2$ | $132.8 \pm 7.1$ | $136.4 \pm 8.5$ |
| DAGMA | $53.0 \pm 11.9$ | $77.6 \pm 9.3$ | $96.0 \pm 7.6$ | $124.0 \pm 5.3$ |
| NOCURL | $149.2 \pm 7.7$ | $159.8 \pm 8.4$ | $153.2 \pm 5.4$ | $158.4 \pm 5.3$ |
| NOTEARS | $185.2 \pm 10.4$ | $172.6 \pm 5.7$ | $173.2 \pm 5.0$ | $188.8 \pm 4.4$ |
| **ALIAS** (Ours) | $29.4 \pm 6.2$ | $58.2 \pm 4.9$ | $82.2 \pm 6.1$ | $106.6 \pm 2.5$ |

### D.5 Results on nonlinear data with MLPs

In this section, we study the performance of our ALIAS method on nonlinear causal models with neural networks, which are popular among gradient-based methods (Zheng et al., 2018; Bello et al., 2022; Massidda et al., 2024). Specifically, similar to the GP data, we consider 10-node ER-4 graphs with multiple-layer perceptron (MLP) causal mechanisms with standard Gaussian noise, as used in NOTEARS and DAGMA. These MLPs have one hidden layer with 100 units and sigmoid activation, and we generate 1,000 samples per dataset.

We employ the **ALIAS** variant that uses GP regressor with the exact configuration as in the GP experiments (Section 6.3), but now using the equal variance variant of the BIC score. This is compared against gradient-based baselines that support MLPs as is. These include NOTEARS, DAGMA, and COSMO, all of which model nonlinearity using MLPs with the same configuration, namely one hidden layer of 10 units with sigmoid activation.

The results are reported in Table 22, showing that despite the potential disadvantage of using GP regression for MLP-based SEMs, our method still achieves the lowest SHD of near zero compared to all three baselines, which are specifically designed to model MLP data.

Table 22: Causal discovery performance on nonlinear data with MLP model. The data is generated with 10-node ER-4 graphs and MLP model with one hidden layer of 100 units and sigmoid activation as causal mechanisms. The performance metrics are Structural Hamming Distance (SHD), False Detection Rate (FDR), and True Positive Rate (TPR). Lower SHD and FDR values are preferable, while higher values are better for TPR. The numbers are *mean ± standard errors* over 5 independent runs. Since the graphs are dense, we also study the effect of pruning the output graphs.

| | No Pruning | | | CAM Pruning | | |
|---|---|---|---|---|---|---|
| Method | SHD ($\downarrow$) | FDR ($\downarrow$) | TPR ($\uparrow$) | SHD ($\downarrow$) | FDR ($\downarrow$) | TPR ($\uparrow$) |
| NOTEARS (Zheng et al., 2020) | $11.0 \pm 1.2$ | $0.21 \pm 0.01$ | $0.93 \pm 0.04$ | $10.0 \pm 1.9$ | $0.11 \pm 0.03$ | $0.80 \pm 0.04$ |
| DAGMA (Bello et al., 2022) | $7.6 \pm 1.7$ | $0.09 \pm 0.03$ | $0.84 \pm 0.05$ | $8.8 \pm 1.6$ | $0.07 \pm 0.03$ | $0.78 \pm 0.04$ |
| COSMO (Massidda et al., 2024) | $9.4 \pm 1.0$ | $0.14 \pm 0.01$ | $0.87 \pm 0.03$ | $9.8 \pm 1.2$ | $0.09 \pm 0.02$ | $0.79 \pm 0.02$ |
| **ALIAS** (Ours) | $\mathbf{0.8 \pm 0.4}$ | $\mathbf{0.01 \pm 0.01}$ | $\mathbf{0.99 \pm 0.01}$ | $\mathbf{3.8 \pm 0.4}$ | $\mathbf{0.00 \pm 0.00}$ | $\mathbf{0.89 \pm 0.01}$ |

## D.6 Results on regular weight range $\mathcal{U}\left([-2, -0.5] \cup [0.5, 2]\right)$

In addition to our experiments on linear-Gaussian data with the large weight range $\mathcal{U}\left([-5, -2] \cup [2, 5]\right)$ in Figure 2, here we also consider the regular range $\mathcal{U}\left([-2, -0.5] \cup [0.5, 2]\right)$ for the linear weights. In Figure 6 we present the causal discovery results for both weight ranges under varying graph sizes. It can be seen that larger weights pose significant challenges to several methods, including NOTEARS, NOCURL, RL-BIC, RCL-OG, and CORL. Meanwhile, our method consistently identifies true DAG in all cases for both weight ranges, signifying its robustness to the data variance.

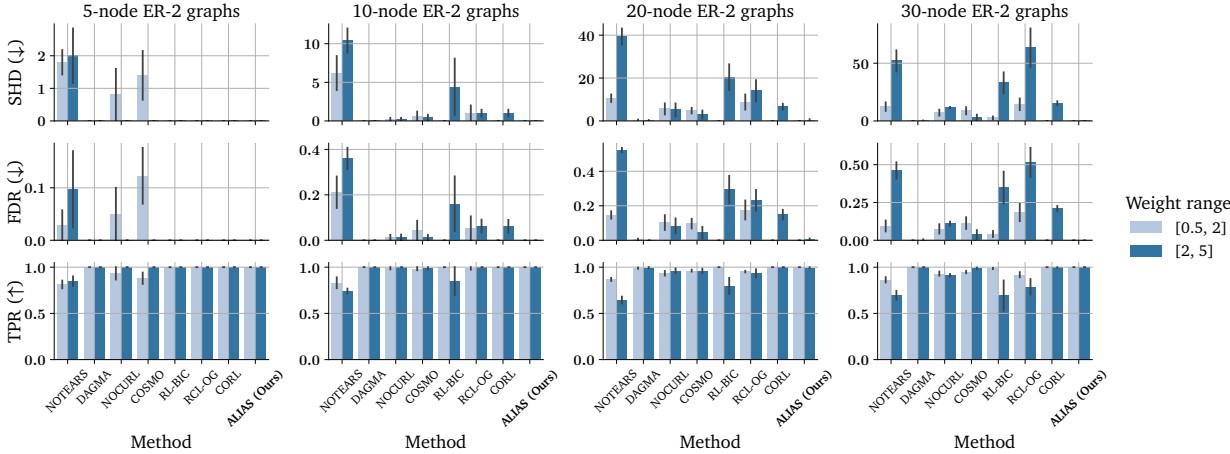

Figure 6: Causal Discovery Performance on linear-Gaussian data with small ($\mathcal{U}\left([-2, -0.5] \cup [0.5, 2]\right)$) and large ($\mathcal{U}\left([-5, -2] \cup [2, 5]\right)$) weight ranges. The error bars depict standard errors over 5 independent runs.

## D.7 Comparison with Constraint-based methods

In addition to the score-based baselines considered so far, in this section, we also investigate the comparative performance of our **ALIAS** method against constraint-based methods, which are also a prominent approach in causal discovery. To be more specific, we consider the classical PC method (Spirtes et al., 2000), as well as more recent advances MARVEL (Mokhtarian et al., 2021) and RSL (Mokhtarian et al., 2022). In this case, we emply gCastle's implementation for PC and the official implementations provided at `https://rcdpackage.com/` for RSL and MARVEL.

We evaluate these methods on linear-Gaussian datasets with varying scales and densities, and the sample size is fixed to 1,000. The baselines were configured according to their recommended settings, including the Fisher's z test with significance levels of 0.05 for PC and $\frac{2}{d^2}$ for RSL and MARVEL, where $d$ is the number

of nodes. Since constraint-based methods may not orient all edges, for a fair evaluation, we compare the undirected skeletons of the estimated and true graphs. The evaluation metrics included SHD, precision, recall, and F1 scores. The results presented in Table 23 reveal that our method demonstrates significantly higher accuracy than constraint-based approaches in recovering the skeleton, achieving near-zero skeleton SHD across all scenarios. In contrast, the baseline methods face considerable challenges, particularly when applied to large and dense graphs.

Table 23: Causal Discovery Performance in Comparison with Constraint-based Methods. We compare the proposed method **ALIAS** with PC (Spirtes et al., 2000), MARVEL (Mokhtarian et al., 2021), and RSL (Mokhtarian et al., 2022) on linear-Gaussian datasets with the regular weight range $\mathcal{U}\left([-2, -0.5] \cup [0.5, 2]\right)$. The performance metrics are SHD (lower is better), Precision (higher is better), Recall (higher is better), and $F_1$ score (higher is better) between the skeletons of the estimated and true graphs. The numbers are *mean $\pm$ standard errors* over 5 independent runs.

| Data | Method | Skeleton SHD ($\downarrow$) | Skeleton Precision ($\uparrow$) | Skeleton Recall ($\uparrow$) | Skeleton $F_1$ ($\uparrow$) |
|---|---|---|---|---|---|
| Sparse graphs (30-node ER-2) | PC (Spirtes et al., 2000) | $25.8 \pm 2.8$ | $0.88 \pm 0.03$ | $0.64 \pm 0.03$ | $0.74 \pm 0.03$ |
| | MARVEL (Mokhtarian et al., 2021) | $15.6 \pm 2.1$ | $0.93 \pm 0.02$ | $0.79 \pm 0.03$ | $0.85 \pm 0.02$ |
| | RSL (Mokhtarian et al., 2022) | $13.8 \pm 1.5$ | $0.9 \pm 0.02$ | $0.86 \pm 0.01$ | $0.88 \pm 0.01$ |
| | **ALIAS (Ours)** | $\mathbf{0.0 \pm 0.0}$ | $\mathbf{1.0 \pm 0.0}$ | $\mathbf{1.0 \pm 0.0}$ | $\mathbf{1.0 \pm 0.0}$ |
| Dense graphs (30-node ER-8) | PC (Spirtes et al., 2000) | $216.0 \pm 3.6$ | $0.65 \pm 0.02$ | $0.13 \pm 0.01$ | $0.21 \pm 0.02$ |
| | MARVEL (Mokhtarian et al., 2021) | $221.8 \pm 3.8$ | $0.71 \pm 0.03$ | $0.05 \pm 0.0$ | $0.1 \pm 0.01$ |
| | RSL (Mokhtarian et al., 2022) | $185.4 \pm 5.6$ | $0.66 \pm 0.02$ | $0.39 \pm 0.01$ | $0.49 \pm 0.01$ |
| | **ALIAS (Ours)** | $\mathbf{0.4 \pm 0.2}$ | $\mathbf{1.0 \pm 0.0}$ | $\mathbf{1.0 \pm 0.0}$ | $\mathbf{1.0 \pm 0.0}$ |
| Large graphs (200-node ER-2) | PC (Spirtes et al., 2000) | $216.6 \pm 10.5$ | $0.87 \pm 0.01$ | $0.57 \pm 0.02$ | $0.69 \pm 0.01$ |
| | MARVEL (Mokhtarian et al., 2021) | $167.2 \pm 8.4$ | $0.91 \pm 0.01$ | $0.67 \pm 0.01$ | $0.77 \pm 0.01$ |
| | RSL (Mokhtarian et al., 2022) | $167.2 \pm 8.8$ | $0.87 \pm 0.01$ | $0.7 \pm 0.01$ | $0.78 \pm 0.01$ |
| | **ALIAS (Ours)** | $\mathbf{0.8 \pm 0.6}$ | $\mathbf{1.0 \pm 0.0}$ | $\mathbf{1.0 \pm 0.0}$ | $\mathbf{1.0 \pm 0.0}$ |

## D.8  On Varsortability

Recently, it was suggested by Reisach et al. (2021) that marginal variances share high dependencies with the topological ordering of the true DAG for simple additive noise models. As such, sorting the variables according to the variances may be sufficient to recover the DAG, voiding the need for more sophisticated methods. More particularly, the metric `Varsortability` $\in [0, 1]$ is proposed to assess the correlation between the ordering of marginal data variances and the causal ordering of the true DAG. Essentially, `Varsortability` calculates the portion of directed paths in the ground-truth DAG that have increasing marginal variances, and thus, if it is close to 1 then there is a high chance that simply sorting the variances would reveal the true DAG, and vice versa. It was shown that simple additive noise data commonly employed in existing studies, starting with Zheng et al. (2018), exhibit high values of `Varsortability`.

However, we show here that this is not entirely the case for our study. Particularly, while the majority of existing studies employ *sparse* graphs for evaluation, where the densest graphs have an expected in-degree of about 4 at most (Zheng et al., 2018; Yu et al., 2021; Bello et al., 2022; Massidda et al., 2024), in this study, we also consider much *denser* graphs with an expected in-degree of up to 8. This distinction is of high importance because we find that `Varsortability` decreases rapidly with the graph density and reaches near-zero at ER-8 graphs (Table 24), indicating that our data settings is much more nontrivial, and cannot be resolved simply by sorting the variances. Furthermore, since `Varsortability` is close to 0 in our data, one may expect that simply sorting the nodes by *decreasing* variances may work. In Table 25 we show that this is not the case, where the `sortnregress` algorithm proposed in conjunction to `Varsortability` (Reisach et al., 2021)—which first sorts the nodes by marginal variances and then perform linear feature selection to convert the ordering to a DAG—cannot recover the true DAG regardless of the sorting direction. Nevertheless, under this intricate scenarios, our method still robustly recovers the true DAG with a near-zero SHD.

Table 24: **Varsortability as function of graph density**. We compute Varsortability (Reisach et al., 2021) for linear-Gaussian data on 30-node ER graphs (Section. 6.2) with varying expected in-degrees. The numbers are *mean ± standard errors* over 100 simulations.

| Expected in-degree | Varsortability |
|:---:|:---:|
| 1 | $0.97 \pm 0.01$ |
| 2 | $0.87 \pm 0.03$ |
| 4 | $0.35 \pm 0.06$ |
| 6 | $0.05 \pm 0.01$ |
| 8 | $0.01 \pm 0.00$ |

Table 25: **Comparison with sortnregress on dense data**. We consider linear-Gaussian datasets on dense 30-node ER-8 graphs. The numbers are *mean ± standard errors* over 5 simulations.

| Method | SHD ($\downarrow$) | FDR ($\downarrow$) | TPR ($\uparrow$) |
|:---:|:---:|:---:|:---:|
| sortnregress (decreasing variances) | $360.2 \pm 2.2$ | $0.97 \pm 0.01$ | $0.04 \pm 0.01$ |
| sortnregress (increasing variances) | $88.2 \pm 17.7$ | $0.27 \pm 0.05$ | $0.94 \pm 0.01$ |
| **ALIAS (ours)** | $\mathbf{0.2 \pm 0.2}$ | $\mathbf{0.00 \pm 0.00}$ | $\mathbf{1.00 \pm 0.00}$ |

## D.9 Effect of Data Standardization

Given the issues raised by Reisach et al. (2021) as discussed in the previous section, it is interesting to study the performance of causal discovery methods under data standardization, where the variance ordering information is completely removed.

**Linear-Gaussian data.** For linear-Gaussian data, the causal graph is identifiable under the equal-noise variance assumption (Peters et al., 2014). Hence, standardizing our linear-Gaussian data node-wise would render the noise variances unequal and reduce the causal model to a general linear-Gaussian model, where the causal graph is known to be *unrecoverable* (Spirtes et al., 2000). As such, it is expected that causal discovery performance on standardized linear data is worse than its non-standardized counterparts. However, in Table. 26 we show that our **ALIAS** method can still significantly outperform the baselines under this setting with the lowest SHD, which is only half of that of the second-best method, highlighting **ALIAS**'s robustness against data standardization.

**Nonlinear data.** On the contrary with linear causal models, nonlinear additive noise models are generally identifiable regardless of unequal noise variances (Hoyer et al., 2008), meaning standardizing data should have minimal influence on the causal discovery performance. Indeed, in Table. 26 we show that our **ALIAS** method still performs equally well when the data is standardized compared to when the data is not standardized (Table. 4) with a very low SHD of only around 1, which is much lower relative to other baselines.

Table 26: **Causal Discovery performance on standardized data**. We consider linear-Gaussian data and nonlinear data with Gaussian processes. Observed data is standardized dimension-wise to remove variance information. For **ALIAS**, we use the non-equal variances BIC version to reflect the data condition. The numbers are *mean ± standard errors* over 5 independent runs. **Bold**: best performance, underline: second-best performance.

| | Linear data (10-node ER-2 graphs) | | | Nonlinear data (10-node ER-4 graphs) | | |
|:---|:---:|:---:|:---:|:---:|:---:|:---:|
| Method | SHD ($\downarrow$) | FDR ($\downarrow$) | TPR ($\uparrow$) | SHD ($\downarrow$) | FDR ($\downarrow$) | TPR ($\uparrow$) |
| NOTEARS (Zheng et al., 2020) | $\underline{10.2 \pm 2.0}$ | $\underline{0.39 \pm 0.04}$ | $\mathbf{0.84 \pm 0.02}$ | $7.8 \pm 2.3$ | $0.14 \pm 0.07$ | $0.91 \pm 0.01$ |
| DAGMA (Bello et al., 2022) | $17.4 \pm 2.8$ | $0.70 \pm 0.08$ | $0.39 \pm 0.07$ | $29.8 \pm 5.5$ | $0.4 \pm 0.16$ | $0.28 \pm 0.15$ |
| COSMO (Massidda et al., 2024) | $16.4 \pm 1.9$ | $0.74 \pm 0.06$ | $0.29 \pm 0.06$ | $30.8 \pm 4.9$ | $0.5 \pm 0.16$ | $0.27 \pm 0.12$ |
| **ALIAS (ours)** | $\mathbf{6.0 \pm 1.8}$ | $\mathbf{0.30 \pm 0.07}$ | $\underline{0.79 \pm 0.05}$ | $\mathbf{1.4 \pm 0.9}$ | $\mathbf{0.0 \pm 0.01}$ | $\mathbf{0.97 \pm 0.03}$ |

