# OpenReview forum: "Reinforcement Learning for Causal Discovery without Acyclicity Constraints"
_TMLR — Accepted by TMLR_

### Review · Reviewer_w4vc · 2024-12-22

**Summary Of Contributions:**

The authors present a novel parameterization called *Vec2DAG* of a directed acyclic graph (DAG) in terms of a continuous, real-valued vector. The proposed parameterization alleviates the requirement of laboriously maintaining the acyclicity constraint as it is satisfied by construction. The authors prove that Vec2DAG is theoretically sound in that any DAG can be represented with the given parameterization. In addition, the authors propose *ALIAS*, a score-based causal discovery algorithm based on reinforcement learning that optimizes over the continuous vector to estimate the Markov equivalence class of the underlying causal graph, using only observational data. Plenty of experiments and ablation studies on synthetic and real-world data demonstrate the merits of ALIAS in comparison to prior work.

**Audience:**

Yes

**Broader Impact Concerns:**

None.

**Claims And Evidence:**

Yes

**Requested Changes:**

I will proceed to list requested changes one-by-one. Points that are critical for meeting the acceptance criteria are highlighted by *(critical)*.

**Title**

- *Reinforced DAG Learning without Acyclicity Constraints Journal Submissions* In my opionion, the title should be modified. Why does it say *Journal Submissions*? Also I am not sure if *reinforced DAG learning* makes sense (I could be wrong). Why not just choose a title like *Reinforcement Learning for Causal Discovery without Acyclicity Constraints*?

**Introduction**

- *The knowledge of causal relationships required to understand the nature in many scientific sectors,* This sentence does not seem gramatically correct. Perhaps the authors mean something like *Obtaining insights into causal relationships has traditionally required deep domain knowledge*. Please clarify.

- *, especially in intricate situations where randomized experiments are impractical, has motivated the development of causal discovery methods that aim to infer cause-effect relationships from purely passive data over the last decade* This should be an individual sentence. The gist is roughly intelligible, but the grammar and sentence structure is quite poor. I kindly ask the authors to rewrite this sentence.

- *and the intricate acyclicity constraint* Is that really true? I believe that removing the acyclicity constraint makes the problem even harder, because even more graphs are considered.

- *brief overview of the literature in Appendix A* Given that there is no page limit, I do not understand why the literature review is in the appendix. I would kindly ask the authors to put it into the main text or clarify why this decision was made.

**Vec2DAG: Unconstrained Parametrization of DAGs**

- Equation 3: It would be great if the authors could go a bit more into the intuition for Vec2DAG. To me, it seems that having both the node potential and the edge potential is redundant. My confusion may have to do with the fact that I do not quite understand where these potentials even come from. Are they learned? It would be great if the authors could clarify this with a few words.

**3.2 Properties of Vec2DAG**

- I do not understand how Lemma 1 is about *scaling and translation invariance*. To me, the statement of the lemma rather reads as some kind of continuity statement: Two sufficiently close vectors w.r.t. to their potentials yield the same DAG. In this light, I also do not understand the remark *representations of different DAGs are close to each other*. Please clarify.

- Lemma 2 does make sense to me with the given explanation, but it seems to show something very different to Lemma 1. It would be nice to have some more explanation for how these two lemmas relate to each other.

**4.1 Policy Gradient for DAG Search**

- (critical) I do not understand how it is justified to bring reinforcement learning into the game here (see weakness 2). Without an explanation, I am not convinced of the soundness of the proposed method. The authors need to explain why the same results could not be achieved via straight-forward optimization.

- (critical) ALIAS seems to optimize only over $\mathbf{z}$. I did not understand how the edge and node potentials are parameterized by $\mathbf{z}$ (I.e., what are $\mathbf{E}(\mathbf{z})$ and $\mathbf{p}(\mathbf{z})$?). This seems to be highly relevant to the method, so it would be important to discuss this.

**4.2 Post Processing**

- *With limited sample sizes, due to overfitting, redundant edges may still be present in the returned DAG whose highest score* This sentence seems gramatically wrong and makes no sense to me. I guess the authors mean something like *With limited sample sizes, due to overfitting, redundant edges may still be present in the returned DAG that achieves the highest score*. Please clarify.

**5.6 Ablation Studies**

- *Furthermore, to show that the effectiveness of ALIAS is not only thanks to the Vec2DAG parametrization alone, but also the application of RL, we replace RL in our method with a continuous optimizer* This is exactly where I am sceptical. I understand that ALIAS is about optimizing a non-differentiable objective function. However, that does not mean that it is about reinforcement learning. I believe that ALIAS is a zero-order optimization algorithm and the comparison should be with respect to other zero-order optimization algorithms. Furthermore, it is not clear to me what a *continuous optimizer* is. I guess the authors mean *first-order optimizer* or *gradient-based optimizer*.

**Appendix**

- (critical) *These methods can only identify the causal graph up to its Markov equivalent class (MEC), which contains potentially many DAGs inducing the same data* I do not think that ALIAS can generally discover the causal graph beyond Markov equivalence class. Score-based methods, from what I know, generally cannot do that. If ALIAS can actually do that, this would be quite amazing. Therefore, if the authors claim that this is true, it must be proved. Otherwise, the claim must be removed.

**Strengths And Weaknesses:**

**Strengths**

1. The contribution is quite solid. It seems this simple, yet novel mapping from vectors to DAGs (Vec2DAG) has strong merits over prior approaches, which is demonstrated both theoretically and experimentally.

2. Overall, the writing is very good and the text is easy to follow. I also think the figures are very pretty and clean.

3. Extensive experiments on both synthetic and real-world data are presented, with plenty of ablations. The authors did an amazing job here.

**Weaknesses**

1. No source code for reproducing the experiments seems to be available. It would be great if the authors could supply this for maximum reproducibility and to make the lifes of other authors easier in the future.

2. I am not totally convinced that it is justified to associate ALIAS with reinforcement learning. The policy takes only a single step and there are no states, transition dynamics or stochasticity. To my understanding, the latter three points are what fundamentally characterizes reinforcement learning. I see that the authors compare their method to first-order optimization of a differentiable version of Vec2DAG. However, I would still interpret the proposed ALIAS method as some kind of zero-order optimization rather than reinforcement learning. This point should certainly be discussed in greater detail (for instance, having a discussion section for this would be nice).

3. The related work section belongs in the main text. There is no page limit for TMLR, so I do not see a sound reason to put this section in the appendix.

Apart from that, I would be glad if the authors could take into account my requested changes for the revised manuscript.

---

> ### Author Response · Authors · 2025-01-12
> **Response to Reviewer w4vc (1/4)**
>
> We sincerely thank Reviewer w4vc for the thoughtful comments and valuable suggestions. We have carefully revised our manuscript to incorporate all of your requested changes, which we believe have significantly enhanced its overall quality and presentation. Below, we address your major concerns and outline the corresponding changes point by point:
>
> **No source code for reproducing the experiments seems to be available. It would be great if the authors could supply this for maximum reproducibility and to make the lifes of other authors easier in the future.**
>
> As per your request, we have included the source code necessary to reproduce our experiments in the Supplementary material. We have taken steps to ensure the code is user-friendly, easy to execute, and designed to facilitate further development.
>
> **I am not totally convinced that it is justified to associate ALIAS with reinforcement learning**
>
> We would like to emphasize that our study is fundamentally rooted in RL. First, our method is inspired by and builds upon the established line of RL-based approaches (RL-BIC, CORL, RCL-OG, etc.) Specifically, we follow the idea introduced by RL-BIC, the very first study to apply RL to score-based causal discovery, which suggests that "an RL agent with a stochastic policy can automatically determine where to search given the uncertainty information of the learned policy, which can be promptly updated by the stream of reward signals." Our method adopts this RL framework and improves upon the DAG-generating policy, thereby firmly establishing ALIAS as an RL-based method. We have elaborated in detail on the development of this line of RL research in the "Related Work" section, which now has been moved to the main text for clarity, as per your request.
>
> Second, our optimization objective—the expected return, which is central to policy gradient—and the gradient formula are directly derived from RL theory. Hence, disassociating ALIAS from RL while utilizing these RL-specific components would be unnatural. In the revised manuscript, we have provided a clear rationale for employing these components: the expected return allows ALIAS to update its parameters after exploring multiple DAGs, promoting broader exploration and resulting in higher DAG scores. Third, ALIAS is strongly associated with RL because it enables a clear adaptability of various RL methods, such as vanilla policy gradient, A2C, and PPO, to optimize our objective, as mentioned in Section "Policy Gradient for DAG Search". For these reasons, it would not be appropriate to exclude RL from the foundation of ALIAS.
>
> **The policy takes only a single step and there are no states, transition dynamics or stochasticity. To my understanding, the latter three points are what fundamentally characterizes reinforcement learning.**
>
> This is not necessarily the case. First, the original RL-based method, RL-BIC, also employed a policy with a single step and did not have states and transition dynamics. Specifically, it learns a policy that outputs a graph in just one step. Despite this, the power of RL was still effectively leveraged, with policy gradient successfully used to optimize the policy. This demonstrates that the applicability of RL is not limited to multi-step problems, as a one-step environment still qualifies as a valid Markov decision process and thus remains compatible with most RL algorithms.
>
> Second, one-step RL, where an agent learns from feedback or rewards received after taking a single action in an environment, without considering long-term consequences or sequences of actions, is also a well-recognized approach in RL [1, 2]. This again justifies the application of RL to our one-step environment.
>
> Third, we would like to point out that our method does have stochasticity in its policy, which provides an exploration capability inherently tied to RL. This exploration ability is crucial for the accuracy of our method, as it enables the exploration of a significantly larger number of DAGs compared to existing gradient-based methods, which only explore along the path suggested by the gradient, resulting in the exploration of fewer DAGs. Because our stochastic policy facilitates broader exploration, it leads to the discovery of better solutions. This is clearly demonstrated in Figures 1c and 1d, where the number of unique DAGs explored by our method constantly increases during learning, resulting in a sharp rise in the reward (which is also the graph score).
>
> **References:**
>
> [1] Achab et al. "One-step distributional reinforcement learning." TMLR, 2023.
>
> [2] Ghraieb et al. "Single-step deep reinforcement learning for open-loop control of laminar and turbulent flows." Physical Review Fluids, 2021.

---

> ### Author Response · Authors · 2025-01-12
> **Response to Reviewer w4vc (2/4)**
>
> **However, I would still interpret the proposed ALIAS method as some kind of zero-order optimization rather than reinforcement learning.**
>
> As previously explained, ALIAS is fundamentally rooted in the RL framework, with RL serving as the theoretical foundation of every component of our optimization, including the formulation of the expected reward objective $J(\theta)=\mathbb{E}[R(z)]$ and the policy gradient update rule $\nabla_\theta J(\theta)=\mathbb{E}[\nabla_\theta \ln \pi_\theta(z) R(z)]$. Particularly, it is not trivial why the expectation is used and how the gradient is derived without referencing the policy gradient theory. Therefore, it is more appropriate to view ALIAS as an RL-based zero-order optimization method rather than disregarding its connection to RL entirely.
>
> **This point should certainly be discussed in greater detail (for instance, having a discussion section for this would be nice)**
>
> Following your suggestion, we have incorporated a discussion in a new Section 5.1 (Motivation for Reinforcement Learning) in the revised manuscript, highlighting the points raised in this rebuttal regarding the RL's significance in ALIAS. We sincerely appreciate your valuable input.
>
> **In my opionion, the title should be modified [...]**
>
> Following your feedback, we have changed the title to "Reinforcement Learning for Causal Discovery without Acyclicity Constraints".
>
> **Introduction: [...] I kindly ask the authors to rewrite this sentence.**
>
> We have revised the premise as per your suggestion, and the changes are highlighted in red in the revised manuscript. Thank you for bringing this to our attention.
>
> **and the intricate acyclicity constraint Is that really true? I believe that removing the acyclicity constraint makes the problem even harder, because even more graphs are considered.**
>
> We want to clarify that our point here is simply that maintaining the acyclicity constraint during the search process adds complexity compared to searching without any constraints. This is because checking for acyclicity has historically been challenging due to its combinatorial nature, which is why the ingenious soft constraint approach in NOTEARS was developed.
>
> **Given that there is no page limit, I do not understand why the literature review is in the appendix. I would kindly ask the authors to put it into the main text or clarify why this decision was made.**
>
> We would like to point out that during submission, we were given the option to choose between a regular submission (12 pages) and a long submission (more than 12 pages). We opted for the former, which necessitated moving our Related Work section to the Appendix due to the space constraint. However, in the revision, we have relocated the Related Work section to Section 2 and will adjust the length as needed if further changes are required.
>
> **Equation 3: It would be great if the authors could go a bit more into the intuition for Vec2DAG. To me, it seems that having both the node potential and the edge potential is redundant. My confusion may have to do with the fact that I do not quite understand where these potentials even come from. Are they learned? It would be great if the authors could clarify this with a few words.**
>
> We would like to clarify that in our method, the potentials are learned, and having both node potential and edge potential is not redundant. To further clarify the intuition behind Vec2DAG, we have added the following explanation to the revised manuscript:
> - "Specifically, $\mathbf{p}(\mathbf{z})$ represents the node potential vector formed by the first $d$ elements of $\mathbf{z}$, while $\mathbf{E}(\mathbf{z})$ is the edge potential matrix, with the elements above the main diagonal derived from the last $\frac{d \cdot (d-1)}{2}$ elements of $\mathbf{z}$ (see our code in Figure 5)."
> - "The intuition behind Vec2DAG is that the first term in Eq. (3) defines a symmetric binary adjacency matrix, determining whether two nodes are connected. The directions of these connections are then dictated by the second term in Eq. (3), resulting in a binary matrix that represents a directed graph. Additionally, this directed graph is guaranteed to be acyclic due to the use of the gradient flow operator."

---

> ### Author Response · Authors · 2025-01-12
> **Response to Reviewer w4vc (3/4)**
>
> **I do not understand how Lemma 1 is about scaling and translation invariance. To me, the statement of the lemma rather reads as some kind of continuity statement: Two sufficiently close vectors w.r.t. to their potentials yield the same DAG. In this light, I also do not understand the remark representations of different DAGs are close to each other. Please clarify.**
>
> We would like to clarify that Lemma 1 does not state "Two sufficiently close vectors w.r.t. their potentials yield the same DAG." Instead, it asserts that scaling the potential by any positive constant $\alpha$ results in the same DAG ($\text{Vec2DAG}(\mathbf{z}) = \text{Vec2DAG}(\alpha \cdot \mathbf{z})$), and translating the potential by an amount $\beta$ (which can be large, provided it does not change the ordering of $\mathbf{p}$ or the element-wise positivity of $\mathbf{E}$) also results in the same DAG ($\text{Vec2DAG}(\mathbf{z}) = \text{Vec2DAG}(\mathbf{z} + \beta)$). We have added this clarification to the revised manuscript to improve the understanding of Lemma 1.
>
> **Lemma 2 does make sense to me with the given explanation, but it seems to show something very different to Lemma 1. It would be nice to have some more explanation for how these two lemmas relate to each other.**
>
> We would like to clarify that Lemma 1 directly leads to Lemma 2. Specifically, as mentioned in the manuscript, Lemma 1 establishes that any DAG can be represented in infinitely many ways through scaling and translation. Consequently, for any two distinct DAGs, it is always possible to find their representations within an arbitrarily small ball of radius $\epsilon > 0$ around the origin, by scaling arbitrary representations of them. This demonstrates that two different DAGs can have representations that are arbitrarily close to each other, which is precisely the implication of Lemma 2. We have included this clarification in the revised manuscript.
>
> **I do not understand how it is justified to bring reinforcement learning into the game here (see weakness 2). Without an explanation, I am not convinced of the soundness of the proposed method. The authors need to explain why the same results could not be achieved via straight-forward optimization.**
>
> We would like to clarify that we did not claim that "the same results could not be achieved via straightforward optimization." While we acknowledge that RL is not the only solution to our optimization problem, it is a valid and very practical approach, as supported by prior works. Furthermore, while we recognize the existence of many black-box optimization frameworks that may solve our objective, our focus was intentionally limited to specifically investigating the RL approach, like previous RL-based studies, as thoroughly studying multiple optimization methods is too big of a scope for our study.
>
> That said, we argue that our optimization problem is not "straightforward" for existing techniques due to the high dimensionality of the search space and the large number of trials potentially required. Specifically, consider the maximization of $S(\text{Vec2DAG}(\mathbf{z}))$ over $\mathbf{z} \in \mathbb{R}^{d(d+1)/2}$ (rather than the expected reward objective used in our RL approach). This optimization is challenging because the dimensionality of the search space grows quadratically with $d$. For instance, with as few as 30 nodes, the search space already spans 465 dimensions. Black-box optimization techniques, such as Bayesian Optimization (BO), may be considered "straightforward" frameworks but are known to perform poorly in high-dimensional settings and with a large number of trials. BO typically scales poorly, with cubic growth in computational cost relative to the number of trials, and is generally limited to problems with only tens of dimensions and hundreds of trials. In contrast, our RL-based approach can efficiently handle millions of trials (see Figure 1) because it scales linearly with both dimensionality and the number of trials, making it very well-suited to our problem.
>
> In short, while alternative optimization approaches may be feasible, addressing our optimization problem requires a carefully chosen and well-designed method that can handle both high dimensionality and large number of trials. This renders the problem non-trivial for existing techniques and highlights RL as an effective and practical solution. We have incorporated this point to our revised manuscript. Thank you for pointing this out.

---

> ### Author Response · Authors · 2025-01-12
> **Response to Reviewer w4vc (4/4)**
>
> **ALIAS seems to optimize only over $z$. I did not understand how the edge and node potentials are parameterized by (I.e., what $p(z)$ are and $E(z)$?). This seems to be highly relevant to the method, so it would be important to discuss this.**
>
> In the section "The Vec2DAG Operator," we have explained that $z$ is formed by combining the node potentials $p$ and the edge potentials $E$, and we denoted the node and edge potential components of $z$ as $p(z)$ and $E(z)$, respectively. Additionally, the code in Figure 5 illustrates that $z$ is the only learnable parameter, where $p(z)$ corresponds to the first $d$ elements of $z$, and $E(z)$ corresponds to the remaining $d(d-1)/2$ dimensions. We have provided a clearer explanation of this in the revised manuscript.
>
> **With limited sample sizes, due to overfitting, redundant edges may still be present in the returned DAG whose highest score This sentence seems gramatically wrong and makes no sense to me. I guess the authors mean something like With limited sample sizes, due to overfitting, redundant edges may still be present in the returned DAG that achieves the highest score. Please clarify.**
>
> Thank you for your suggestion. Your understanding is correct and we have fixed our phrasing according to your recommendation.
>
> **Furthermore, to show that the effectiveness of ALIAS is not only thanks to the Vec2DAG parametrization alone, but also the application of RL, we replace RL in our method with a continuous optimizer. This is exactly where I am sceptical. I understand that ALIAS is about optimizing a non-differentiable objective function. However, that does not mean that it is about reinforcement learning. I believe that ALIAS is a zero-order optimization algorithm and the comparison should be with respect to other zero-order optimization algorithms.**
>
> We would like to clarify that the motivation for this ablation study was to address potential concerns from readers about whether a standard gradient-based approach could work well with our new Vec2DAG representation, as this could make RL unnecessary in our method. We did not intend to prove that RL is absolutely necessary compared to every other optimization approach. While we acknowledge the existence of many zero-order optimization algorithms, we chose to compare only with gradient-based optimization because it is the most commonly considered method in the causal discovery literature. In contrast, the application of other zero-order optimization techniques, such as Bayesian optimization and genetic programming, to causal discovery is less well-established or even nonexistent (as of the time of writing our manuscript). These approaches deserve independent studies, as our optimization problem, as explained earlier, is not straightforward. On the other hand, RL is also a zero-order optimization method that has been successfully applied to causal discovery, making it a natural comparison between RL and gradient-based optimization.
>
> **These methods can only identify the causal graph up to its Markov equivalent class (MEC), which contains potentially many DAGs inducing the same data. I do not think that ALIAS can generally discover the causal graph beyond Markov equivalence class. Score-based methods, from what I know, generally cannot do that. If ALIAS can actually do that, this would be quite amazing. Therefore, if the authors claim that this is true, it must be proved. Otherwise, the claim must be removed.**
>
> We would like to clarify that our point was that score-based methods, when combined with additional parametric assumptions, have the potential to discover the true causal graph beyond the Markov equivalence class, provided the causal model is identfiable, whereas constraint-based methods typically do not. For example, in the case of linear-Gaussian data with equal noise variances (which is identifiable), the BIC score, thanks to its consistency, will assign the highest score to the true DAG, allowing it to be recovered. We have formalized this in Lemma 1 and provided the theoretical proof in Appendix B.1, and our empirical evidence strongly agrees with this, where our method consistently achieves zero-SHD in nearly all instances of this model. In contrast, constraint-based methods like PC typically do not incorporate assumptions about the causal mechanism and are thus limited to recovering the MEC. That said, in response to your request, we have removed this statement to avoid any further confusion.
>
> We hope this rebuttal adequately addresses your concerns, and we are open to further discussions to resolve any further issues. In addition, we kindly ask for the opportunity to address any additional concerns you may have before a final decision is made.

---

> > ### Comment · Reviewer_w4vc · 2025-01-13
> >
> > I would like to thank the authors for the studious revision. I would like to comment on two points. For clarity, I will quote my own points and not the response.
> >
> > **I am not totally convinced that it is justified to associate ALIAS with reinforcement learning**
> >
> > Generally, I recommend the authors not to use existing works as an excuse for doing something. There are many questionable practices in the machine learning community. I believe each decision should be justifiable without the context of other papers. That being said, I am still quite sceptical: Reinforcement learning considers an expected *cumulative* reward. With a Markov decision process that considers just one step, it is questionable to me how this interpretation is meaningful. I am not an expert in zero-order optimization, but I would be greatly surprised if other methods (there are certainly many other methods besides Bayesian optimization) would not yield results that are at least as good (likely better).
> >
> > However, I appreciate that the authors at least discuss this point in the revised manuscript. I will therefore consider this point to be addressed sufficiently. I would still like to motivate the authors to be more critical about this RL-based causal discovery framework in future works.
> >
> > **These methods can only identify the causal graph up to its Markov equivalent class (MEC), which contains potentially many DAGs inducing the same data.**
> >
> > I thank the authors for the explanation. The point *combined with additional parametric assumptions* is extremely important and I am happy that the authors include that in Lemma 4 of the revised manuscript. I guess by *parametric assumptions*, the authors mean something like, for example, non-linear additive Gaussian. It would not hurt to mention such specific, exemplary parametric assumptions, but I am also fine with the current formulation.
> >
> > Overall, I am content with the revised manuscript and I am willing to recommend accept. I once again thank the authors for their efforts.

---

> > > ### Author Response · Authors · 2025-01-14
> > >
> > > We sincerely appreciate your critical and constructive feedback, as well as your encouraging evaluation of our work. Your advices are invaluable and we will carefully consider them in our next revision, as well as our future developments. Once again, thank you for your time and effort in reviewing our paper.

---

### Review · Reviewer_YrDq · 2024-12-22

**Summary Of Contributions:**

This paper introduces ALIAS, which uses a NoCurl-type parameterization of DAGs in a reinforcement learning approach to causal discovery with observational data. This entails modifying the NoCurl parameterization of DAGs to obtain `Vec2DAG`, which essentially discretizes the NoCurl parameterization to binary adjacency matrices using the Heavyside function. A Bayesian information criterion-based score is then introduced as the reward for use with policy gradient-based methods such as A2C or PPO. Finally, extensive empirical evidence is presented showing the effectiveness of ALIAS using PPO and A2C in linear and nonlinear SEMs, including several ablation studies.

**Audience:**

Yes

**Broader Impact Concerns:**

None.

**Claims And Evidence:**

Yes

**Requested Changes:**

## Critical Changes

**Regarding Algorithm 1**  Algorithm 1 only presents the vanilla policy gradient. This can be misleading, since VPG is not the method typically used or advocated for (e.g., only A2C and PPO appear in the experiments). Please revise accordingly, for example retitling as "ALIAS with Vanilla Policy Gradient", or editing line 4 to say "Update policy, for example with vanilla policy gradient: [...]".

**Standard Error vs. Standard Deviation** In the figures, it is claimed that standard error is shown, but in the tables, it is claimed that the standard deviation is shown. But these seem to be quoting the same value; for example, the quoted standard error of DAGMA in Figure 3 with 1000 samples and 30 nodes seems to be ~14, which is closer to the quoted standard deviation of 17.8 than the corresponding standard error of ~8. This would also be more consistent with the existing literature. Please double-check that the correct value is quoted.

**Use `\citep` Where Appropriate** There are several places that `\citep` should be used, including: at the top of page 2 for (Zhu et al., 2020) and (Wang et al., 2021); at the beginning of section 2.2 for (Scwarz, 1978; ...); at the beginning of section 3.3 for (Zheng et al., 2018; ...). There might be some others, and I suspect this is because of calls to `\cite` rather than `\citet` directly, which should be searchable.

**Title** The title has "Journal Submissions" at the end. Please remove this.

**Experiments** Please include additional experimental evidence, as mentioned in Weaknesses.

## Minor Changes

**Regarding Related Work** I found the overview of related works in Sections 1 and 2 somewhat terse; in contrast, I found Appendix A well-written and helpful, especially in understanding the RL-based methods that I'm less familiar with. I think it would help the readability of the manuscript if Appendix A were to appear in the main text, especially since TMLR does not have page limits. This is ultimately subjective, though, and merely a suggestion.

Section 2.2 says that other methods use $l_0$ regularization, but typically actually use $l_1$ regularization.

# References

Yu, Y., Gao, T., Yin, N., & Ji, Q. (2021). DAGs with No Curl: An efficient DAG structure learning approach. In International Conference on Machine Learning (pp. 12156-12166). PMLR.

Waxman, D., Butler, K., & Djurić, P. M. (2024). Dagma-DCE: Interpretable, Non-Parametric Differentiable Causal Discovery. IEEE Open Journal of Signal Processing.

Zheng, X., Dan, C., Aragam, B., Ravikumar, P., & Xing, E. (2020). Learning sparse nonparametric dags. In International Conference on Artificial Intelligence and Statistics (pp. 3414-3425). PMLR.

**Strengths And Weaknesses:**

## Strengths

`Vec2DAG` is well-motivated, as is ALIAS. Moreover, the authors provide substantial empirical evidence that ALIAS works well, with impressive results in a variety of settings. While this method is more expensive than some continuous optimization-based methods, it is considerably faster than existing RL-based methods (and, of course, combinatorial search space methods).

## Weaknesses

Most of my concerns have to do with clarity and some parts of the experimental setup. I anticipate that these are all relatively minor.

**Runtime** On large graphs, the proposed method has a runtime more than 10 times longer than many other methods, and is over 100 times slower than DAGMA, in particular. This is especially a drawback in the linear case, where methods such as NOTEARS or DAGMA simplify considerably and run fast on CPU, whereas the implemented versions of ALIAS still require optimizing an MLP for the policy.

**Details Lacking in Presentation** There are some omitted details that I think should appear in the manuscript. For example, the definition of $E(z)$ and $p(z)$ are not provided; I had to look in the code to realize that $p(z)$ is just the first $d$ components of $z$, and $E(z)$ is an (triangular) unvec of the remaining components. This should be explained more clearly in the text.

Likewise, there are no details provided surrounding the GP regression in the nonlinear case: what kernels are used? Are approximate GPs used to deal with the poor scaling with respect to $\\lvert \\mathcal{D} \\rvert$?

**Originality of `Vec2DAG`** I find the originality of `Vec2DAG` to be overstated, for example in the introduction where NoCurl is not mentioned at all. Introducing the edge potential and computing $E(z) + E(z)^\\top$ is the straightforward way to parameterize a symmetric matrix without diagonal elements (which is admissable by Remark 3.6 of Yu et al. (2021)), and the remaining novelty is adding the Heavyside function, which is also a straightforward way to obtain binary adjacency matrices. Likewise, the proofs are essentially the same as in NoCurl. Originality is not a big concern for acceptance here, but I think the inspiration of `Vec2DAG` should be stated earlier and more prominently.

**Lack of Clarity in Some Proofs** I found the proofs sound, but think their clarity can be considerably improved. For example, consider Lemma 4: $p_{i}$ and $E_{ij}$ are given, but it is not explicitly computed that the hypothesized DAG is recovered. The actual computations provide some clarity into why this works, though: taking E as constructed provides the *undirected* version of the causal graph, minus the diagonal. And since
$\\text{grad}(p)\_{ij} = p\_{j} - p\_{i}$, it is clear that $\\text{grad}(p)_{ij} > 0$ if and only if $p_j > p_i$, which fits into the ancestry condition mentioned. Thus, in this construction, $E$ constructs the undirected causal graph, and $p$ encodes the directionality. Then, in the Hadamard product, we mask the undirected causal graph by the causal directions. Note that this is quite similar to the interpretation given in the corresponding proof of Yu et al. (2021). Similarly, the conditions on $\\beta$ in Lemma 1/5 are not straightforward to interpret, but important for understanding the lemma's actual implications.

## Questions

**Regarding Synthetic Data** In several places, it is claimed that "to make this setting more challenging" a wider range of edge weights are used, i.e., generating from $\\mathcal{U}([-5, -2] \\cup [2, 5])$ instead of $\\mathcal{U}([-2, -0.5] \\cup [0.5, 2.0])$. I'm concerned that this may inadvertently cause problems with other methods by fundamentally changing the $l_1$ penalization, as well as the optimal learning rate. Could the authors also provide some experiments using the default hyperparameters *and* the standard generating process?

**On Training of Sparsity Regularizers** Likewise, the standard in the literature is to leave a reasonable but untuned sparsity regularization term. Meanwhile, Table 18 shows that the proposed method is extremely sensitive to the $l_0$ penalty term, with the optimal value used in all experiments. What would be the corresponding effect of tuning sparsity penalties in other methods?

**On Runtime Setting** For Section 5.5/Figure 4, what random datasets are used (i.e., linear or nonlinear)? If non-linear, what are the effects of using larger MLPs in DAGMA/NOTEARS, so that all runtimes are similar?

**On GP-Based Nonlinear SEMs** One could argue that using a GP-based generative model in the synthetic data is unfair to DAGMA/NOTEARS/etc., which use MLPs as interpolators. Indeed, in the NOTEARS+ paper (Zheng et al., 2020), NOTEARS seems to be stronger on MLP-based SEMs than GP-based SEMs when compared to other methods (c.f. Figure 1, or Table 2 of Zheng et al., 2020). This is somewhat addressed in Appendix E.4, but it would be better if models were also tested on the "MLP" SEMs used in NOTEARS/DAGMA (note that these may, on the other hand, be somewhat favorable to NOTEARS/DAGMA from an identifiability standpoint, c.f. Section VI.C of Waxman et al., 2024).

---

> ### Author Response · Authors · 2025-01-12
> **Response to Reviewer YrDq (1/4)**
>
> We sincerely thank Reviewer YrDq for the meticulous review of our manuscript and for providing exceptional comments and suggestions. We have carefully incorporated all of your suggested amendments, and in the rebuttal below, we address each of your concerns and explain the changes we have made, point by point.
>
> **Runtime. On large graphs, the proposed method has a runtime more than 10 times longer than many other methods, and is over 100 times slower than DAGMA, in particular. This is especially a drawback in the linear case, where methods such as NOTEARS or DAGMA simplify considerably and run fast on CPU, whereas the implemented versions of ALIAS still require optimizing an MLP for the policy.**
>
> We would like to clarify that our study intentionally trades off runtime for accuracy, with a focus on improving the reliability of causal discovery while minimizing overhead. While existing methods are fast, they often lack accuracy, especially when handling complex data and graphs. Specifically, existing causal discovery methods are predominantly gradient-based, which explore graphs only along the gradient path. This limits the number of candidate graphs considered, leading to lower accuracy. In contrast, our approach leverages the exploration-exploitation capabilities of RL to explore a broader range of DAGs, resulting in a larger candidate set and a higher likelihood of finding better solutions. As a result, while our method may not be as fast as gradient-based methods, it achieves significantly better accuracy. For instance, on dense graphs, large graphs, and nonlinear data, our method attains ideal SHD, whereas gradient-based methods produce much higher errors (see Tables 2 and 4).
>
> This trade-off is essential because the primary goal of causal discovery is more about recovering the correct DAG, and less about runtime. In real-world applications, we believe that causal discovery may not be typically performed on multiple large datasets at once, and data is rarely simple. Therefore, we believe that spending a few hours on a large dataset is acceptable, provided it leads to accurate results, rather than relying on fast methods that may fail to handle complex data effectively. This justifies the need to sacrifice some runtime for improved accuracy in order to advance causal discovery.
>
> That said, we do acknowledge the importance of runtime, and thus we have also worked to optimize the computational efficiency of our method. As a result, its speed far exceeds existing RL-based methods for any given computational budget (Figures 4a), where it can generate a DAG up to more than 2000 times faster compared with CORL and RCL-OG. Additionally, Figure 4b shows that for graphs of up to 50 nodes, our method takes only under ten minutes and is even faster than some gradient-based methods such as NOCURL and COSMO.
>
> **Details Lacking in Presentation. There are some omitted details that I think should appear in the manuscript. For example, the definition of $E(z)$ and $p(z)$ are not provided; I had to look in the code to realize that $p(z)$ is just the first $d$ components of $z$, and $E(z)$ is an (triangular) unvec of the remaining components. This should be explained more clearly in the text.**
>
> We would like to point out that we have mentioned that $z$ is the combination of $p$ and $E$, and we denote by $p(z)$ and $E(z)$ the node and edge potential components constituting $z$, respectively. However, following your suggestion, we have clarified it more in the revision:
> > Specifically, $\mathbf{p}(\mathbf{z})$ represents the node potential vector formed by the first $d$ elements of $\mathbf{z}$, while $\mathbf{E}(\mathbf{z})$ is the edge potential matrix, with the elements above the main diagonal derived from the last $\frac{d \cdot (d-1)}{2}$ elements of $\mathbf{z}$ (see our code in Figure 5).
>
> **Likewise, there are no details provided surrounding the GP regression in the nonlinear case**
>
> We would like to clarify that we use vanilla GP regression provided by scikit-learn with the commonly used RBF kernel. The length scale is learnable, and we set the regularization hyperparameter $\alpha=1$ for nonlinear data, as the noise variance is close to one, which is similar to CORL and RCL-OG. This clarification has been added to the revised manuscript.

---

> ### Author Response · Authors · 2025-01-12
> **Response to Reviewer YrDq (2/4)**
>
> **Originality of Vec2DAG. [...] I think the inspiration of Vec2DAG should be stated earlier and more prominently.**
>
> Thank you for bringing this to our attention. We would like to clarify that we did not intentionally overlook the inspiration of NoCurl in our manuscript. In fact, we did acknowledge it in the Background, Related Works, Method sections, as well as in the proofs. As per your suggestion, we have now referenced the line of work on constraint-free DAG representations, pioneered by NoCurl, earlier in the Introduction of the revised manuscript.
>
> **Lack of Clarity in Some Proofs. I found the proofs sound, but think their clarity can be considerably improved. For example, consider Lemma 4: $p_i$ and $E_{ij}$ are given, but it is not explicitly computed that the hypothesized DAG is recovered [...]**
>
> Following your suggestion, we have included a verification step in the proof to show that the constructed pair of potentials yields the exact desired DAG. Thank you for your kind suggestion.
>
> **Similarly, the conditions on $\beta$ in Lemma 1/5 are not straightforward to interpret, but important for understanding the lemma's actual implications.**
>
> Thank you for your valuable feedback. We have added an interpretation of Lemma 1 that explains the conditions on $\beta$:
> > Intuitively, it indicates that scaling the potential by any positive constant $\alpha$ results in the same DAG ($\text{Vec2DAG}(\mathbf{z}) = \text{Vec2DAG}(\alpha \cdot \mathbf{z})$), and translating the potential by an amount $\beta$ (which can be large, provided it does not change the ordering of $\mathbf{p}$ or the element-wise positivity of $\mathbf{E}$) also results in the same DAG ($\text{Vec2DAG}(\mathbf{z}) = \text{Vec2DAG}(\mathbf{z} + \beta)$).
>
> **Regarding Synthetic Data. In several places, it is claimed that "to make this setting more challenging" a wider range of edge weights are used [...]. I'm concerned that this may inadvertently cause problems with other methods by fundamentally changing the penalization, as well as the optimal learning rate. Could the authors also provide some experiments using the default hyperparameters and the standard generating process?**
>
> We would like to clarify that we did also use the common weight range $[0.5, 2]$ in many of our experiments, particularly on dense and large graph datasets (Table 2), varying sample sizes (Figure 3), runtime analysis (Figure 4b), etc. In these scenarios, our method also consistently outperforms all baselines, including gradient-based approaches. Importantly, we apply the same set of untuned default hyperparameters across datasets with both large and small weight ranges and still achieve ideal SHDs.
>
> That said, in response to your request, we have included additional experiments in Appendix D.6 of the revised manuscript, comparing the causal discovery performance of all methods on the standard weight range $[0.5, 2]$ with the larger range $[2, 5]$ under varying graph sizes. We report parts of our findings, specifically for 30-node ER-2 graphs, in the table below (the numbers are mean ± standard error over 5 independent runs).
>
> || SHD|| FDR|| TPR||
> | :------------------------ | :----------- | :---------- | :----------- | :---------- | :----------- | :---------- |
> | **Method \ Weight range** | **[0.5, 2]** | **[2, 5]**  | **[0.5, 2]** | **[2, 5]**  | **[0.5, 2]** | **[2, 5]**  |
> | NOTEARS| 12.4±3.7     | 52.0±9.2    | 0.09±0.04    | 0.46±0.05   | 0.86±0.03    | 0.7±0.05    |
> | DAGMA| 0.0±0.0      | 0.2±0.2     | 0.0±0.0      | 0.0±0.0     | 1.0±0.0      | 1.0±0.0     |
> | NOCURL| 7.2±2.6      | 11.6±0.5    | 0.08±0.03    | 0.11±0.01   | 0.93±0.02    | 0.91±0.01   |
> | COSMO| 8.8±3.1      | 3.2±2.3     | 0.11±0.04    | 0.04±0.03   | 0.95±0.01    | 0.99±0.01   |
> | RL-BIC| 2.6±1.3      | 33.0±9.1    | 0.04±0.02    | 0.35±0.11   | 0.99±0.01    | 0.69±0.16   |
> | RCL-OG| 14.4±5.0     | 63.5±16.6   | 0.18±0.06    | 0.51±0.1    | 0.92±0.03    | 0.79±0.08   |
> | CORL| 0.0±0.0      | 15.4±1.7    | 0.0±0.0      | 0.21±0.02   | 1.0±0.0      | 1.0±0.0     |
> | **ALIAS (Ours)**| **0.0±0.0**  | **0.0±0.0** | **0.0±0.0**  | **0.0±0.0** | **1.0±0.0**  | **1.0±0.0** |
>
> The results indeed confirm our anticipation that larger weights are more challenging, where most baselines experience a significant increase in SHD for larger weights. Meanwhile, the results also demonstrate the insensitiveness of our approach to the weight range, as it always achieves zero SHD for both cases.

---

> ### Author Response · Authors · 2025-01-12
> **Response to Reviewer YrDq (3/4)**
>
> **On Runtime Setting. For Section 5.5/Figure 4, what random datasets are used (i.e., linear or nonlinear)?**
>
> We would like to clarify that the runtime experiment in Figure 4 utilizes standard linear-Gaussian datasets with the commonly used weight range $[0.5, 2]$. This clarification has been included in the revised manuscript. Thank you for bringing this to our attention.
>
> **On Training of Sparsity Regularizers. Likewise, the standard in the literature is to leave a reasonable but untuned sparsity regularization term. Meanwhile, Table 18 shows that the proposed method is extremely sensitive to the penalty term, with the optimal value used in all experiments.**
>
> We would like to clarify that we did follow the standard in the literature by employing an untuned set of hyperparameters across all experiments. Specifically, as mentioned in section "Experiment Setup", for our main experiments, we used the BIC score (as also done for RL-BIC, CORL, and RCL-OG), which does not require selecting a regularization weight and thus there was no tuning at all for our method. Therefore, our comparisons remain fair, as no method is tuned.
>
> Regarding Table 18, the purpose is simply to demonstrate the impact of varying regularization strengths specifically for the LS score, which we included to showcase our method's ability to use different scoring functions. It should be noted that we did not use the LS score in our main comparative experiments. Notably, the default value for this hyperparameter was chosen prior to the ablation study shown in Table 18, and we included it into the table because the corresponding data was already available. More particularly, we selected a small default regularization coefficient for the LS score to lightly penalize extra edges, as the pruning step can effectively eliminate most other redundant edges. This aligns with our ablation results in Table 18, which show that our SHD remains very low for small regularization coefficients.
>
>
> **On GP-Based Nonlinear SEMs. One could argue that using a GP-based generative model in the synthetic data is unfair to DAGMA/NOTEARS/etc., which use MLPs as interpolators. Indeed, in the NOTEARS+ paper (Zheng et al., 2020), NOTEARS seems to be stronger on MLP-based SEMs than GP-based SEMs when compared to other methods (c.f. Figure 1, or Table 2 of Zheng et al., 2020). This is somewhat addressed in Appendix E.4, but it would be better if models were also tested on the "MLP" SEMs used in NOTEARS/DAGMA (note that these may, on the other hand, be somewhat favorable to NOTEARS/DAGMA from an identifiability standpoint, c.f. Section VI.C of Waxman et al., 2024).**
>
> We have incorporated your suggestion by adding a new Section D.5, which includes additional nonlinear experiments with MLPs. Specifically, similar to the GP data, we consider 10-node ER-4 graphs with MLP SEMs with standard Gaussian noise as used in NOTEARS and DAGMA. These MLPs have one hidden layer with 100 units and sigmoid activation, and we generate 1,000 samples per dataset.
>
> We compare our ALIAS variant that uses GP as the interpolator (maintaining the exact configuration as in the GP experiments, but now using the equal variance variant of the BIC score), against gradient-based baselines that support MLPs as is. These include NOTEARS, DAGMA, and COSMO, all of which are set with the same configuration (one hidden layer of 10 units with sigmoid activation). The results are summarized in the table below (the numbers are mean ± standard error over 5 random simulations).
>
> ||No pruning|||CAM Pruning|||
> |-|-|-|-|-|-|-|
> | **Method**       | **SHD ($\downarrow$)** | **FDR ($\downarrow$)** | **TPR ($\uparrow$)**| **SHD ($\downarrow$)** | **FDR ($\downarrow$)** | **TPR ($\uparrow$)**|
> | NOTEARS          | 11.0±1.2               | 0.21±0.01              | 0.93±0.04           | 10.0±1.9               | 0.11±0.03              | 0.80±0.04           |
> | DAGMA            | 7.6±1.7                | 0.09±0.03              | 0.84±0.05           | 8.8±1.6                | 0.07±0.03              | 0.78±0.04           |
> | COSMO            | 9.4±1.0                | 0.14±0.01              | 0.87±0.03           | 9.8±1.2                | 0.09±0.02              | 0.79±0.02           |
> | **ALIAS (Ours)** | **0.8±0.4**            | **0.01±0.01**          | **0.99±0.01**       | **3.8±0.4**            | **0.0±0.0**            | **0.89±0.01**       |
>
> The results for NOTEARS and DAGMA align with their reported results (Figure 13 of DAGMA paper), while the results for COSMO are even better than reported (Appendix E.13 of COSMO paper). More importantly, despite the potential disadvantage of using GP regression for MLP-based SEMs, our method still achieves the lowest SHD compared to all three baselines, which are specifically designed to model MLP data.

---

> ### Author Response · Authors · 2025-01-12
> **Response to Reviewer YrDq (4/4)**
>
> **Algorithm 1 only presents the vanilla policy gradient. This can be misleading, since VPG is not the method typically used or advocated for (e.g., only A2C and PPO appear in the experiments). Please revise accordingly, for example retitling as "ALIAS with Vanilla Policy Gradient", or editing line 4 to say "Update policy, for example with vanilla policy gradient: [...]".**
>
> We have revised the manuscript as per your suggestion, explicitly highlighting that Algorithm 1 demonstrates ALIAS using the vanilla policy gradient. Thank you for your feedback.
>
> **Standard Error vs. Standard Deviation. In the figures, it is claimed that standard error is shown, but in the tables, it is claimed that the standard deviation is shown. But these seem to be quoting the same value; for example, the quoted standard error of DAGMA in Figure 3 with 1000 samples and 30 nodes seems to be ~14, which is closer to the quoted standard deviation of 17.8 than the corresponding standard error of ~8. This would also be more consistent with the existing literature. Please double-check that the correct value is quoted.**
>
> Thank you for your suggestion. Following your recommendation, we have reviewed and ensured the consistent use of standard error throughout the manuscript, adhering to common practice in the literature.
>
> **Use \citep Where Appropriate. There are several places that \citep should be used, including: at the top of page 2 for (Zhu et al., 2020) and (Wang et al., 2021); at the beginning of section 2.2 for (Scwarz, 1978; ...); at the beginning of section 3.3 for (Zheng et al., 2018; ...). There might be some others, and I suspect this is because of calls to \cite rather than \citet directly, which should be searchable.**
>
> Thank you for your suggestion. We have reviewed the manuscript and replaced \cite with \citep wherever appropriate in the revised version.
>
> **Please include additional experimental evidence, as mentioned in Weaknesses**
>
> As per your request, we have included the following additional experiments:
> - Appendix D.5: Results on nonlinear data with MLPs
> - Appendix D.6: Results on regular weight range $\mathcal{U}\left(\left[-2,-0.5\right]\cup\left[0.5,2\right]\right)$
>
> **Regarding Related Work. I found the overview of related works in Sections 1 and 2 somewhat terse; in contrast, I found Appendix A well-written and helpful, especially in understanding the RL-based methods that I'm less familiar with. I think it would help the readability of the manuscript if Appendix A were to appear in the main text, especially since TMLR does not have page limits. This is ultimately subjective, though, and merely a suggestion.**
>
> We would like to explain that during the submission, we were given the option to choose between a regular submission (12 pages) and a long submission (more than 12 pages). We opted for the regular submission, so we had to move our Related Work section to the Appendix due to the space limitation. However, in the revision, we have relocated the Related Work section to Section 2 and will make further adjustments to the length if necessary.
>
> **Section 2.2 says that other methods use $l_0$ regularization, but typically actually use $l_1$ regularization.**
>
> We have clarified our phrasing with:
> > A simpler yet widely adopted alternative is the least squares (LS) ([...]). With an additional $l_{0}$ regularization, we define the LS score as [...]
>
> We hope this response sufficiently addresses your concerns and are open to further discussions to resolve any remaining issues. In addition, we kindly request the chance to address any additional feedback you may have before a final decision is reached.

---

> > ### Comment · Reviewer_YrDq · 2025-01-13
> >
> > Thank you for the quick and detailed reply. I think the revised manuscript is significantly improved in clarity and content. I have a few (small) remaining points based on the revised manuscript.
> >
> > ---
> > ## Old Comments
> >
> > > Specifically, as mentioned in section "Experiment Setup", for our main experiments, we used the BIC score (as also done for RL-BIC, CORL, and RCL-OG), which does not require selecting a regularization weight and thus there was no tuning at all for our method.
> >
> > I see, thank you for the clarification! I think I might have misunderstood based on Table 7, which the authors may consider revising. While I understand the meaning now, the column "linear data" specifically states, e.g., Table 3, and an $l_0$ regularization. Perhaps there should be separate columns for "linear data using the BIC" and "linear data using the LS score," or some other way to disambiguate when BIC/LS were used.
> >
> > ## New Comments
> >
> > **Small Comment in Section 5.1** I find the newly added Section 5.1 very nice. However, I request a small change to the quote
> >
> > > Since RL is the only black-box optimization approach that has been studied in the score-based causal discovery literature (up to when our manuscript is written) [...]
> >
> > While I am not immediately aware of another method (e.g., BayesOpt) being used (at least, in the observational data case), I find it a bit strong and unnecessary to state authoritatively that no other method has been studied. I request a qualifier such as "to the best of our knowledge" is added.
> >
> > Also, "black-box" is consistently used, except for in the phrase "which is one of the most popular blackbox optimization methods," where the hyphen is missing.
> >
> > **Lemma 4 Wording** The first sentence of Lemma 4 is a bit difficult to parse. I'd recommend revising to something like "assume causal identifiability and causal minimality, that is, there is a unique causal model with no redundant edges that generates the observed data. Further, assume that the BIC score is used to define the reward $R(z)$ as in Eq. (6). [...]"

---

> > > ### Author Response · Authors · 2025-01-14
> > >
> > > We immensely appreciate your detailed additional feedback, and we will immediately integrate your recommendations into our next revision. Thank you once again for your time and effort in reviewing our paper!

---

### Review · Reviewer_y43i · 2025-01-08

**Summary Of Contributions:**

This paper introduces ALIAS, an RL-based framework for causal discovery (CD) that generates DAGs in a single step without acyclicity constraints by mapping a continuous vector space onto the space of all DAGs, using Vec2DAG. The authors prove surjectivity of their parameterization, integrate it with policy-gradient algorithms, and empirically demonstrate superior performance over previous CD methods on both synthetic and real-world datasets.

**Audience:**

Yes

**Claims And Evidence:**

Yes

**Requested Changes:**

1. It would be great if the authors could compare their method with constraint-based methods in causal discovery, like the most often-used one, PC (Spirtes et al. (2000)), or the more recent recursive causal discovery methods, such as [RSL or MARVEL (Mokhtarian et al. (2024))](https://rcdpackage.com/).
2. Could the authors prove that the DAG returned from ALIAS is in the MEC? Or, perhaps that the DAG returned is precisely the underlying DAG given some restrictions on the FCM, such as it being a linear Guassian model.

### Minor comments
1. The paper title in the PDF on the first page is "Reinforced DAG Learning without Acyclicity Constraints Journal Submissions." I am guessing that the "Journal Submissions" part is not a part of the title and is a typo.
2. The first paragraph of the second page starting with "from Zheng et al. (2018) into the reward..." is missing punctuations.

**Strengths And Weaknesses:**

### Strengths
1. Use of Vec2DAG and generating DAGs in one step is efficient and interesting.
2. The paper offers a theoretical proof of the surjectivity of Vec2DAG and scaling/translation invariance.
3. Extensive experiments on small to large graphs that show that the proposed ALIAS method outperforms other score-based causal discovery methods.

### Weaknesses & questions
1. No comparisons are made with constraint-based causal discovery methods.
2. There are no theoretical guarantees on whether the returned DAG from ALIAS is in the MEC.
3. The ALIAS method is slow, especially on larger graphs, taking more than 1.5 hours to process one graph of 200 nodes.

---

> ### Author Response · Authors · 2025-01-12
> **Response to Reviewer y43i (1/2)**
>
> We sincerely thank Reviewer y43i for your thorough review of our manuscript and for offering valuable comments and suggestions. We have thoughtfully implemented all of your recommended changes and, in the following rebuttal, we address each of your concerns in detail, explaining the revisions made point by point
>
> **The ALIAS method is slow, especially on larger graphs, taking more than 1.5 hours to process one graph of 200 nodes.**
>
> We would like to clarify that our study prioritizes accuracy over runtime, aiming to enhance the reliability of causal discovery while keeping computational overhead reasonable. Although existing methods are faster, they often compromise on accuracy, particularly when dealing with complex datasets and dense graphs. Specifically, most current causal discovery approaches rely on gradient-based techniques, which limit exploration to the gradient path, reducing the number of candidate graphs considered and consequently lowering accuracy. In contrast, our method leverages the exploration-exploitation capabilities of RL to examine a wider range of DAGs, generating a larger candidate set and increasing the likelihood of identifying better solutions. As a result, while our approach may not match the speed of gradient-based methods, it delivers significantly higher accuracy. For instance, on dense graphs, large graphs, and nonlinear data, our method achieves ideal SHD, whereas gradient-based methods produce substantially higher errors (see Tables 2 and 4).
>
> This trade-off is crucial, as the primary goal of causal discovery is more about recovering the correct DAG, rather than optimizing runtime. In practical applications, causal discovery is unlikely to be performed on multiple large datasets at once, and datasets are often complex. Therefore, dedicating a few hours to analyze a large dataset is a reasonable compromise if it ensures more accurate results, as opposed to relying on faster methods that may not work well with complex data. This highlights the importance of prioritizing accuracy over speed to advance causal discovery.
>
> Nevertheless, we do not completely disregard the importance of runtime. In fact, we have also made significant efforts to optimize the computational efficiency of our method. As a result, it is much faster than existing RL-based approaches for any given computational budget, generating DAGs over 2000 times faster than CORL and RCL-OG (Figure 4a). Moreover, as shown in Figure 4b, our method processes graphs with up to 50 nodes in merely under ten minutes and is even faster than some gradient-based methods, including NOCURL and COSMO.
>
> **Could the authors prove that the DAG returned from ALIAS is in the MEC? Or, perhaps that the DAG returned is precisely the underlying DAG given some restrictions on the FCM, such as it being a linear Guassian model.**
>
> Following your request, in the revised manuscript, we have proven that the optimal policy, obtained by maximizing the expected reward in our method, will almost surely output the true underlying DAG in the large sample limit, provided that the causal model is identifiable and the BIC score is used. Below, we outline the key intuitions behind this result.
>
> First, the consistency of the scoring function ensures that only the true DAG receives the highest score. This is guaranteed for identifiable causal models with causal minimality when using the BIC score, which is known to be consistent. We have updated Section "DAG Scoring" to include this additional information (Lemma 1), and the detailed proof is provided in Appendix B.1. Our experiments further support this result, as for all identifiable causal models used in our experiments (e.g., linear-Gaussian models with equal noise variance and nonlinear additive noise models), our method correctly identifies the true DAG in nearly all cases.
>
> Second, we have shown that the optimal policy, which maximizes the expected reward, outputs the highest-scoring DAG in the large sample limit. Since the optimal policy assigns all its probability mass to the DAG with the highest score, and only the true DAG has the highest score (almost surely in the large sample limit), the optimal policy will only output the true DAG. This has been explained in Lemma 4 of Section 5.2 in the revision, with the detailed proof included in Appendix B.5.

---

> ### Author Response · Authors · 2025-01-12
> **Response to Reviewer y43i (2/2)**
>
> **It would be great if the authors could compare their method with constraint-based methods in causal discovery, like the most often-used one, PC (Spirtes et al. (2000)), or the more recent recursive causal discovery methods, such as RSL or MARVEL (Mokhtarian et al. (2024)).**
>
> We would like to note that the PC method was not included in our initial submission due to its poor performance. However, following your suggestion, we have conducted additional experiments to compare our approach with constraint-based methods, including the classical PC algorithm as well as more advanced methods, namely RSL and MARVEL. These results have been added in the new Section D.7 of the Appendix. For these experiments, we used gCastle's implementation for PC and the implementations provided at https://rcdpackage.com/ for RSL and MARVEL.
>
> Specifically, we evaluate these methods on linear-Gaussian datasets with varying scales and densities, and the sample size is fixed to 1,000. The baseline methods are configured according to their recommended settings, including using the Fisher's z test with significance levels of 0.05 for PC and $\frac{2}{d^2}$ for RSL and MARVEL, where $d$ is the number of nodes. Since constraint-based methods may not orient all edges, for a fair evaluation, we compare the undirected skeletons of the estimated and true graphs. Particularly, the evaluation metrics include SHD, precision, recall, and F1 scores between the estimated and true skeletons. The results are presented in the table below (the numbers are mean ± standard error across 5 random simulations).
>
> | Data                         | Method                           | Skeleton SHD ($\downarrow$) | Skeleton Precision ($\uparrow$) | Skeleton Recall ($\uparrow$) | Skeleton F1 ($\uparrow$) |
> |------------------------------|----------------------------------|-----------------------------|---------------------------------|------------------------------|----------------------------------------|
> | Sparse graphs (30-node ER-2) | PC (Spirtes et al., 2000)        | 25.8±2.78            | 0.88±0.03                | 0.64±0.03             | 0.74±0.03        |
> |                              | MARVEL (Mokhtarian et al., 2021) | 15.6±2.09            | 0.93±0.02                | 0.79±0.03             | 0.85±0.02        |
> |                              | RSL (Mokhtarian et al., 2022)    | 13.8±1.53            | 0.9±0.02                 | 0.86±0.01             | 0.88±0.01        |
> |                              | **ALIAS (Ours)**                 | **0.0±0.0**              | **1.0±0.0**                  | **1.0±0.0**               | **1.0±0.0**          |
> | Dense graphs (30-node ER-8)  | PC (Spirtes et al., 2000)        | 216.0±3.56           | 0.65±0.02                | 0.13±0.01             | 0.21±0.02        |
> |                              | MARVEL (Mokhtarian et al., 2021) | 221.8±3.81           | 0.71±0.03                | 0.05±0.0              | 0.1±0.01         |
> |                              | RSL (Mokhtarian et al., 2022)    | 185.4±5.57           | 0.66±0.02                | 0.39±0.01             | 0.49±0.01        |
> |                              | **ALIAS (Ours)**                 | **0.4±0.24**             | **1.0±0.0**                  | **1.0±0.0**               | **1.0±0.0**          |
> | Large graphs (200-node ER-2) | PC (Spirtes et al., 2000)        | 216.6±10.46          | 0.87±0.01                | 0.57±0.02             | 0.69±0.01        |
> |                              | MARVEL (Mokhtarian et al., 2021) | 167.2±8.37           | 0.91±0.01                | 0.67±0.01             | 0.77±0.01        |
> |                              | RSL (Mokhtarian et al., 2022)    | 167.2±8.83           | 0.87±0.01                | 0.7±0.01              | 0.78±0.01        |
> |                              | **ALIAS (Ours)**                 | **0.8±0.58**             | **1.0±0.0**                  | **1.0±0.0**               | **1.0±0.0**          |
>
> It can be seen that our method demonstrates significantly higher accuracy than constraint-based approaches in recovering the skeleton, achieving near-zero skeleton SHD across all scenarios, even on very large graphs of 200 nodes. In contrast, the baseline methods face considerable challenges, particularly when applied to large and dense graphs.
>
> We hope that this rebuttal adequately addresses your concerns and are open to further discussions to resolve any outstanding issues. Additionally, we would appreciate the opportunity to address any further feedback you may have before a final decision is made.

---

> ### Author Response · Authors · 2025-01-22
>
> Dear Reviewer y43i,
>
> We sincerely appreciate the time and effort you have dedicated to reviewing our paper. In our rebuttal, we have carefully addressed all of your concerns, including conducting additional comparative experiments with constraint-based baselines and providing theoretical guarantees for our framework's ability to recover the exact DAG.
>
> We hope our responses sufficiently address your feedback, and should you have any further questions or concerns, we would be more than happy to address them. Once again, thank you for your valuable insights to improving our work.

---

> > ### Comment · Reviewer_y43i · 2025-01-27
> > **Reply to authors' response**
> >
> > I thank the authors for their thorough and detailed response, and apologies for the delay on my part. I have reviewed the new lemma (Lemma 1) and the new experimental results. The authors have effectively addressed my concerns, and I have no further issues.

---

### Author Response · Authors · 2025-01-12
**Common response to all Reviewers**

Dear Reviewers,

We would like to express our sincere gratitude for the time and effort you have dedicated to reviewing our manuscript. In response to your valuable comments, we have thoroughly addressed your concerns in the respective reviews and significantly revised the manuscript to incorporate your feedback. For your convenience, **all changes and additions in the revision are clearly marked in red**. Below, we provide a summary of the major revisions we have made.

To enhance the clarity and quality of our manuscript, we have:

- Moved the Related Work section from the Appendix to Section 2.
- Revised the title to "Reinforcement Learning for Causal Discovery without Acyclicity Constraints."
- Clarified the theoretical claims and proofs.
- Expanded the explanation of the intuition behind Vec2DAG.
- Included additional details on GP regression.
- Elaborated further on the significance of RL in our approach (Section 5.1).
- Provided a proof of the consistency of the BIC score (Lemma 1 and Appendix B.1).
- Added a proof ensuring the guarantee of obtaining the true DAG in our method (Lemma 4 and Appendix B.5).
- Used standard error consistently across all results.

To strengthen the empirical evaluation of our work, we have:
- Included the source code for reproducibility of our experiments (Supplementary material), covering 13 of our experiments, which are easy to run.
- Added experiments on nonlinear data with MLPs (Appendix D.5).
- Added experiments on linear-Gaussian data with the small weight range $\mathcal{U}\left(\left[-2,-0.5\right]\cup\left[0.5,2\right]\right)$ (Appendix D.6).
- Incorporated a comparison with constraint-based methods (Appendix D.7).

We hope these revisions adequately address your concerns and reflect our commitment to improving the manuscript. We look forward to your feedback and are eager to engage in further discussions before a final decision is made.

---

### Decision · Action_Editor_JMnK · 2025-03-02

**Recommendation:** Accept with minor revision

**Comment:**

Reviewers after clarifying various aspects are in agreement for its acceptance. I agree with reviewer assessment mostly.

My principle concern that is outstanding is with respect to experiments. It has been pointed out in the literature that if you use random causal models with additive noise, ordering due to marginal variance of the variables is a valid topological order consistent with that of the true causal graph. So it would be great to actually do experiments where this **variance sortability** is not trivially possible. Please see this reference  https://arxiv.org/abs/2102.13647. The reference cited pointed out this issue with prior gradient based methods and there are various papers following this that examine this issue in more detail.

 I only ask authors to consider an experimental setting where this alternative cannot be trivially used for linear methods at least (following the experimental setting recommendation in that paper). I think variance sortability will also be an issue even in the Gaussian Process setting. So that experiment also may need reconsideration

I would like the authors to clarify the above in the revision.

**Audience:**

The method parameterizes DAGs by creating a surjective map from vectors to DAGs such that one can optimize a distribution that samples over continuous vectors and applying the map on top of it enables them to sample the space of DAGs. It extends the NoCurl paper to non linear SCMs.

This enables score based causal discovery that is enabled by policy gradient and can have immense value turning causal discovery into a differentiable learning enterprise.

I only have the two reservations above listed in "Claims and Evidence". Can you please address that in a minor revision ? I am in favor of acceptance pending that.

**Claims And Evidence:**

Claims in the paper are justified adequately - both theoretical and experimental. I have the following mild concerns which I hope the authors can address in a minor revision
a) Main numerical experiments (in comparison with previous baselines) are done with additive noise models.  There is a line of work (starting from the reference I provide here) https://arxiv.org/abs/2102.13647 that says that simulated DAGs with additive noise models naturally has marginal variance of the variables sorted in the right DAG order. Therefore, variance based sortability techniques are enough and no need for gradient based techniques (or other techniques ).

Can authors do one more ablation to show that their RL method does not suffer from **variance sortability** issues - you can follow the above paper or some papers that follow it which recommend some normalized settings (Table 3 experiments in the current draft may need change specifically).

b) The method is just a policy gradient theorem applied as a stand in for some zeroth order method. Authors have clarified this. But may be authors can modify section 5 to stress more directly on the method (policy gradient) than imply it is an RL algorithm. Uusually for some people there needs to some non trivial MDP to call it RL.